# Mitochondria are positioned at dendritic branch induction sites, a process requiring rhotekin2 and syndapin I

Jessica Tröger [1], Regina Dahlhaus[1,2], Anne Bayrhammer[1,3], Dennis Koch [1], Michael M. Kessels [1] ✉ & Britta Qualmann [1] ✉

Proper neuronal development, function and survival critically rely on mitochondrial functions. Yet, how developing neurons ensure spatiotemporal distribution of mitochondria during expansion of their dendritic arbor remained unclear. We demonstrate the existence of effective mitochondrial positioning and tethering mechanisms during dendritic arborization. We identify rhotekin2 as outer mitochondrial membrane-associated protein that tethers mitochondria to dendritic branch induction sites. Rhotekin2-deficient neurons failed to correctly position mitochondria at these sites and also lacked the reduction in mitochondrial dynamics observed at wild-type nascent dendritic branch sites. Rhotekin2 hereby serves as important anchor for the plasma membrane-binding and membrane curvature-inducing F-BAR protein syndapin I (PACSIN1). Consistently, syndapin I loss-of-function phenocopied the rhotekin2 loss-of-function phenotype in mitochondrial positioning at dendritic branch induction sites. The finding that rhotekin2 deficiency impaired dendritic branch induction and that a syndapin binding-deficient rhotekin2 mutant failed to rescue this phenotype highlighted the physiological importance of rhotekin2 functions for neuronal network formation.

Mitochondria are indispensable for proper function and survival of neurons[1]. In addition to mitochondrial ATP production required to meet the immense energy demands of neurons[2] and the exceptional challenges in maintaining energy homoeostasis, mitochondria play an important role in $Ca^{2+}$ homeostasis, the production and sequestration of reactive oxygen species, apoptosis and neurotransmitter metabolism[3–5]. Since neurons are morphologically complex cells, which develop extended axons and elaborate dendritic arbors ensuring a high degree of interconnectivity in neural networks, specialized mechanisms are required for efficient distribution of mitochondria[4–9]. Deficiencies in mitochondrial dynamics (mitochondrial transport, fission and fusion) impair the development of the nervous system and are causative in many neurological disorders[4,10,11]. A particular biological challenge is the guidance of mitochondria to sites of high-energy demands and $Ca^{2+}$ signaling, such as synapses, growth cones but also axonal and dendritic branching points. Interestingly, dendritic branch points have been reported to represent hotspots for $Ca^{2+}$ signals[12,13] and dendrite branch induction sites sometimes coincided with previous local $Ca^{2+}$ signals as well as with bursts of F-actin formation at the base of nascent dendritic protrusions[14].

Dendritic arborization involves F-actin formation by the actin nucleator Cobl (Cordon-bleu) at dendritic branch induction sites[14,15]. Cobl has been demonstrated to be controlled by $Ca^{2+}$/CaM[14]. The action of Cobl and its evolutionary distant ancestor Cobl-like[16] at nascent dendritic branch sites is choreographed by the F-BAR protein syndapin I (PACSIN1)[17–19], which specifically localizes at plasma membrane surfaces at nascent dendritic protrusions and accumulates with its cytoskeletal binding partners in

[1]Institute of Biochemistry I, Jena University Hospital - Friedrich Schiller University Jena, Nonnenplan 2-4, 07743 Jena, Germany. [2]Present address: Research Division for Neurodegenerative Diseases, Faculty of Medicine/Dentistry, Danube Private University, Steiner Landstraße 124, 3500 Krems-Stein, Austria. [3]Deceased: Anne Bayrhammer. ✉e-mail: Michael.Kessels@med.uni-jena.de; Britta.Qualmann@med.uni-jena.de

a spatially and temporally coordinated manner prior to dendritic branch induction[19,20].

While contributions of mitochondria in synapses are documented[6–9], a putative support of dendritic branch induction is far less clear. Studies reported thus far mostly aimed at correlating mitochondria with established dendritic branches[21,22]. This led to conflicting data on influences of general depletions or impairments of mitochondria on dendritic arborization and outgrowth[21,23–30]. Thus, mitochondrial organization and dynamics at nascent branch sites have to be resolved spatiotemporally. Progress is also hampered by lack of identifications of molecules that may provide spatio-temporal connections between nascent dendritic branch sites and mitochondria.

Our molecular mechanistic studies identify a physical and functional connection as well as a cooperation of mitochondrially localized rhotekin2 (RTKN2[31]) with syndapin I, which specifically and transiently concentrates on nascent dendritic branch sites. Detailed studies in neurons reveal the importance of rhotekin2 and syndapin I for correct mitochondrial recruitment to nascent dendritic branch sites indispensable for proper dendritic arbor development underlying neuronal network formation.

## Results

### Syndapin loss-of-function studies reveal an influence on mitochondrial morphology control

The molecular mechanisms eliciting dendritic branches are far from understood. Syndapin I loss-of-function impairs different aspects of neuronal cell shape and function including dendritic branching[18,32–35]. Yet, when we carefully analyzed hippocampal sections from *syndapin I* KO mice by transmission electron microscopy, we also noticed mitochondrial alterations (Fig. 1a, b; Supplementary Fig. 1a, b). Quantitative analyses demonstrated that cross-section areas of individual mitochondria were highly significantly larger in *syndapin I* KO compared to WT mice (Fig. 1c). Correspondingly, also the lengths of mitochondria were increased upon *syndapin I* KO (Fig. 1d). These defects in mitochondrial shaping were accompanied by a reduction in mitochondrial density in *syndapin I* KO compared to WT (Fig. 1e) and suggested some thus far unknown defects in mitochondrial fission in *syndapin I* KO mice.

Corresponding analyses of NIH3T3 cells subjected to siRNA-based knockdown[36] or overexpression of syndapin II (the rather ubiquitously expressed syndapin isoform[37,38]) showed that the identified mitochondrial changes represented a general function of the syndapin family. The syndapin II loss-of-function and syndapin I gain-of-function phenotypes in NIH3T3 cells are in line with the effects of syndapin I in neurons (Supplementary Fig. 2) irrespective of whether the mitochondrial analyses were conducted via immunostaining of the endogenous mitochondrial protein cytochrome c (Cytc) (Supplementary Fig. 2a-g,n-p) or via mitoGFP overexpression and use of the skeleton software tool (ImageJ) (Supplementary Fig. 2j-m, q-s). In line, mitoGFP signals and anti-Cytc-immunolabelings largely colocalized (Supplementary Fig. 2i).

Taken together, all three different functional data sets strongly suggested that syndapins affect mitochondria by some yet to be elucidated molecular mechanisms.

### Identification of rhotekin2 as syndapin binding partner

Yeast-two-hybrid (Y2H) screening with syndapin I as bait[39] to further elucidate molecular mechanisms underlying the mitochondrial phenotypes observed upon syndapin deficiencies did not identify any candidates with known roles in mitochondrial organization.

Isolated, however, were two rhotekin2 (RTKN2 isoform 2; name alternative to rhotekin2, pleckstrin homology domain-containing family K member 1 (PLEKHK1); NP_001346248.1)) clones. Clone#541 expressed rhotekin2 isoform 2 (marked as RTKN2' throughout) and

clone#336 expressed rhotekin2'[457-601] (Fig. 1f). Retransformations with isolated plasmids confirmed the Y2H hits by growth on drop-out plates and β-Gal activity. Using syndapin I[ΔSH3] as bait furthermore revealed that the syndapin I SH3 domain was required for the interaction (Fig. 1g).

Sequence similarities of rhotekin2 with rhotekin1 (RTKN)[40] are extremely limited (Supplementary Fig. 3a). The name rhotekin2 may thus be somewhat misleading. The suggested N-terminal Rho effector motif class 1 (REM-1) of rhotekin2 (aa1-74) failed to show any interaction with RhoA[41]. Highlighting the fundamental differences between the two similarly named proteins neither the predicted pleckstrin homology (PH) domain nor the C-terminal proline-rich domain of rhotekin2 (Fig. 1f) show any considerable sequence similarities to rhotekin1 (Supplementary Fig. 3a). Rhotekin2 was first identified as a human protein showing high mRNA levels in lymphoid tissues and cells[31]. Consistently, further studies reported high rhotekin2 mRNA levels in cancer cells and suggested some role in proliferation, cell cycle progression and/or apoptosis resistance[41–44].

Coprecipitation analyses with GFP-RTKN2'[457-601] and GST-syndapin I confirmed the identified syndapin I interaction (Fig. 1h). Also full-length syndapin II and III interacted with rhotekin2' in a specific and efficient manner (Fig. 1h). Reconstitutions with purified proteins demonstrated that GST-syndapin I specifically precipitated TrxHis-RTKN2' (Fig. 1i). The syndapin I/rhotekin2 interaction therefore is direct.

Rhotekin2 contains 3 PxxP motifs (Supplementary Fig. 3b). We hypothesized that the second one represented the syndapin binding site, as it was highly conserved among vertebrate rhotekin2 proteins and closely resembled syndapin binding sites identified in Cobl, ProSAP1/Shank2, ProSAP2/Shank3, Cobl-like and ankycorbin[18–20,34] (Fig. 1j). Coprecipitation analyses demonstrated that (untagged) full-length rhotekin2 lacking the identified motif (RTKN2[ΔKRAP]) indeed showed impaired binding to all syndapin family members (Fig. 1k; for anti-rhotekin2 antibody validation see Supplementary Fig. 4).

Coimmunoprecipitation experiments demonstrated the relevance of the rhotekin2 interaction with syndapins in cells. GFP-syndapin I as well as the short and long syndapin II isoforms, respectively, were coimmunoprecipitated with Flag-tagged rhotekin2' in a specific manner (Fig. 1l).

### Identification of rhotekin2 as protein associating with the outer mitochondrial membrane

In NIH3T3 fibroblasts, our affinity-purified anti-RTKN2 antibodies immunolabeled subcellular structures showing a clear spatial overlap (Pearson´s coefficient, 0.8) with the mitochondrial marker cytochrome c (Fig. 2a, b). Colocalization with mitochondria was also observed for endogenous rhotekin2 in primary neurons (Fig. 2c).

Mitochondria purifications from fibroblasts and from brain material of adult rats confirmed by independent means that endogenous rhotekin2 was present at mitochondria. In contrast to non-mitochondrial proteins, but similar to the mitochondrial protein cytochrome c oxidase (COX) IV, rhotekin2 was enriched in fractions containing crude and purified mitochondria, respectively (Fig. 2d, e).

Extractions of purified mitochondria with $Na_2CO_3$ suggested that rhotekin2 associates with the outer mitochondrial membrane (Fig. 2e). This is supported by high-resolution imaging of dendritic mitochondria in hippocampal neurons. The immunosignals of rhotekin2 surrounded those of the internal mitochondrial protein cytochrome c (Fig. 2f). However, as imaging alone is insufficient to resolve at which membrane side a membrane-associated protein is localized, we next conducted accessibility studies. Rhotekin2 was immunodetected at non-permeabilized purified mitochondria, whereas cytochrome c was only accessible for immunolabeling once the mitochondrial membranes had been permeabilized with Triton X-100. These experiments clearly demonstrated that rhotekin2 resided at the cytosolic side of the outer mitochondrial membrane (Fig. 2g, h).

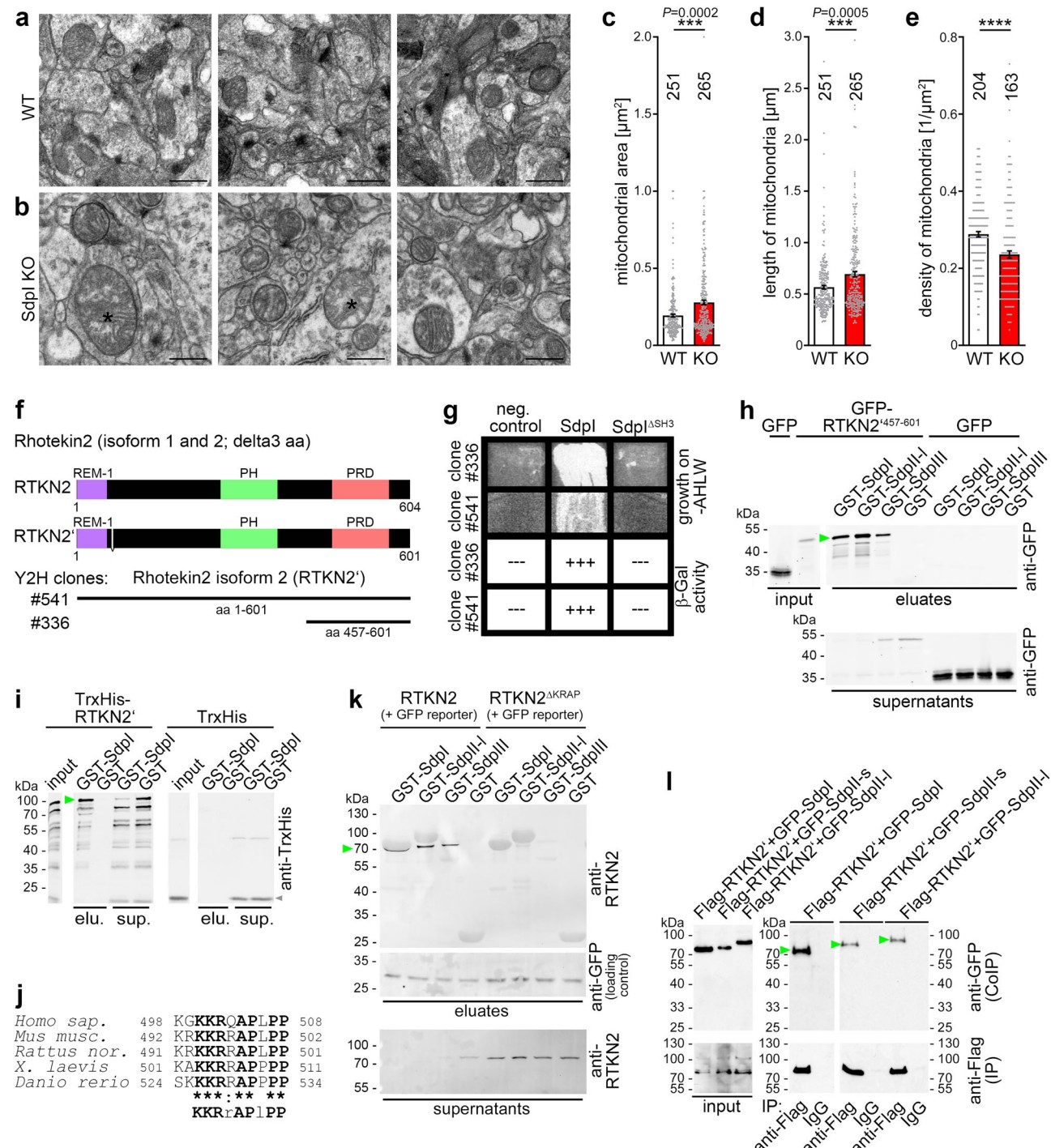

Reconstitution experiments with purified GST-rhotekin2' confirmed its mitochondrial interaction. The rhotekin2' association did hereby not rely on any proteins associated with the outer mitochondrial membrane, as also $Na_2CO_3$-extracted mitochondria-associated specifically with GST-rhotekin2' (Fig. 2i, j).

Flotation assays with purified components demonstrated rhotekin2's direct membrane binding ability. Rhotekin2 specifically floated together with liposomes to fraction 2 of density gradients (Fig. 2k, l).

### Rhotekin2 shapes mitochondria and this function is KRAP motif-dependent

The identification of rhotekin2 as novel mitochondrial protein and as binding partner for syndapins, which we found to have profound

impact on mitochondrial morphology (Fig. 1), prompted us to next study the impact of rhotekin2 on mitochondria. Rhotekin2 RNAi (for validation see Supplementary Fig. 4d-f) led to highly significantly larger mitochondria in dendrites of primary rat hippocampal neurons (Fig. 3a-e; Supplementary Fig. 5a-f; ****$P < 0.0001$, ****$P < 0.0001$ and **$P < 0.01$ in the three technically independent analyses). Length distribution analyses of mitoGFP-marked mitochondria were in line with the literature[45] for control cells and showed that especially mitochondria with lengths of >3 μm occurred more often upon rhotekin2 RNAi than in control cells (Fig. 3f). Since the mitochondrial length increase was also observed in live-microscopy studies (Supplementary Fig. 5f), the identified rhotekin2 phenotype clearly did not represent a fixation artifact.

**Fig. 1 | Syndapin I shapes mitochondria and associates with rhotekin2.**
**a, b** Transmission electron microscopy images of brain sections of WT (**a**) and
*syndapin I* KO (SdpI KO) mice (**b**) (age, approx. 20 weeks) showing mitochondria in
neurons of the hippocampal CA3 region. Asterisks mark examples of particularly
large mitochondria more frequently occurring in *syndapin I* KO samples. Bars,
500 nm. For enlarged electron microscopy (EM) images with marks of evaluations
for mitochondrial area and length see Supplementary Fig. 1. **c–e** Quantitative
analyses of the sizes (area, **c**; length, **d**) and densities of neuronal mitochondria (**e**)
in WT and *syndapin I* KO (KO) hippocampi. Data, mean ± SEM visualized as bar/dot
plots. WT, $n = 251$ and KO, $n = 265$ mitochondria (**c, d**). WT, $n = 204$ and KO, $n = 163$
images (**e**). Statistical significance, Mann-Whitney (**c–e**). *** $P < 0.001$; ****
$P < 0.0001$. For exact $P$ values ≥ 0.0001 see figure. For corresponding syndapin I
gain-of-function and syndapin II loss-of-function experiments of different kinds see
Supplementary Fig. 2. **f** Schematic representation of mouse rhotekin2 (RTKN2)
isoform 1 (RTKN2, NP_001346247.1) and isoform 2 (RTKN2′, NP001346248.1) and
of the two independent Y2H clones from an embryonal mouse (*mus musculus*) brain
library encoding for full length rhotekin2 (clone2541; RTKN2′) and a C-terminal
fragment thereof (clone3336; RTKN2′[457-601]; corresponds to aa460-604 in rhotekin2
isoform 1). REM-1, Rho effector motif class 1; PH, pleckstrin homology, PRD, proline-
rich domain. For sequence comparison see Supplementary Fig. 3. **g** Verification of
the syndapin I interaction of two prey clones (clone5336 and full-length clone3541)
isolated by Y2H screening with full-length syndapin I as a bait via retransformation
of isolated plasmids by reporter gene activity assessment (growth on quadruple
drop-out plates and β-Gal activity, respectively) demonstrating a SH3 domain-
dependent syndapin I interaction. pGBTK7 served as negative control for the bait

vectors. **h** Immunoblotting analyses of coprecipitations of GFP-RTKN2′[457-601] with
immobilized GST-syndapin I, II-l and III and GST (control), respectively. Green
arrowhead marks specifically coprecipitated GFP-RTKN2′[457-601]. **i** Immunoblotting
analyses of coprecipitation experiments with purified full-length fusion proteins of
syndapin I (GST-SdpI; immobilized) and rhotekin2′ (TrxHis-RTKN2′) clearly proving
a direct interaction. Green arrowhead marks specifically coprecipitated TrxHis-
rhotekin2′. GST and TrxHis (grey arrowhead) served as specificity controls.
**j** Alignment of rhotekin2 proteins from different vertebrates revealing a highly
conserved KKRrAPlPP ("KRAP") motif (human, Q8IZC4; mouse, NP001346247.1;
rat, NP_001101095; frog, Q5XGX5; zebrafish, Q5XIZ9). For rhotekin2 sequence and
domain details see Supplementary Fig. 3. **k** Coprecipitation analyses confirming
that (untagged) full-length rhotekin2 associates (green arrowhead) with immobi-
lized recombinant syndapins and demonstrating that rhotekin2[ΔKRAP] lacking the
amino acids 492-503 is deficient for complex formation with all three syndapin
isoforms and thus was exclusively detected in the supernatants. Very light grey,
thick bands visible in the anti-rhotekin2-immunoblotted eluates represent the GST
fusion proteins used for the coprecipitations. For validation of the anti-rhotekin2
antibody see Supplementary Fig. 4. (**l**) Immunoblotting analyses of coimmuno-
precipitation experiments from lysates of HEK293 cells coexpressing Flag-RTKN2′
together with GFP-syndapin I (GFP-SdpI) and a short and long isoform of syndapin II
(GFP-SdpII-s and GFP-SdpII-l), respectively. Specific coimmunoprecipitation of
RTKN2′/syndapin complexes (green arrowheads) were obtained by anti-Flag anti-
bodies but not by control IgGs. White lines/lanes indicate lanes omitted (**h, i**) and
therefore are not accompanied with additional standards. All experiments shown as
representative images (**h, i, k, l**) have been done at least twice with similar results.

---

The increase in mitochondrial length was also observed in the
axon of developing hippocampal neurons (Supplementary Fig. 6) and
in non-neuronal cells transfected with rhotekin2 RNAi (Supplementary
Fig. 7). The identified rhotekin2 loss-of-function phenotype thus
occurred cell compartment- and cell type-independently.

Rhotekin2 overexpression in NIH3T3 cells resulted in effects
opposite to rhotekin2 loss-of-function, as mitochondria were shorter
irrespective of how mitochondria were detected. The mitochondrial
density was increased by about 22% when compared to control cells
(Supplementary Fig. 8).

The increase in mitochondrial length upon rhotekin2 RNAi was
specifically attributed to rhotekin2 loss-of-function because co-
expression of RNAi-insensitive rhotekin2 (RTKN2*) rescued the phe-
notype completely in fibroblasts (Supplementary Fig. 7) and in hip-
pocampal neurons (Fig. 3a-e; Supplementary Fig. 5a-e).

Importantly, the successful restoration of normal mitochondrial
size was not obtained when the cells were resupplied with
rhotekin2*[ΔKRAP] (Fig. 3a-e; Supplementary Fig. 5a-e). Thus, rhotekin2's
mitochondrial functions critically required the syndapin I binding
interface.

Taken together, rhotekin2 gain-of-function and loss-of-function
phenotypes mirrored those of syndapin gain-of-function and loss-of-
function and furthermore rescue experiments revealed a requirement
of the syndapin binding site, i.e. the KRAP motif, for rhotekin2 func-
tions in shaping mitochondria.

In line with these functional results, rhotekin2 coprecipitated
endogenous syndapin I from brain lysates (Fig. 3g). Furthermore,
affinity-purified anti-rhotekin2 antibodies specifically coimmunopre-
cipitated syndapin I from rat brain lysates (Fig. 3h).

Together, these experiments demonstrated the in vivo relevance
of the identified complex formation of rhotekin2 with syndapin I in
the brain.

Mechanistically, the observed mitochondrial shape changes by
the mitochondrial anchor protein rhotekin2 and syndapin I may relate
to the ability of syndapin I to recruit fission components. While syn-
dapin I did not associate with Drp1[46], it coprecipitated not only dyna-
min1 but also dynamin2 (Supplementary Fig. 9), which has been
suggested to play a role in mitochondrial dynamics[47], albeit it is dis-
puted whether it is critical for this function[48].

### Rhotekin2 deficiency does not lead to increased apoptosis or to a reduced presence of the enlarged mitochondria in peripheral areas of the dendritic arbor of neurons

Rhotekin2 deficiency was correlated with apoptosis[42,43]. However,
labeling with annexin V, determination of caspase 3 activation and
visualizing mitochondrial depolarization via JC-1 dye excluded the
possibility that the observed mitochondrial size changes were related
to apoptosis of non-neuronal cells and primary hippocampal neurons,
respectively (Supplementary Fig. 10).

Another obvious hypothesis for the physiological relevance of the
rhotekin2-dependent mitochondrial changes was that rhotekin2 loss-
of-function-mediated mitochondrial size increases may impair the
efficient distribution of mitochondria inside of the dendritic arbor and
may lead to reduced mitochondrial presence in the often narrow
peripheral dendrite segments. Yet, when peripheral and proximal
densities of mitochondria were analyzed separately, the density of
mitochondria was not reduced in peripheral areas of rhotekin2-
deficient neurons compared to control neurons (Fig. 3i-k; Supple-
mentary Fig. 11a, b). Thus, there seemed to be no hypothesized
selection of mitochondria of different sizes for either peripheral or
proximal dendritic segments. In line, there also was no indication of
any spatially restricted occurrence of the rhotekin2 loss-of-function
phenotype, as the mitochondrial length changes upon rhotekin2
deficiency and the KRAP motif-dependence of this phenotype occur-
red in a similar manner in both peripheral and proximal areas of the
dendritic tree. Furthermore, the absolute mitochondrial length data
for proximal and peripheral areas were very similar (Fig. 3l, m; Sup-
plementary Fig. 11c, d).

### Mitochondria are recruited to central positions at nascent dendritic branch induction sites

While syndapin I shows basal levels during early development and its
levels rise postnatally reflecting an additional role of syndapin I in
synaptic rather than dendritic functions[33–35], the temporal expression
profile of rhotekin2 with the observed high expression until about
postnatal day 8 (P8) in the brain suggested that rhotekin2 functions in
some developmental process rather than in homeostasis or in synaptic
functions of mature neurons. Rhotekin2 expression also persisted in
the brains of 12 days, 4 weeks and 8 weeks old mice. The observed high

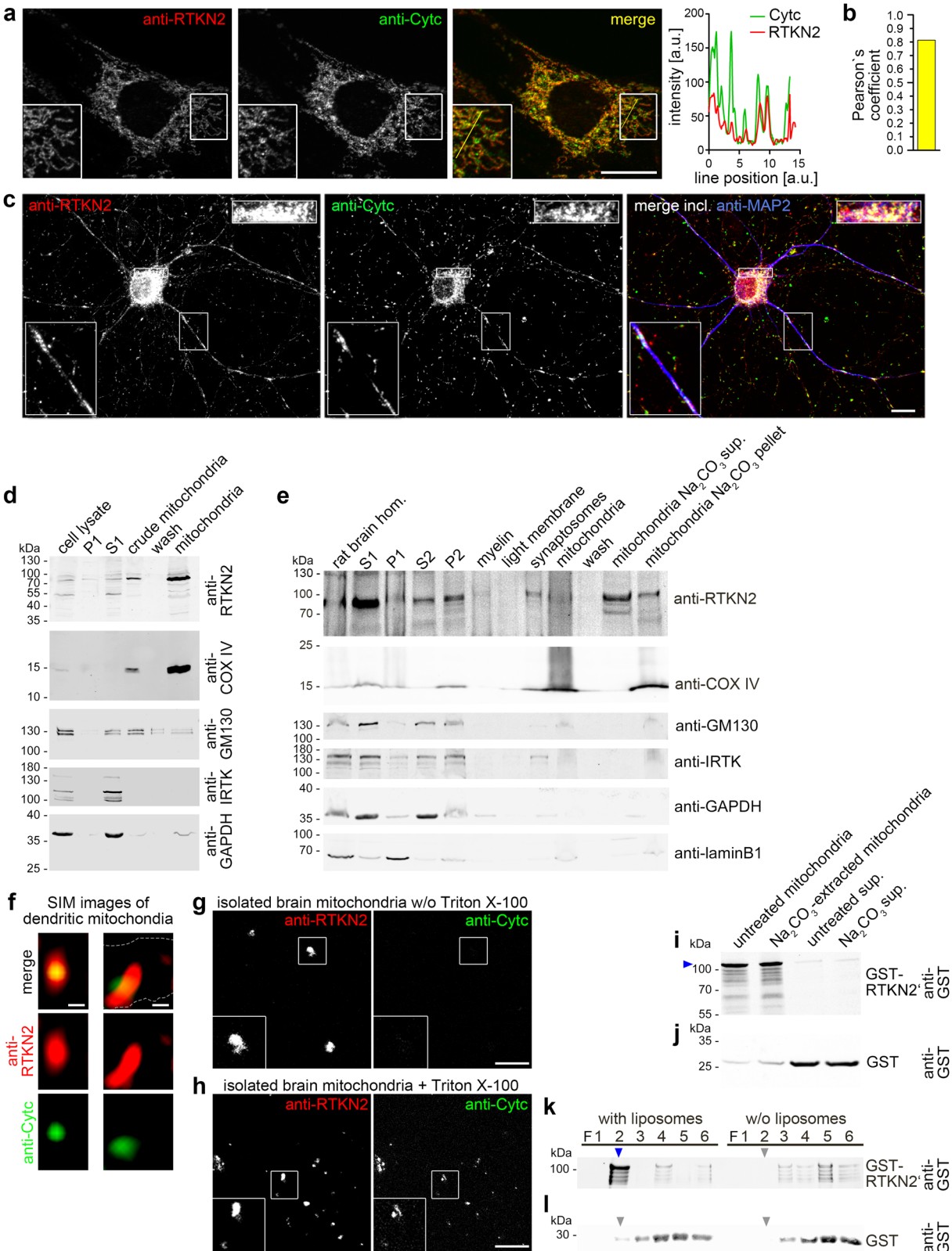

rhotekin2 expression until about postnatal day 8 (P8) and the following decline (Supplementary Fig. 12) correlated with the time window of dendritic arbor formation. We therefore next analyzed putative correlations between mitochondria distribution and dendritic branch induction by 3D-time-lapse imaging. Interestingly, quantitative analyses revealed that the frequency of mitochondria in dendritic

segments undergoing branch induction was about twice as high as in those not forming branches (Fig. 4a).

Furthermore, 3D-time-lapse imaging of neuronal morphogenesis revealed that mitochondria were not just occurring somewhere in the vicinity of nascent branch sites but were explicitly located close to the central axis of the newly forming dendritic branch (Fig. 4b;

**Fig. 2 | Rhotekin2 associates with the outer mitochondrial membrane. a** Sum intensity projections of ApoTome immunofluorescence images of NIH3T3 cells showing the localizations of endogenous rhotekin2 (RTKN2; red in merge) and cytochrome c (Cytc; mitochondrial marker; green in merge). The boxed area is shown at higher magnification as an inset in the respective image. The yellow line in the merge inset represents the line for showing an intensity profile histogram of fluorescence intensities of both channels, clearly demonstrating rhotekin2's spatial overlap with cytochrome c (yellow in merges). Insets, enlargement of boxed area. Bar, 20 μm. **b** Pearson's coefficient (calculated using the ImageJ plug-in JaCoP) of the overlap of the anti-RTKN2 and anti-cytochrome c signals (**a**). **c** Maximum intensity projections (MIPs) of anti-rhotekin2 (red in merge), anti-cytochrome c (Cytc; green in merge) and anti-MAP2 (neuronal marker for dendritic compartment; blue in merge) immunostained DIV10 hippocampal neurons showing that the localization of endogenous rhotekin2 overlaps with that of mitochondria in the cell soma and in dendrites. Insets, enlargement of boxed area. Bars, 10 μm. **d** Immunoblotting analyses of a fractionation experiment of NIH3T3 cells confirming that rhotekin2 is present and enriched at mitochondria (marker, inner mitochondrial membrane protein cytochrome c oxidase (COX IV) also showing enrichment in mitochondrial fractions). S, supernatant. P, pellet. Immunoblottings for further components reflecting other cellular compartments (insulin receptor tyrosine kinase (IRTK), plasma membrane; GM130, Golgi apparatus; GAPDH, cytosol) confirm the specific RTKN2 enrichment in the mitochondrial fractions. **e** Immunoblotting analyses of biochemical fractionations of brain homogenate obtained from adult rats towards mitochondria and of a $Na_2CO_3$ extraction of purified mitochondria. Markers of enriched mitochondria (COX IV) and markers as

above in **d** representing other cellular compartments and additionally laminB1 representing the nucleus. **f** MIPs of high-resolution structured illumination microscopy (SIM) images showing two examples of dendritic mitochondria of hippocampal neurons (DIV7). Note that the anti-RTKN2 (red) immunolabeling extends beyond the localization of the immunolabeling of the internal mitochondrial protein cytochrome c (Cytc). Dashed line in right example delineates the course of the dendrite (as tracked by CherryF coexpression). Bars, 100 nm. **g, h** MIPs of non-permeabilized (w/o Triton X-100), isolated mouse brain mitochondria immunostained with antibodies against rhotekin2 and cytochrome c (**g**), as well as of Triton X-100-permeabilized mitochondria (**h**). Note that rhotekin2 was detected in both preparations (i.e. is associated with the cytosolic leaflet of the outer mitochondrial membrane), whereas cytochrome c immunodetection required membrane permeabilization (internal protein). Insets, enlargement of boxed area. Bars, 10 μm. **i, j** Immunoblotting analyses of $Na_2CO_3$-extracted and untreated mitochondria that were incubated with GST-RTKN2' (**i**) and GST (**j**), respectively. Note that rhotekin2' (pointed out by blue arrowhead) is able to bind to $Na_2CO_3$-extracted mitochondria. **k, l** Anti-GST immunoblottings of density gradient fractionations showing a membrane-binding ability of rhotekin2'. Liposomes were incubated with either GST-RTKN2' (**k**) or GST (**l**). Note that GST-RTKN2' was detected in the liposome-containing fraction F2 (blue arrowhead), whereas GST-RTKN2' stayed at the bottom of the gradient (F4-6) in the absence of liposomes (**k**). Grey arrowheads mark the F2 density gradient fractions of control conditions devoid of accumulated, floating protein. All experiments shown as representative images (**a**, **c–l**) have been done at least twice with similar results.

Supplementary Fig. 13; Supplementary Movie 1). Despite the general mitochondrial dynamics, such mitochondria usually maintained this central position during the initial phase of dendritic branch formation (Fig. 4c; Supplementary Fig. 13).

Detailed quantitative analyses revealed that already 60 s prior to any visible induction of dendritic branching mitochondria showed a high occurrence probability in the central dendritic segment, which one minute later, at time zero, will give rise to a new dendritic branch. This mitochondrial positioning zone was very much spatially restricted. With dendritic analysis segments set as short as 1 μm, even at -60 s, a clear and highly significantly elevated probability of mitochondrial presence at the central area (500 nm to each side of the central axis of the forming protrusion) was determined. The determined probability was almost 0.6 instead of the theoretical value of 0.33 for an equal distribution between the 3 dendritic segments. The increased probability at the central area was accompanied with much lower probabilities of mitochondrial occurrence in neighboring 1 μm segments (proximal, 0.29; distal, 0.13) (Fig. 4d). At the time of the first morphologically detectable dendritic branch induction (t = 0 s), mitochondria showed an even higher probability of residing at such central positions of dendritic branch induction sites and reached a maximal probability of 0.77 (Fig. 4e). This very high probability of central positioning was still maintained 60 seconds after dendritic branch induction (t = +60 s; 0.75) (Fig. 4f). This argued for to our knowledge thus far unnoticed, very effective and extremely exact mitochondria positioning processes at dendritic branch induction sites.

### Mitochondrial positioning at nascent dendritic branch sites requires rhotekin2

Rhotekin2 could represent a key anchoring point at mitochondria allowing for proper mitochondrial positioning during dendritic branch induction. Indeed, while rhotekin2 deficiency did not cause a reduction of mitochondrial presence in the dendritic arbor in general (Fig. 3; Supplementary Fig. 5), rhotekin2 RNAi neurons showed a strongly reduced occurrence of mitochondria at branching dendritic sites upon rhotekin2 RNAi (0.8/μm compared to 1.5/μm in control neurons) (Fig. 5a; Supplementary Movie 2). The rhotekin2 loss-of-function phenotype was so severe that the data for branching dendritic segments in rhotekin2-deficient neurons (0.8/μm) were indistinguishable

from those of non-branching segments of control cells (also 0.8/μm; compare Fig. 4a). The mitochondrial accumulation at branch induction areas observed in control cells thus was completely disrupted upon rhotekin2-deficiency.

More extended time-resolved analyses demonstrated that one minute prior to dendritic branch induction (t = -60 s), there was absolutely no preference of mitochondrial localization to the central area in rhotekin2-deficient neurons. Instead, distal (lateral segment towards dendrite tip), central and proximal (lateral segment towards cell body) dendritic segments all showed positioning probabilities of about one-third (Fig. 5b). Also at the time of dendritic branch induction (t = 0 s), the mitochondrial positioning probability in the central dendritic segment still did not significantly rise in rhotekin2-deficient neurons (Fig. 5c). Only at t = +60 s, the probability finally rose to about 0.55 (Fig. 5d). Thus, at the time points prior to and right at dendritic branch induction proper mitochondrial accumulation completely depended on rhotekin2.

Direct comparisons (Fig. 5e-m) of the probabilities obtained for the three different zones and the three different time points in rhotekin2-deficient (Fig. 5b-d) and in control neurons (Fig. 4d-f) clearly highlighted the significantly reduced mitochondrial presence at central dendritic branch induction sites prior to and during dendritic branch induction (Fig. 5f, i).

### The mitochondrial mobility at dendritic branch sites is increased in rhotekin2-deficient neurons and this reflects a rhotekin2-dependent and syndapin I-supported mitochondrial tethering mechanism underlying mitochondrial positioning at dendritic branch sites

In line with the hypothesis that the observed recruitment and positioning of mitochondria at nascent dendritic branching points (Figs. 4 and 5) is brought about by a rhotekin2-dependent tethering mechanism, comparisons of mitochondrial mobility in branching versus non-branching dendritic areas revealed an attenuation of mitochondrial mobility in control cells at dendritic branch sites (−60 s to +60 s) (Fig. 6a, b). Normalized to data determined for neighboring non-branching areas of the same dendrite, the mobility of mitochondria at dendritic branch sites was ~20% lower than that in non-branching dendritic segments (Fig. 6b; *P* = 0.0201; absolute data without

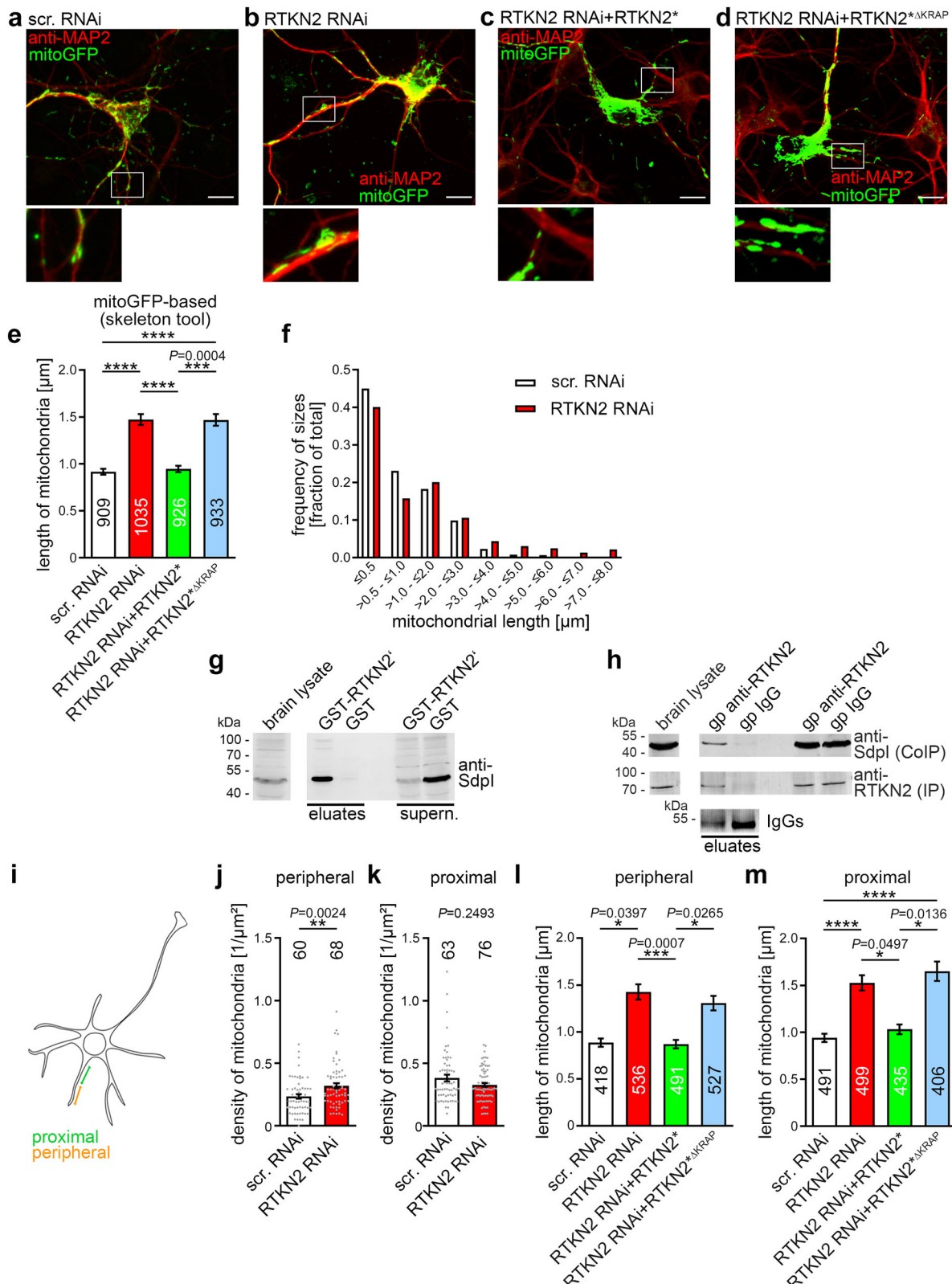

dendrite-intrinsic normalization, -30%). In control neurons, mitochondria thus showed clear mobility reductions at dendritic sites undergoing branching, as both the average maximum track speed and the spot acceleration were reduced (Fig. 6b, c). Similar observations were made in experiments, in which the temporal resolution was improved at the expense of 3D-information (Supplementary Fig. 14). The temporal resolution of both imaging conditions thus turned out to

be sufficient to detect motility changes of the mitochondria tethered at dendritic branch induction sites. Strikingly, such a significant decrease in mitochondrial mobility at dendritic branch induction sites when compared to non-branching areas was absent in rhotekin2-deficient neurons (Fig. 6e, f; Supplementary Fig. 14c, d).

These observations suggested that specifically the mobility of mitochondria at dendritic branch sites was increased in rhotekin2-

**Fig. 3 | Rhotekin2 shapes mitochondria in the dendritic arbor of hippocampal neurons and interacts with syndapin I in the brain. a–d** Sum intensity projections of anti-MAP2 immunostained primary hippocampal neurons (DIV6) that were transfected with scr. RNAi and IRES-mitoGFP (**a**) or RTKN2 RNAi and IRES-mitoGFP (**b**), RTKN2 RNAi and RTKN2*-IRES-mitoGFP (**c**) or RTKN2 RNAi and RTKN2*$^{\Delta KRAP}$-IRES-mitoGFP (**d**) at DIV4 (RTKN2*, wild-type, RNAi-insensitive rhotekin2; RTKN2*$^{\Delta KRAP}$, RNAi-insensitive rhotekin2 mutant lacking the syndapin binding site). Boxed areas are shown as magnifications below. Bars, 10 μm. **e** Determinations of the length of dendritic mitochondria employing mitoGFP and the skeleton software tool show a syndapin binding site-dependent function of rhotekin2 in mitochondria length control. **f** Length distributions for dendritic mitochondria of mitoGFP-transfected control neurons (scr. RNAi) and of rhotekin2-deficient neurons (RTKN2 RNAi). **g** Immunoblotting analyses of coprecipitation experiments showing that rhotekin2 interacts with endogenous syndapin I from mouse brain lysates. **h** Immunoblotting analyses of endogenous coimmunoprecipitation experiments. Guinea pig anti-rhotekin2 but not unrelated guinea pig antibodies (IgG) immunoprecipitated endogenous rhotekin2 and coimmunoprecipitated syndapin I from brain lysates. White lines/lanes indicate lanes of the blot omitted (**g**, **h**). **i** Scheme of a neuron highlighting the positions of the proximal and peripheral dendritic segments analyzed. (**j**, **k**) Analysis of mitochondrial distribution within peripheral (**j**)

and proximal (**k**) dendritic arbor segments of 20 μm in length each. **l**, **m** Quantitative analyses of mitochondrial length in peripheral (**l**) and in proximal (**m**) dendritic segments of neurons demonstrating that the rhotekin2 loss-of-function effects, the successful rescue by reexpressing RNAi-insensitive rhotekin2 (RTKN2*) and the failed rescue attempt with RTKN2*$^{\Delta KRAP}$ were observed irrespective of the position of the mitochondria in the dendrites. All experiments shown as representative images (**g**, **h**) have been done at least twice with similar results. Data, mean ± SEM visualized as bar (**e**, **l**, **m**) and bar/dot plots (**j**, **k**), respectively. **e**, **f** Scr. RNAi, $n = 909$; RTKN2 RNAi, $n = 1035$; RTKN2 RNAi+RTKN2*, $n = 926$ and RTKN2 RNAi+RTKN2*$^{\Delta KRAP}$, $n = 933$ dendritic mitochondria in 63, 56, 64 and 57 neurons, respectively, from 5 independent coverslips and 2 neuronal preparations; **j** Peripheral, scr. RNAi, $n = 60$; RTKN2 RNAi, $n = 68$ dendritic segments. **k** Proximal, scr. RNAi, $n = 63$; RTKN2 RNAi, $n = 76$ dendritic segments. **l** Scr. RNAi, $n = 418$; RTKN2 RNAi, $n = 536$; RTKN2 RNAi+RTKN2*, $n = 491$ and RTKN2 RNAi+RTKN2*$^{\Delta KRAP}$, $n = 527$ mitochondria. **m** Scr. RNAi, $n = 491$; RTKN2 RNAi, $n = 499$; RTKN2 RNAi +RTKN2*, $n = 435$ and RTKN2 RNAi+RTKN2*$^{\Delta KRAP}$, $n = 406$ mitochondria from 63, 56, 64 and 57 neurons, respectively, in 5 independent coverslips and 2 neuronal preparations. Statistical significances, Kruskal-Wallis/Dunn´s post-test (**e**, **l**, **m**) and Mann-Whitney (**j**, **k**), respectively. * $P < 0.05$, ** $P < 0.01$, *** $P < 0.001$, **** $P < 0.0001$. For exact $P$ values $\geq 0.0001$ see figure.

deficient developing neurons. Dendritic branch sites indeed showed a highly significant increase of maximum track speeds of mitochondria in rhotekin2-deficient neurons (Fig. 6g, 135% of control; Supplementary Fig. 14e, 189% of control). Spot acceleration determinations at sites of dendritic branching also showed a strong and statistically significant increase upon rhotekin2 deficiency (Fig. 6h, 164% of control; Supplementary Fig. 14f, 284% of control). Similar increases compared to control were observed when the displacement length at branching dendritic segments of rhotekin2-deficient neurons was determined (Fig. 6i, 143% of control; Supplementary Fig. 14g, 155% of control).

Rhotekin2 resides at the mitochondrial surface and could therefore reach out towards components accumulating at the plasma membrane of dendritic branch induction sites. Immunogold labeling of freeze-fractured plasma membranes of developing neurons had demonstrated that endogenous, plasma membrane-anchored syndapin I accumulated at dendritic branch induction sites[20]. We identified syndapin I as binding partner of rhotekin2. Syndapin I accumulations at dendritic branch induction sites could thus provide attachment points for rhotekin2-decorated mitochondria. SIM of developing neurons revealed that, despite the disadvantage that the anti-rhotekin2 antibody epitope overlaps with the syndapin I-binding area of rhotekin2, anti-rhotekin2 immunosignals at mitochondria were indeed also partially overlapping with anti-syndapin I immunosignals (Fig. 6j).

The hypothesized tethering of mitochondria via rhotekin2's interaction with syndapin would be substantiated by experimental evidence demonstrating that i) mitochondria can interact with syndapin I, that ii) such an interaction would indeed depend on mitochondrial surface proteins and that iii) explicitly rhotekin2 as mitochondrial surface protein can mediate such interaction. The explicitness of these questions demanded full biochemical reconstitutions. Digestion with trypsin provided mitochondria largely devoid of any endogenous rhotekin2 (Fig. 6k) at their surface. Also endogenous syndapin I was almost undetectable at their surface (for high-sensitivity scans see Supplementary Fig. 15).

Reconstitutions with these mitochondria and with purified syndapin I demonstrated that trypsin-treated mitochondria showed impaired syndapin I association, whereas untreated mitochondria still containing endogenous rhotekin2 clearly did associate with syndapin I (Fig. 6k). Importantly, addition of purified GST-rhotekin2' specifically and fully restored the ability of trypsin-treated mitochondria to interact with syndapin I (Fig. 6k). Thus, explicitly rhotekin2 as a mitochondrial surface protein is required for mitochondria to associate with syndapin I.

Further reconstitutions demonstrated that syndapin I-enriched surfaces are indeed able to specifically anchor mitochondria (Fig. 6l). Quantitative analyses confirmed that significantly more mitochondria are recruited to syndapin I-GFP-enriched surfaces than to those presenting only GFP (Fig. 6m). Parallel experiments with trypsin-treated mitochondria demonstrated that this recruitment of mitochondria to syndapin I-enriched surfaces was dependent on mitochondrial proteins, such as rhotekin2, which provides an anchor point at the mitochondrial surface (Fig. 6n, o).

**Syndapin I deficiency leads to reduced densities of mitochondria at nascent dendritic branch sites and to defects in mitochondrial positioning in the central area of dendritic branch induction sites and thereby mimics the loss-of-function phenotypes of rhotekin2**

Mechanistically, it seemed that syndapin I acts as a bridge between the plasma membrane, to which syndapin I can bind directly using its N-terminal F-BAR domain[32,34,49], and the mitochondrial anchor protein rhotekin2, to which syndapin I binds via its C-terminal SH3 domain. Provided that this molecular mechanism also plays a crucial role in vivo, not only rhotekin2 deficiency but also syndapin I deficiency should have detrimental effects on mitochondrial positioning at dendritic branch induction sites. Indeed, 3D-time-lapse analyses of developing neurons revealed that syndapin I RNAi resulted in a lack of mitochondria at nascent dendritic branch sites. With more than 50% decline, the effect was as severe as that of rhotekin2 RNAi and highly statistically significant (Fig. 7a; *** $P = 0.0001$).

More detailed spatio-temporal analyses revealed that – also similar to rhotekin2 loss-of-function (Fig. 5) – syndapin I deficiency specifically led to a severe and statistically highly significant defect in mitochondrial positioning. While control neurons showed very high probabilities of mitochondrial presence at central positions during, right before and right after dendritic branch initiation, syndapin I RNAi neurons failed to show a similar positioning (Fig. 7b-j; Supplementary Movie 3 and 4). In the central area of nascent dendritic branch induction sites, syndapin I RNAi neurons showed a severe and statistically highly significant reduction to only about 50% probability at $t = 0$ and $t = +60$ s (Fig. 7i, j). Before dendritic branch initiation ($t = -60$ s), mitochondrial localization probabilities between distal, central and proximal areas in syndapin I-deficient neurons even were comparable (Fig. 7e). Syndapin I-deficient neurons thus completely failed to show any increased mitochondrial localization probability in the central area prior to dendritic branch induction (Fig. 7h).

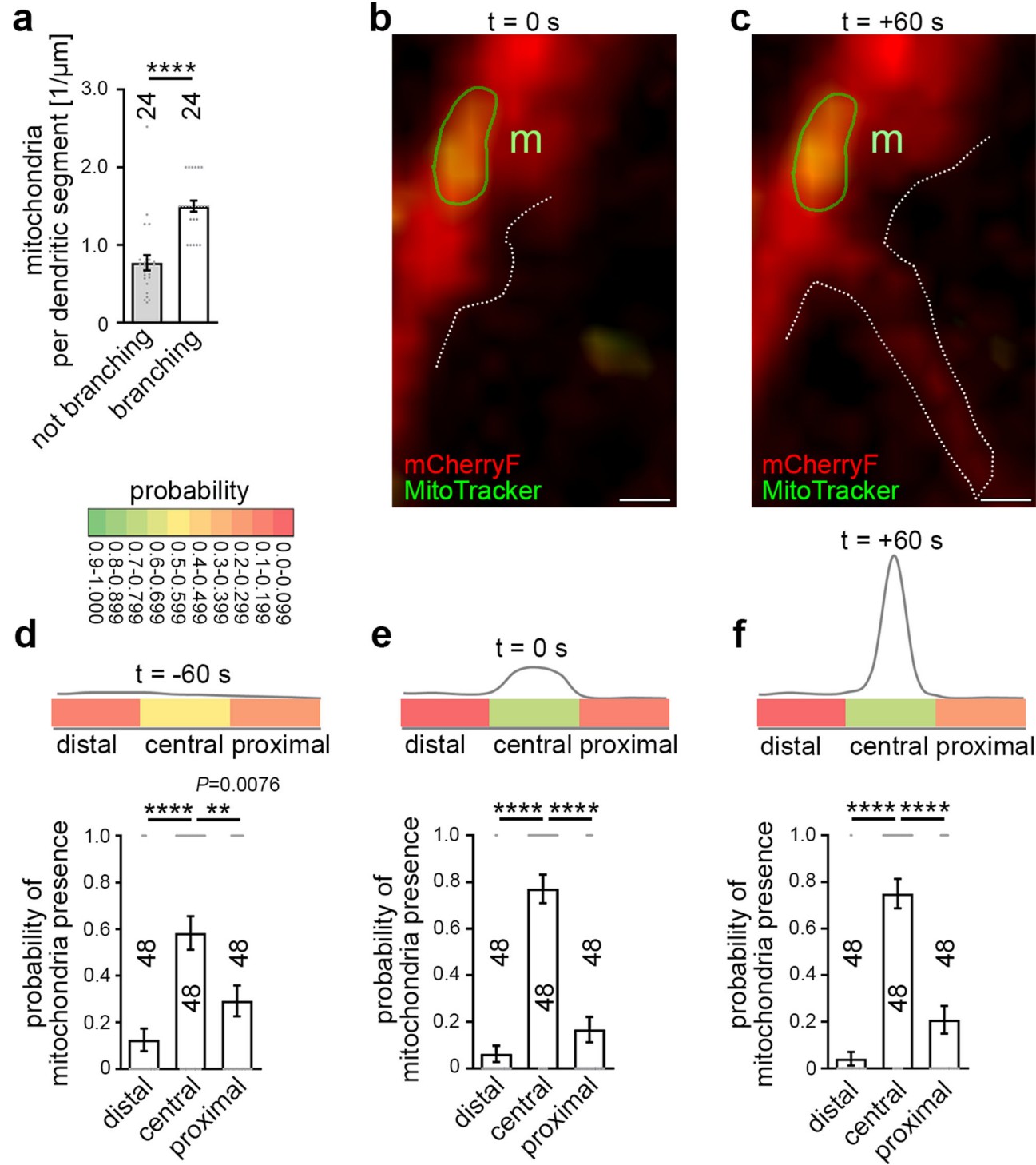

## The mitochondrial surface protein rhotekin2 and interactions via its KRAP motif are important for dendrite arbor formation in developing neurons

The data thus far revealed that developing neurons show a precise positioning and tethering of mitochondria at dendritic branch initiation sites and uncovered the molecular mechanisms underlying this cell biological process. An appealing hypothesis for a putative physiological relevance of the discovered process was that mitochondria specifically placed and held at dendritic branch induction sites may support dendritic branching during neuronal development by contributing to energy and/or $Ca^{2+}$ homeostasis. Yet, while the imaging conditions for resolving profiles of the very short-lived $Ca^{2+}$ transients

and the 3D-long-term imaging required for following dendritic morphology development are incompatible and do not allow for directly addressing such a mechanism, it was possible to address whether the identified rhotekin2 RNAi-induced defects in mitochondrial positioning and tethering would indeed be related to any detectable consequence for efficient dendritic arbor development. We thus carefully analyzed the morphologies of dendritic trees of rhotekin2-deficient neurons undergoing dendritic arbor formation (Fig. 8a-i; Supplementary Fig. 16 and 17).

In comparison to control cells, rhotekin2 RNAi led to a significant defect in overall dendritic tree growth (Fig. 8a, b, e; Supplementary Fig. 17a). Importantly, this could mostly be attributed to defects in

**Fig. 4 | Mitochondria occur more frequently at nascent dendritic branching sites in developing neurons and are aligned to a central position prior to branch induction. a** Mitochondrial occurrence (mitochondria per μm dendrite segment) in non-branching dendritic segments and in dendritic segments undergoing dendritic branch induction, as determined by quantitative analyses of 3D-time-lapse imaging (t = 0 s of branch induction; branching, 1 μm segments evaluated; frame rate, 1 full z-stack/10 s; all mitochondria included (i.e. irrespective of their speed)). (**b, c**) Merges of MIPs of t = 0 s frame (**b**) and t = +60 s frame (**c**) of 3D-time-lapse spinning disc microscopic recordings of neuronal morphology highlighted by plasma membrane visualization with farnesylated mCherry (transfection with mCherryF-reported scrambled RNAi plasmids at DIV4; outlined by dashed line at site of protrusion to enhance visibility) and mitochondrial position detected by MitoTracker (m; green circle) in DIV7 primary rat hippocampal neurons (see Supplementary Fig. 13 for images of individual fluorescence channels). Bars, 500 nm. **d–f** Quantitative analyses of mitochondrial positioning in temporal and spatial relation to newly developing dendritic branch sites in control DIV7 rat hippocampal neurons at three different time points, 60 s prior to branch induction (t = −60 s; **d**), during dendritic branch initiation (t = 0 s; **e**) and 60 s after the induction of a dendritic branch (t = +60 s; **f**; branch establishment and outgrowth). Dendritic segment length of the three zones (distal, lateral segment towards dendrite tip; central, directly at dendritic branching site; proximal, lateral segment towards cell body) 1 μm each. Color-coded visual presentation (upper panels) (for color coding see legend) and in form of quantitation graphs (lower panels). Data, mean ± SEM visualized as bar/dot plots. **a** n = 24 forming dendritic branching sites at 1 μm segments from 24 neurons evaluated at t = 0 s; n = 24 non-branching dendritic segments (cumulated length of non-branching segments, 138.47 μm) of the same neurons. **d–f** n = 48 independent dendritic branch sites from 24 neurons, respectively (each evaluated for the 3 time points). Statistical significances, Mann-Whitney (**a**) and Kruskal-Wallis/Dunn´s post-test (**d–f**). ** P < 0.01; **** P < 0.0001. For exact P value ≥ 0.0001 see figure.

dendritic branching, as the number of dendritic branch points and the number of dendritic terminal points were severely reduced upon rhotekin2 deficiency (Fig. 8f, g; Supplementary Fig. 17b, c).

Sholl analyses highlighted that such rhotekin2 deficiency-mediated defects specifically occurred in proximal and central areas of up to 40 μm distance from the soma center, i.e. in areas of DIV6 neurons that are marked by dendritic branches (Fig. 8h). Branch depth analyses revealed that the defects caused by rhotekin2 loss-of-function occurred both in the formation of primary dendrites but also were very pronounced (about -30%) at higher branch depth levels, i.e. reflected a lack of dendritic branch sites that was highly statistically significant for secondary and tertiary dendrites (Fig. 8i). The mitochondrial positioning and tethering protein rhotekin2 thus plays a critical role in the morphogenesis of developing neurons.

Reexpression of RNAi-insensitive rhotekin2 (RTKN2*) (Fig. 8c) verified the specificity of the rhotekin2 loss-of-function phenotypes in dendritic branching. The rescue attempt successfully suppressed the defects caused by rhotekin2 RNAi. Dendritic branch points, terminal points and dendritic tree length of neurons expressing rhotekin2 RNAi together with untagged RNAi-insensitive rhotekin2 were indistinguishable from those of control neurons (Fig. 8e-i; Supplementary Fig. 17).

Importantly, in contrast to the successful rescue by wild-type rhotekin2 (RTKN2*), reexpression of an RNAi-insensitive rhotekin2 mutant lacking the syndapin binding site (RTKN2*$^{\Delta KRAP}$) completely failed to rescue the observed morphogenesis defects caused by rhotekin2 deficiency. All dendritic parameters of such neurons remained statistically different from those of control neurons and were indistinguishable from those of rhotekin2 RNAi neurons despite the rescue attempt with the mutant (Fig. 8d-i; Supplementary Fig. 17).

The dendritic arborization impairments were specifically attributable to defective dendritic branch induction, as revealed by 3D-time-lapse imaging and quantitative evaluations thereof. Rhotekin2 RNAi neurons showed only about half as many dendritic protrusion starts per length of dendritic segment when compared to developing control neurons (Fig. 8j).

These results highlight the importance of specifically the syndapin I-binding KRAP-motif of rhotekin2 for the functions of this mitochondrial positioning and tethering protein during dendritic arbor formation of developing neurons.

## Discussion

Proper neuronal development, function and survival critically rely on the functions of mitochondria. Remarkable progress over the last years has provided substantial insights into the targeting and anchoring of mitochondria to presynapses and postsynapses[4–9]. In contrast, insights into how neurons are able to ensure a temporally and spatially controlled mitochondrial targeting and tethering during the development of their extended and complex dendritic morphologies are lacking. Our study uncovered that mitochondria are targeted to dendritic branch induction sites immediately before and right at the time of branch initiation and that this targeting involves a very effective and exact positioning and tethering of mitochondria at central areas of nascent dendritic branching sites.

To our knowledge, this mitochondrial positioning and tethering at sites of dendritic branch induction had thus far remained unrecognized – probably because this requires correlations of mitochondrial dynamics with neuronal morphology development difficult to achieve. The observed mitochondrial positioning and tethering at dendritic branch induction sites in developing hippocampal neurons is distinct from observations in retinal explants[22]. In this tissue, some mitochondria accumulations were reported at established, i.e. static, branch points of mature arbors[22] and hypothesized to relate to energy requirements of ER or Golgi outposts deposited at such established sites[50].

In developing neurons, the accumulation and specific positioning of mitochondria right at the central position of dendritic branch induction immediately before and at the time of dendritic branch initiation correlates with the transient local F-actin build-up at the base of forming dendritic protrusions, which seems to provide the required force for protrusion[14]. Actin polymerization processes, such as those triggered by the brain-enriched actin nucleator Cobl[15] and/or its relative Cobl-like[16], which both accumulated at branch induction sites and are critical for dendritic branch induction in developing neurons[14–16,20], can act as major driving force in the initial steps of dendritic branch induction and elongation. Actin dynamics is a highly ATP-consuming process. Furthermore, Cobl and Cobl-like are both controlled by transient Ca$^{2+}$ signals and CaM[14,16,20]. Mitochondrial positioning and tethering at dendritic branch induction sites would represent an elegant way to provide calcium and energy homeostasis for local actin filament formation specifically at the sites of action. Vice versa, it is plausible that the formation of local F-actin structures may be a factor in mitochondrial tethering and/or proper positioning at dendritic branch initiation sites.

We furthermore discovered molecular mechanisms capable of spatiotemporally connecting neuronal mitochondria with nascent dendritic branch sites. Our studies identified rhotekin2 as mitochondrial anchor component. In contrast to the only very distantly related and well-characterized rhotekin (rhotekin-1)[40,51,52], rhotekin2 did not associate with the Rho GTPases RhoA or Rac2[41]. Rhotekin2 was implicated in several types of cancers and suggested to somehow regulate proliferation, cell cycle and/or apoptosis[42–44,53,54]. Our microscopic analyses and biochemical fractionations clearly demonstrated that endogenous rhotekin2 localized to mitochondria both in neuronal and non-neuronal cells. Extraction and reconstitution experiments furthermore demonstrated that rhotekin2 is peripherally associated with the outer mitochondrial membrane and binding to mitochondria is

 

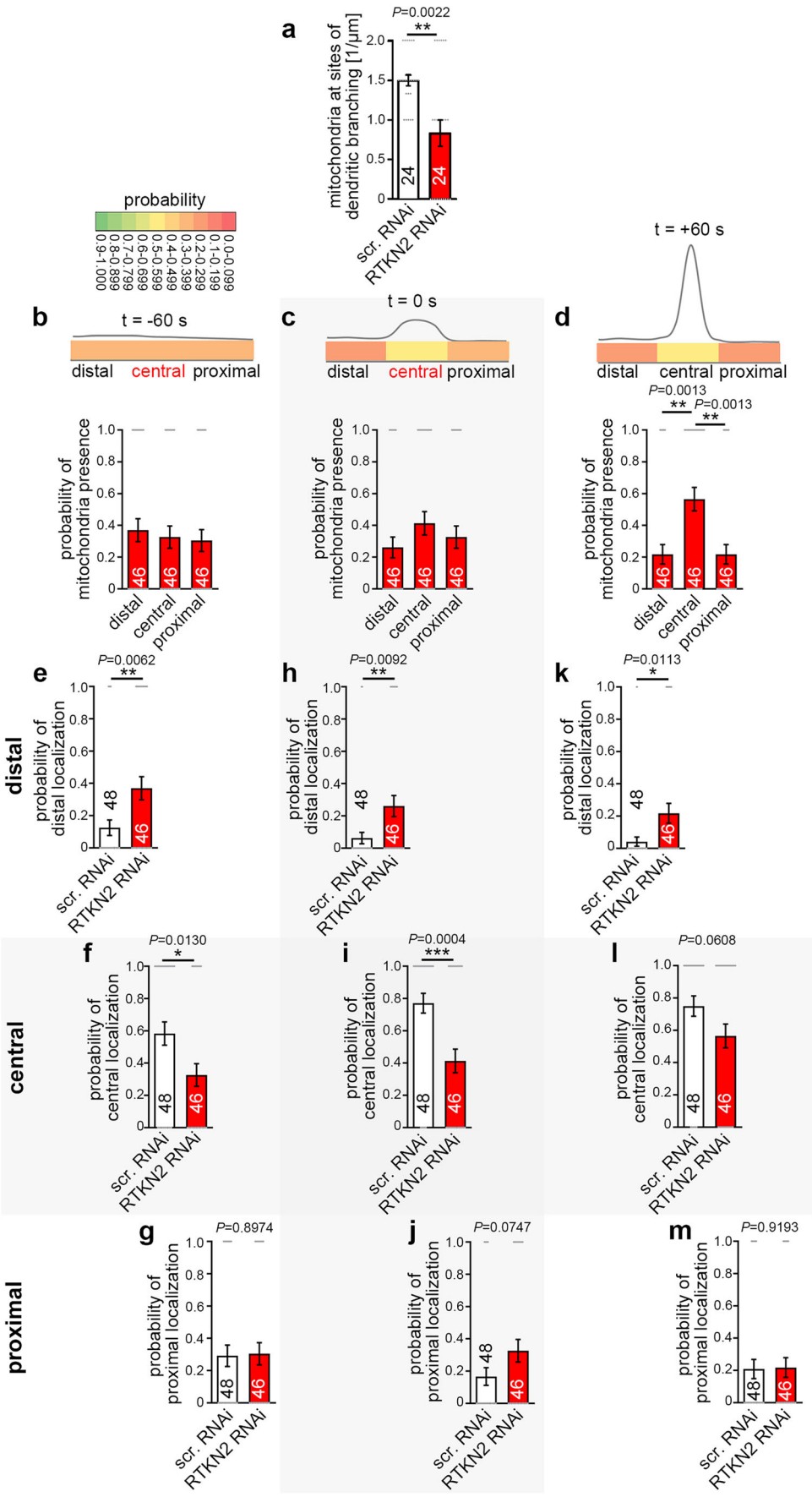

**Fig. 5 | Mitochondrial recruitment and central positioning to dendritic branch sites prior and during dendritic branch induction is impaired upon rhotekin2 deficiency. a** Quantitative analyses of mitochondrial occurrence (irrespective of mitochondrial speed) at forming dendritic branching sites in DIV7 primary rat hippocampal neurons transfected with either scrambled RNAi or RTKN2 RNAi at t = 0 s of dendritic branch induction (3D-live spinning disc microscopy) showing a strong reduction in the occurrence of mitochondria at sites of dendritic branch formation upon rhotekin2 deficiency. Morphology was visualized via mCherryF. Mitochondria were detected with MitoTracker. **b–d** Quantitative analyses of mitochondrial positioning in temporal and spatial relation to newly developing dendritic branch sites in rhotekin2-deficient DIV7 rat hippocampal neurons according to Fig. 4 at t = −60 s; (**b**), t = 0 s (**c**) and t = +60 s (**d**). Color-coded visual

presentation (upper panels) (for color coding see legend) and quantitation graphs (lower panels) show a severe disruption of mitochondrial recruitment and proper central positioning prior to and during dendritic branch induction. **e–m** Comparison of control (scr. RNAi) and RTKN2 RNAi data at t = −60 s (**e–g**); t = 0 s (**h–j**) and t = +60 s (**k–m**) in distal (**e, h, k**), central (**f, i, l**) and proximal (**g, j, m**) positions. Data, mean ± SEM visualized as bar/dot plots. (**a**) n = 24 forming dendritic branching sites from 24 neurons evaluated at t = 0 s; (**b–m**) RTKN2 RNAi, n = 46 dendritic branching sites from 24 neurons (each evaluated for the 3 time points); scr. RNAi data as in Fig. 4 (see legend of Fig. 4). Statistical significances, Mann-Whitney (**a, e–m**); Kruskal-Wallis/Dunn´s post-test (**b–d**). * P < 0.05; ** P < 0.01; *** P < 0.001. For exact P values see figure.

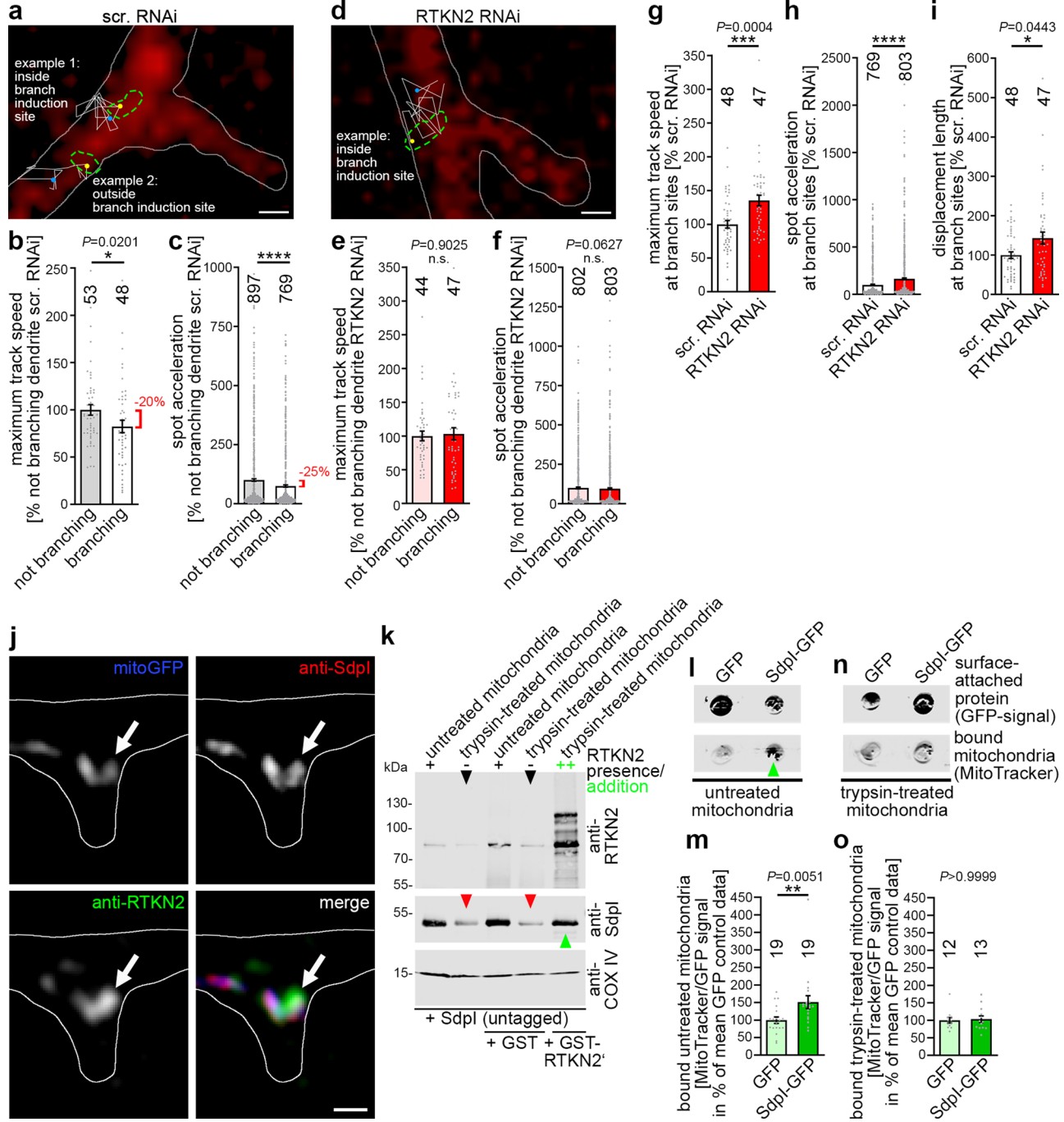

**Fig. 6 | Rhotekin2-dependent attenuation of mitochondrial mobility at dendritic branch sites and spatial overlap of mitochondria, rhotekin2 and synapin I near the plasma membrane of dendritic branch sites. a–i** Mitochondrial tracking in the vicinity of dendritic branch formation sites (**a**, **d**) and quantitative analyses of mitochondrial dynamics (**b**, **c**, **e–i**) (3D-time-lapse imaging using spinning disc microscopy). Shown are images of rat hippocampal neurons transfected at DIV4 with scrambled RNAi (**a**) and RTKN2 RNAi (**d**), respectively, overlayed with trajectories (colored line; observation time, 2 min) of spots representing the position and movement of MitoTracker-stained mitochondria (yellow dots, start; blue dots, end; green dashed lines, shape of mitochondria tracked). Representative mitochondrial examples with lengths ≥ the respective median in MitoTracker-stained mitochondrial length analyses (Supplementary Fig. 5f) (**a**, 0.51 μm and 0.59 μm; median, 0.51 μm; **d**, 0.79 μm; median, 0.72 μm) are shown (**a**, **d**). The morphology of the dendrites was outlined by coexpressed farnesylated mCherry (mCherryF, red; outlined with white line). Note that control neurons show slowed-down mitochondrial movement at branch sites (**b**) and reduction in spot acceleration (**c**), whereas rhotekin2-deficient neurons do not show reductions of mitochondrial movement at specifically dendritic branch initiation sites (**e**, **f**). Bars in **a** and **d**, 0.5 μm. **g–i** Comparisons of maximum track speed (**g**), spot acceleration (**h**) and displacement length (**i**) of mitochondria in control and rhotekin2-deficient neurons at branch sites. Measurements at branching sites (**b**, **c**, **e**, **f**) are presented as % of corresponding intra-assay data measured in non-branching dendrite areas. Results of phenotypic comparisons (**g–i**) are presented as % of intra-assay data for scrambled RNAi control. (**j**) High magnification SIM image (MIP; merged and individual channels) of a mitoGFP- and mCherryF-expressing developing neuron (DIV7) showing spatial overlap of mitoGFP (blue in merge), anti-rhotekin2 (green in

merge) and anti-syndapin I (red in merge) signals at a dendritic branch. An example of colocalization is marked by an arrow in each channel. The mCherryF-marked plasma membrane is shown as white line. Bar, 500 nm. **k** Immunoblotting analyses of syndapin I associations with purified untreated and trypsin-treated mitochondria. Note that trypsin treatment leading to reduced amounts of endogenous rhotekin2 (black arrowheads; also see Supplementary Fig. 15) also results in grossly diminished syndapin I interaction (indicated by red arrowheads) and that readdition of purified GST-RTKN2′ (but not of GST used as control) led to a restoration of the syndapin I binding to mitochondria (green arrowhead). (**l-o**) Representative fluorescence images of surface-attached GFP and syndapin I-GFP, respectively, incubated with either MitoTracker-stained untreated (**l**) or trypsin-treated mitochondria (**n**) and quantitative analyses of fluorescence signals of such reconstitutions (**m**, **o**). Green arrowhead marks the tethering of untreated (still endogenous rhotekin2-decorated) mitochondria to the syndapin I-enriched surface, whereas such a tethering to the syndapin I-enriched surface did not occur when trypsin-treated mitochondria were offered. All experiments shown as representative images (**j**, **k**) have been done at least twice with similar results. Data, mean ± SEM visualized as bar/dot plots. (**b**, **e**, **g**, **i**) n = 53 (scr. RNAi not branching), n = 48 (scr. RNAi branching), n = 44 (RTKN2 RNAi not branching) and n = 47 (RTKN2 RNAi branching) trajectories. (**c**, **f**, **h**) n = 897 (scr. RNAi not branching), n = 769 (scr. RNAi branching), n = 802 (RTKN2 RNAi not branching) and n = 803 (RTKN2 RNAi branching) frame-to-frame accelerations in 24 (scr. RNAi) and 24 (RTKN2 RNAi) hippocampal neurons from 4 independent neuronal preparations. (**m**) Untreated mitochondria n = 19 each, (**o**) trypsin-treated mitochondria n = 12 (GFP) and 13 (SdpI-GFP). Statistical significances, Mann-Whitney. * P < 0.05; ** P < 0.01; *** P < 0.001; **** P < 0.0001. For exact P values ≥ 0.0001 see figure.

independent of any further membrane-associated mitochondria proteins.

Several phenotypic observations support the mitochondrial functions of rhotekin2. Whereas overexpression of rhotekin2 resulted in shorter mitochondria, knockdown of rhotekin2 led to significantly enlarged mitochondria. This loss-of-function phenotype was specifically attributable to rhotekin2, as demonstrated by rescue experiments. The identified rhotekin2 loss-of-function phenotype hereby occurred irrespective of peripheral and proximal areas of the dendritic tree, of neuronal compartment and even of cell-type.

Strikingly, this rhotekin2 role in mitochondrial size control relied on a newly identified binding motif following the consensus KKRrAPlPP (KRAP motif), which is highly evolutionarily conserved in different vertebrate rhotekin2 proteins and turned out to be required for interaction with syndapins. This protein-protein interaction identified by Y2H screening was underscored by coprecipitation and coimmunoprecipitation studies as well as by experiments with purified components demonstrating a direct interaction. Coprecipitation assays and coimmunoprecipitation studies of the endogenous proteins from brain lysates provided in vivo evidence for the relevance of the identified rhotekin2/syndapin I complex formation in the brain.

The mitochondrial shape changes observed clearly involved rhotekin2, syndapins and the syndapin interaction site of rhotekin2. Mechanistically, we provided clear evidence that rhotekin2 is used as mitochondrial anchor point for syndapins. Syndapin I has been demonstrated to interact with dynamin2[33,55] and to be able to act as a dimerized linker that is able to join different SH3 domain interaction partners[18,20,56,57]. Such a mechanism would provide a possibility for a recruitment of dynamin2, which may have an - albeit disputed - role in mitochondrial fission[47,48].

Our 3D-time-lapse imaging studies revealed that mitochondria normally occur with a much higher frequency at branch induction sites. Importantly, mitochondria were much less frequent at these developmentally important sites when rhotekin2 was lacking. Rhotekin2 thus is a crucial component in the identified mitochondrial positioning spatiotemporally correlated with dendritic branch induction.

The positioning process hereby turned out to be astonishingly precise. Importantly, the mitochondrial positioning to the central

branch area prior to and during dendritic branch initiation was absent in rhotekin2-deficient neurons. Rhotekin2-deficient neurons thus lacked the striking spatial preference of mitochondria for branch induction sites observed in control neurons.

Our analyses of mitochondrial mobility revealed a mitochondrial tethering process prior and during dendritic branch induction. The higher track speeds and displacement lengths determined at branch sites in rhotekin2-deficient neurons clearly showed that rhotekin2, which would have the ability to reach out to especially syndapin I-enriched plasma membrane areas undergoing the required membrane topology changes for dendritic branch induction[14,20], is critical for this process. In line, biochemical experiments demonstrated that syndapin I is able to interact with mitochondria in a manner that is not reflecting some interaction with the mitochondrial membrane but instead is mediated by the identified SH3 domain interaction with the outer mitochondrial membrane-associated protein rhotekin2, whereas the membrane interaction of syndapin I is mediated by its F-BAR domain[32,49]. Furthermore, high-resolution imaging showed some overlap of mitochondria, rhotekin2 and syndapin I at plasma membrane areas undergoing dendritic branch induction. Finally, in line with the importance of syndapin I as a bridge between the mitochondrial anchor point rhotekin2 on one side and the plasma membrane on the other side, functional analyses in developing neurons revealed that syndapin I deficiency led to similar defects in mitochondrial positioning at dendritic branch induction sites as those observed for its binding partner rhotekin2.

The rhotekin2-mediated mitochondrial tethering, concentration and proper positioning at nascent and forming dendritic branch induction sites may ensure that crucial mitochondria-dependent functions, such as energy provision and calcium homeostasis, are met in a locally and spatially highly defined manner. The high expression levels of rhotekin2 during brain development correlated with neuronal differentiation and neuromorphogenesis. Due to their extended and complex shapes, neurons face the challenge of properly distributing mitochondria. In line, rhotekin2-deficiency had striking detrimental consequences for dendritic arbor development. The function of rhotekin2 in regulating neuronal morphology hereby critically relied on the syndapin I-binding

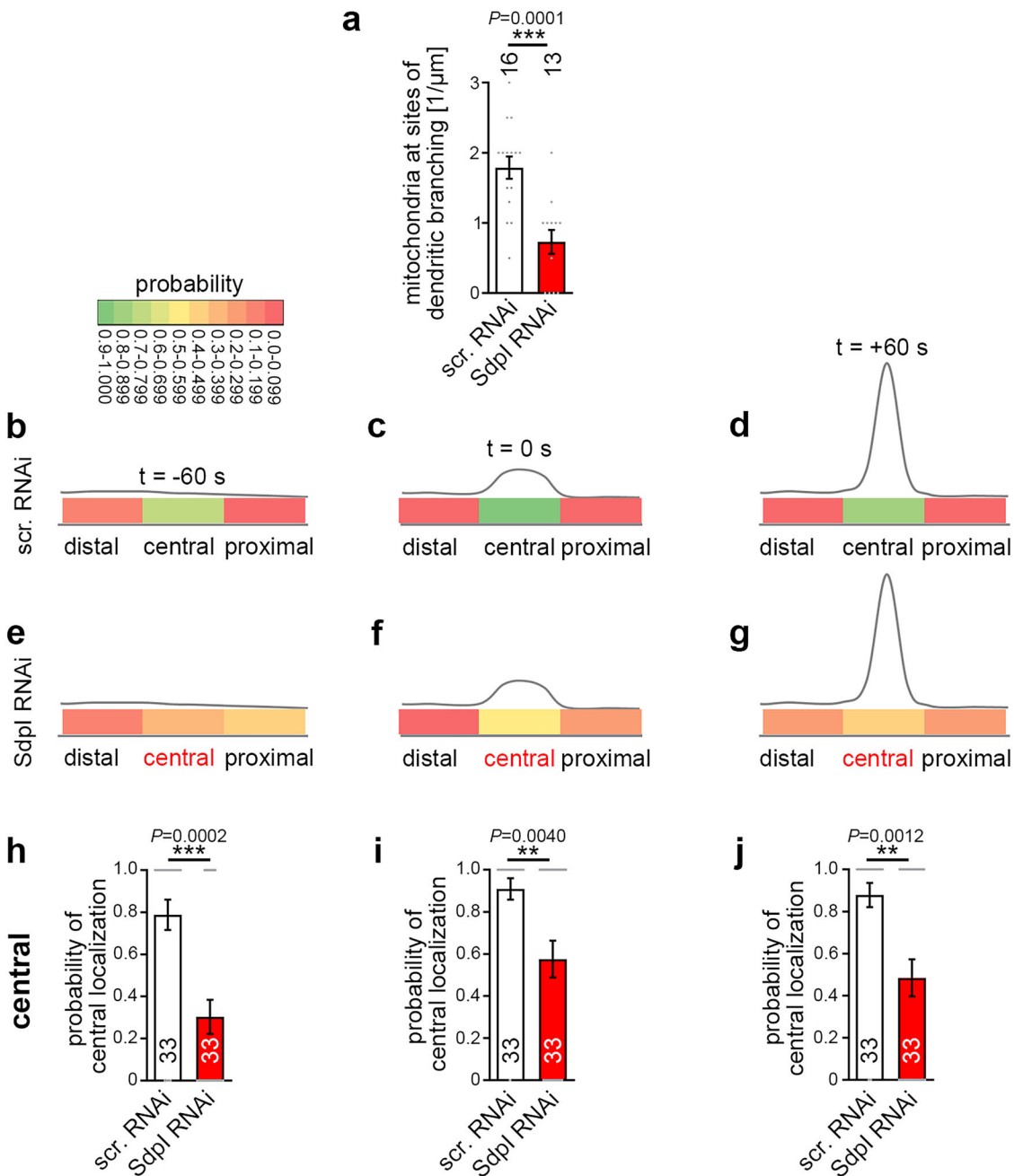

**Fig. 7 | Syndapin I deficiency leads to reduced densities of mitochondria at nascent dendritic branch sites and to defects in mitochondrial positioning in the central area of dendritic branch initiation sites. a** Determinations of mitochondrial densities at nascent dendritic branch sites at t = 0 s in rat hippocampal neurons transfected (at DIV4) with scrambled RNAi and syndapin I RNAi (SdpI RNAi), respectively. Morphology was visualized via mCherryF and mitochondria were detected with MitoTracker using 3D-live spinning disc microscopy. Note that syndapin I RNAi led to a strong reduction in the occurrence of mitochondria at sites of dendritic branch induction. **b–g** Color-coded visual presentations of quantitative analyses of mitochondrial positioning in temporal and spatial relation to newly

developing dendritic branch sites in control (**b–d**) and syndapin I-deficient (**e–g**) DIV7 rat hippocampal neurons at t = −60 s (**b, e**), t = 0 s (**c, f**) and t = +60 s (**d, g**). **h–j** Control (scr. RNAi) versus syndapin I RNAi effects on mitochondrial positioning in central areas of nascent dendritic branch sites at t = −60 s (**h**); t = 0 s (**i**) and t = +60 s (**j**). Data, mean ± SEM visualized as bar/dot plots. (**a**) n = 16 (scr. RNAi) and n = 13 (SdpI RNAi) neurons, respectively (evaluated at t = 0 s). **b–j** Scr. RNAi and SdpI RNAi, n = 33 independent dendritic branch sites from 16 and 13 neurons, respectively (each evaluated for the 3 time points). Statistical significances, unpaired Student's t-test (**a**) and Mann-Whitney (**h–j**). ** P < 0.01; *** P < 0.001. For exact P values see figure.

---

KRAP motif and clearly was a defect in the initiation of dendritic branch formation, as revealed by quantitative 3D-time-lapse analyses. Thus, rhotekin2-mediated functions are crucial for proper control of dendritic arborization (Fig. 9).

The identified crucial role of rhotekin2 in neuronal network formation may also relate to observations that disturbances of mitochondrial dynamics are causative in many neurological

disorders in humans[4,10,11]. Rhotekin2 shows a prominent expression during the developmental time window of dendritic arbor formation but also is readily detectable in adolescent and adult mice. Interestingly, also syndapin I loss-of-function led to neurological impairments[35]. Furthermore, the observed deficits of a schizophrenia-associated syndapin I mutant[58] as well as *syndapin I* KO included impaired recruitment of effectors as well as dendritic

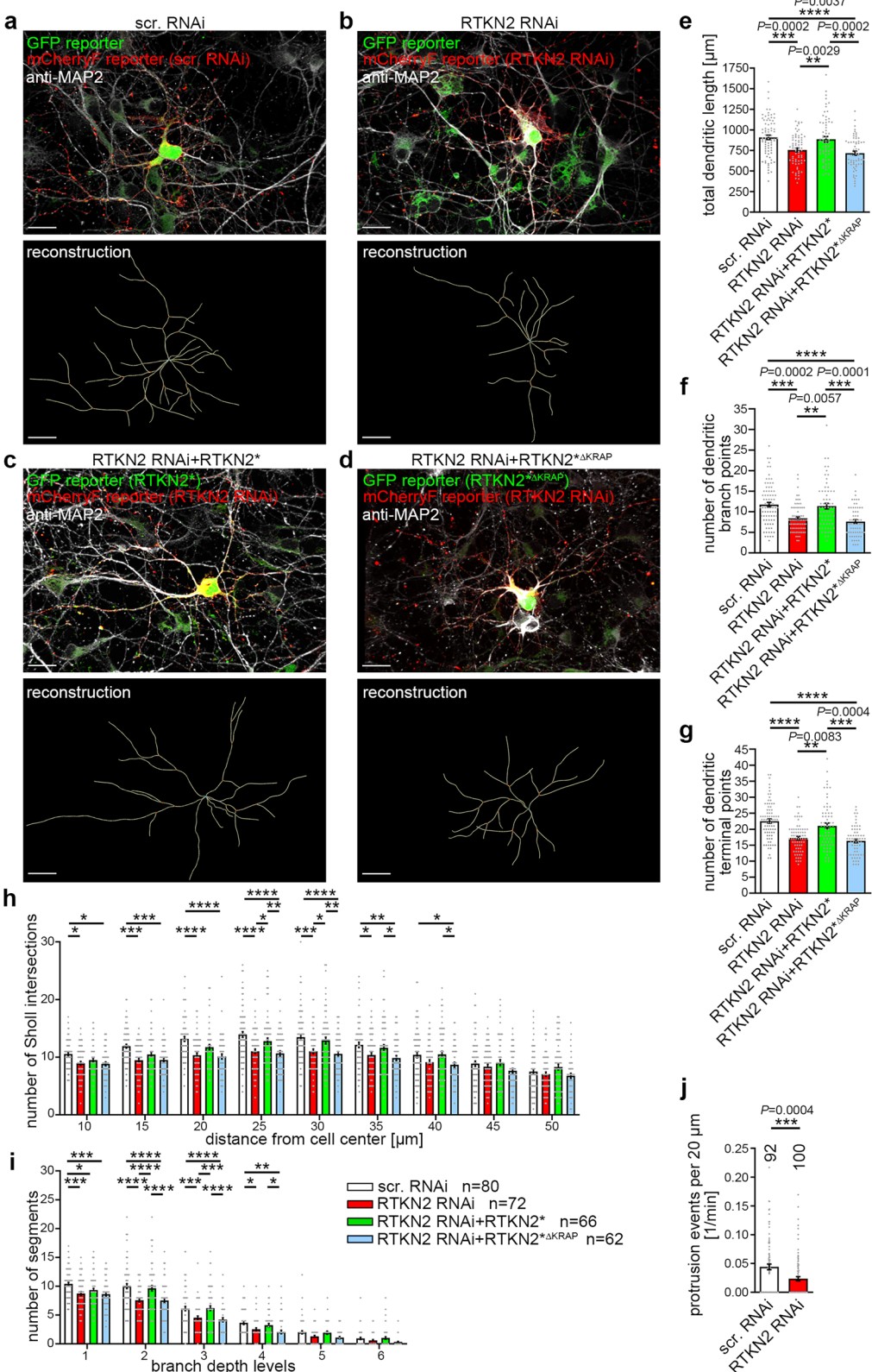

deficits[35]. Also the finding that a variety of schizophrenia risk genes are implicated in mitochondrial dynamics[59] points to an intimate functional relationship of mitochondria and neuronal network formation and function, which, at nascent dendritic branch sites, is represented by the spatiotemporally very precise tethering of mitochondria and the critical role of rhotekin2 therein discovered in this study.

## Methods

### Animals

Animal breeding was in strict compliance with the EU directives 86/609/EWG and 2007/526/EG guidelines for animal experiments and all related procedures were approved by the local government (Thüringer Landesamt, Bad Langensalza, Germany, and Landesverwaltungsamt, breeding allowance UKJ-17-021). Primary

**Fig. 8 | The crucial mitochondria-positioning protein rhotekin2 is important for dendrite arbor formation in developing neurons. a–d** Analyses of dendritic rhotekin2 loss-of-function phenotypes and rescue attempts in primary hippocampal neurons transfected at DIV4, fixed and immunostained for MAP2 48 h thereafter and subjected to anti-MAP2-based analyses of neuronal morphology by 3D-surface reconstruction and dendrite tracking using IMARIS software (lower panels, reconstruction). Neurons transfected with scrambled RNAi control vector (**a**) and RTKN2 RNAi vectors (**b–d**) (both mCherryF-reported) in absence (**b**) or in presence of full-length RNAi insensitive rhotekin (RTKN2*) (**c**) or a rhotekin2 mutant lacking the syndapin I-binding KRAP motif (RTKN2*$^{\Delta KRAP}$) (**d**), marked by GFP reporter coexpression. **e–i** Quantitative determinations of specific defects in dendritic arborization caused by rhotekin2 loss-of-function addressing total dendritic length (**e**), number of dendritic branch points (**f**), number of dendritic

terminal points (**g**), number of Sholl intersections (**h**) and number of segments per branch depth levels (**i**). **j** Determinations of dendritic branch protrusion starts (per 20 μm dendrite segments and per min) from 3D-time-lapse imaging data revealing a highly significant disruption of dendritic branch induction upon rhotekin2 deficiency. Bars, 10 μm. (**e–i**) $n = 80$ (scr. RNAi), $n = 72$ (RTKN2 RNAi), $n = 66$ (RTKN2 RNAi + RTKN2*), $n = 62$ (RTKN2 RNAi+RTKN2*$^{\Delta KRAP}$) neurons for each condition from 5 independent neuronal preparations. (**j**) $n = 92$ (scr. RNAi) and $n = 100$ (RTKN2 RNAi) dendritic segments from 24 neurons each and 2 independent neuronal preparations. Data are mean ± SEM visualized as bar/dot plots. Statistical significances, one-way ANOVA/Sidák's (**e**), Kruskal-Wallis/Dunn's (**f, g**); two-way ANOVA/Bonferroni post-test (**h, i**) and Mann-Whitney (**j**). * $P < 0.05$; ** $P < 0.01$; *** $P < 0.001$; **** $P < 0.0001$. For exact $P$ values ≥ 0.0001 in **e–g** and **j** see figure.

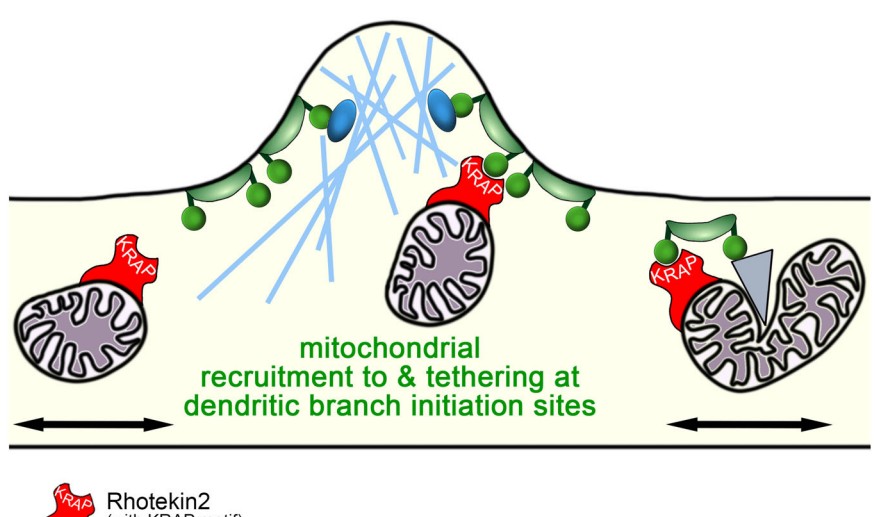

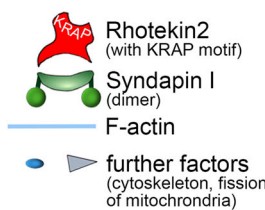

Rhotekin2
(with KRAP motif)

Syndapin I
(dimer)

F-actin

further factors
(cytoskeleton, fission
of mitochrondria)

**Fig. 9 | Model depicting the functions of rhotekin2 and of its binding partner syndapin I in mitochondrial dynamics as well as in proper mitochondrial positioning and tethering accompanying dendritic branch initiation in developing neurons.** The model shows molecular mechanisms underlying the rhotekin2 and syndapin I phenotypes identified in both mitochondrial size changes and in proper mitochondrial positioning and tethering accompanying dendritic branch initiation in developing neurons. Drawn is a dendritic segment of a developing neuron undergoing branch induction. Rhotekin2 serves as important

mitochondrial anchor point for association with syndapin I – a protein that shapes membranes and accumulates at the plasma membrane of dendritic branch induction sites. The syndapin I molecules depicted represent dimers formed by self-association of syndapin I's F-BAR domain leaving the SH3 domains free for interactions, such as with the KRAP motif in rhotekin2 (but presumably also with further syndapin binding partners, such as the cytoskeletal components Cobl and Cobl-like or dynamin2, which may act as a pinchase in mitochondrial dynamics).

culture generation was approved under the allowance UKJ-24-005.

The generation and analysis of *syndapin I* KO mice has been described previously[33,35].

### Yeast-two-hybrid screening

Y2H screenings were performed as described previously[39]. Full-length syndapin I was used as bait to screen a mouse brain library (BD Biosciences Clontech).

Identified prey plasmids were amplified in *E. coli* and after sequence analyses retransformed into yeast together with different bait plasmids encoding the BD domain and fusions thereof (pGBTK7; syndapin I and syndapin I$^{\Delta SH3}$ cloned into pGBTK7) to again analyze the reporter gene activities (growth on −AHLW drop-out plates and β-Gal activity) as described[39,60].

### DNA constructs

TrxHis-RTKN2' was generated by PCR using the RTKN2 isoform2 (RTKN2') sequence isolated by Y2H screening as template and subcloning into pET32 (Novagen). The BALB C mouse sequence of RTKN2 isoform 2 (EDL_32016.1) was used as reference sequences for validation. Note that the BL6 version of rhotekin2 isoform 2 (NP_001346248.1) has 7 amino acid exchanges when compared to the BALB C sequence (Supplementary Fig. 3b)

GST-RTKN2' and GST-RTKN2'$^{457-601}$ were generated by subcloning into pGEX-5X-1 (GE Healthcare).

GFP-RTKN2' full-length and GFP-RTKN2'$^{457-601}$ were obtained by subcloning into pEGFP-C3. Flag-RTKN2' was generated by subcloning into pCMV-Tag2b.

Additionally, plasmids encoding for RTKN2 from BL6 mice were cloned. pIRES-RTKN2 + GFP (RTKN2 isoform 1) and a ΔKRAP mutant

thereof (Δaa492-503 according to RTKN2 reference sequence NP_001346247.1) were generated by PCRs using mouse brain cDNA (BL6) as template, which was generated as described previously[61]. The PCR products were inserted into pIRES (BD Bioscience/Clontech) using the *XhoI* and *SalI* restriction sites. A GFP-encoding sequence was inserted 3′ of the IRES site and used as visual expression reporter for the untagged rhotekin2 and ΔKRAP mutant thereof coexpressed by the construct.

RNAi constructs directed against mouse and rat RTKN2 were generated by primer annealing and insertion into pRNAT-H1.1 according to procedures described previously[15]. Phosphorylated primers for RTKN2 RNAi (bp 317-334 of mouse RTKN2 ORF NM_001081346.1; 5′-GATCCGGAAAGACTCCGATCACTTTTCAAGAGAA AGTGATCGGAGTCTTTCCTTTTT-3′ and 5′-AGCTTAAAAAGGAAAGA CTCCGATCACTTTCTCTTGAAAAGTGATCGGAGTCTTTCC-3′) were annealed and the products were subcloned into pRNAT H1.1-GFP.

A pRNAT vector expressing scrambled RNAi sequence served as control as described previously[34]. The GFP reporter of the pRNAT plasmids was substituted by a sequence encoding for (farnesylated) mCherry (mCherryF) to visualize the plasma membrane of the cells[34]. BFP-reported RTKN2 RNAi plasmids were generated by replacing the GFP of pRNAT by BFP.

For RNAi validation, derivatives of the RNAi vectors were used, in which the GFP reporter was replaced by GFP-RTKN2 by subcloning.

An RNAi-insensitive rhotekin2 (RTKN2*) was generated by introducing several silent mutations into the RNAi site to render the mRNA resistant to RTKN2 RNAi. Primers used were 5′- AAGGA-TAGCGACCATTTCAGTAATAAAGAATGCACACAGC-3′ as forward and 5′- ATGGTCGCTATCCTTCCACATTAATGGTAT-3′ as reverse primer. RNAi-insensitive RTKN2 full-length as well as mutants thereof lacking the syndapin I binding-site (RTKN2*ΔKRAP) were inserted into the pIRES-GFP vector to express untagged rhotekin2 (together with GFP as reporter) and were used for coexpression with the RNAi constructs.

For alternative experiments tracking mitochondria by mitoGFP, the GFP reporter in pIRES-GFP was replaced by mitoGFP – a GFP fusion containing the mitochondrial targeting sequence of human cytochrome c oxidase, as in pEYFP-Mito (Clontech).

BD-syndapin I and BD-syndapin I^ΔSH3 were as described[39,60].

Plasmids encoding for GST-syndapin I, GST-syndapin II-s, GST-syndapin II-l and GST-syndapin III as well as GFP-syndapin I, GFP-syndapin II-s and GFP-syndapin II-l have been described previously[17,37,39,57]. Additionally, syndapin I was inserted into pGEX-6P (GE Healthcare) to generate a GST-syndapin I cleavable by Prescision protease (GE Healthcare)[20] and a GST-syndapin I-GFP was generated by subcloning syndapin I into a GFP-containing derivative of pGEX-6P[62].

Syndapin II siRNA and a scrambled control thereof were as described[36].

A GFP-tagged, RNAi-insensitive syndapin II-l (GFP-syndapin II-l*) has been generated by PCR using the forward primer 5′-AGCAGCAT-CAGTGAGAAGAAGATATCAAAGCTAAG-3′ and the reverse primer 5′-ATAGCTGCTGACATTCTTAGCTTTGATATCTTCTTT-3′ to induce several silent mutations into the RNAi targeting site for the above-described syndapin II-l RNAi.

Plasmids coding for myc-tagged dynamin1 and dynamin2 were described previously[55]. A plasmid encoding Xpress-tag dynamin-related protein 1 (Xpress-Drp1) was were generated via PCR from rat brain cDNA using the forward primer 5′-ATAGGATCCATGGAGGCGCTG-3′ and the reverse primer 5′-CGCGCTAGCTCAATTGCCACTAA-3′ and cloned into pcDNA3.1HisC (Invitrogen).

Correct cloning by PCR was verified by sequencing in all cases.

## Antibodies, reagents and proteins

Guinea pig anti-RTKN2 antibodies were raised and purified against GST-RTKN2′457-601. Terminal bleeds were collected and tested for immunoreactivity. Affinity-purifications of anti-rhotekin2 guinea pig sera were done on immobilized TrxHis-RTKN2′457-601 according to procedures described previously[17]. Polyclonal rabbit and guinea pig antibodies against GST-syndapin I as well as anti-GST and anti-TrxHis antibodies were purified from antisera as described previously[20].

Monoclonal mouse anti-GFP antibodies (JL-8) were from Clontech (1:8000; 632381, Takara). Monoclonal anti-Xpress antibodies (1:1000; R910-25) and anti-myc (9E10, 1:1000; MMS-150R) antibodies were from Invitrogen and Covance, respectively. Monoclonal mouse (M2; 1:1000; F3165) and polyclonal rabbit anti-Flag (1:1000; F7425) as well as monoclonal mouse anti-MAP2 (HM-2; 1:750; M4403) antibodies and monoclonal mouse anti-β-actin antibodies (1:1000; A5441) were from Sigma. Polyclonal goat anti-GAPDH (1:2000; sc-48167), polyclonal rabbit anti-insulin receptor α (IRTK, 1:1000, sc-710) as well as non-immune murine IgGs and guinea pig IgGs used as controls in coimmunoprecipitions and in anti-RTKN2 antibody validation studies were purchased from Santa Cruz Biotechnology (sc-2025 and sc-2711). Polyclonal rabbit anti-MAP2 (1:500; ab32454), anti-laminB1 (1:5000; ab16048), anti-GFP (1:1000; ab6556) and anti-mCherry antibodies (1:2000; ab167453) as well as monoclonal mouse anti-COX IV (1:1000; ab 14744) and anti-cytochrome c antibodies (1:1000; ab3255) were acquired from Abcam. Anti-cleaved caspase 3 antibodies were purchased from Cell Signaling (1:1000; 9661). Monoclonal mouse anti-GM130 antibodies (1:1000; 610823) were from BD Transduction and polyclonal rabbit anti-GAPDH antibodies (1:8000, 10494-1-AP) were from Proteintech.

MitoTracker DeepRed (1:10000; M22426) and AlexaFluor647-conjugated phalloidin (1:250; A22287) were purchased from Molecular Probes.

Secondary antibodies used included Alexa Fluor488- and 568-labeled goat anti-guinea pig antibodies, Alexa Fluor488- and 568-labeled donkey anti-mouse as well as Alexa Fluor647-labeled goat anti-mouse antibodies, Alexa Fluor568-labeled donkey anti-rabbit, Alexa Fluor350-, 647-labeled goat anti-rabbit antibodies (1:600; A-11073, A-11075, A-21202, A-21235, A-21236, A-10042, A-21068, A-21245¸ Molecular Probes), AlexaFluor680-labeled goat anti-rabbit and anti-mouse antibodies and AlexaFluor680-labeled donkey anti-goat (1:600; A-32734, A-21058; A-21084, Molecular Probes); DyLight800-conjugated goat anti-rabbit and anti-mouse antibodies (1:10000; SA5-35571 and SA5-35521; Thermo Fisher Scientific) and donkey anti-guinea pig antibodies coupled to IRDye680 and IRDye800, respectively (1:10000; 926-68077 and 926-32411; LI-COR Bioscience).

The eBioscience™ annexin V Apoptosis Detection kit FITC was purchased from Invitrogen (88-8005). To measure mitochondrial depolarization, JC-1 dye (Molecular Probes #T3168, 2 mg/ml stock in DMSO) was used.

GST- and TrxHis-tagged fusion proteins were purified from *E. coli* as described previously[18,37].

Tag-free syndapin I was generated by pGEX-6P-driven expression of GST-syndapin I in *E. coli*, fusion protein purification, overnight cleavage with Prescission protease and removal of cleaved off GST and of remaining GST-tagged syndapin I by depletion with glutathione-agarose (Antibodies Online) according to previously established procedures[19,20,62]. Syndapin I-GFP was generated from purified GST-syndapin I-GFP using similar procedures.

## Preparation of HEK293 cell lysates

24-48 h after transfection, HEK293 cells were washed with PBS, harvested and subjected to sonification for 10 seconds and lyzed by incubation in lysis buffer (10 mM HEPES pH 7.4, 0.1 mM MgCl$_2$, 1 mM EGTA, 1% (v/v) Triton X-100) containing 150 mM NaCl and Complete EDTA-free protease inhibitor (Roche) for 30 min at 4 °C as described previously[18]. Cell lysates were obtained as supernatants from centrifugations at 20000 x g (20 min at 4 °C).

## Mouse brain lysates

For endogenous coprecipitation and coimmunoprecipitation studies, mouse brains were lyzed in either lysis buffer containing 150 mM NaCl and Complete EDTA-free protease inhibitor or IP buffer (10 mM HEPES pH 7.5, 0.1 mM MgCl$_2$, 1 mM EGTA, 1% (v/v) Triton X-100, 50 mM NaCl and Complete EDTA-free protease inhibitor) using an Ultra Turrax T8 for 2×20 s. The homogenates were incubated (under rotation) for 10 min at 4 °C and centrifuged (1000 x g, 5 min, 4 °C) before use in any experiments.

For analyses of developmental expression profiles, brains from mice of different ages (E16, E18, P0, P4, P8, P12, 4 weeks, 8 weeks) were prepared and immediately homogenized in 10 ml/g tissue weight ice-cold buffer A (320 mM sucrose, 1 mM EDTA, 5 mM HEPES, pH 7.4, containing Complete protease inhibitor (EDTA-free)) using a Potter S homogenizer (Sartorius). Cell debris and nuclei were removed by centrifugation at 1000 x g (5 min, 4 °C).

The protein contents were determined and samples of equal amounts of protein were analyzed by immunoblotting using fluorescently labeled secondary antibodies and a LI-COR Odyssey system for detection and analysis (LI-COR Biosciences).

## Isolations and extractions of mitochondria

Adult rats were sacrificed and the brain was removed and homogenized in ice-cold buffer A containing Complete EDTA-free protease inhibitor using a Potter S homogenizer. Cell debris and nuclei were removed by centrifugation at 1000 x g (5 min; P1). The supernatant S1 was centrifuged at 12000 x g for 20 min to yield supernatant S2 and pellet P2, which represents a crude membrane fraction. P2 was further fractionated by centrifugation via a sucrose step gradient, essentially as described[63]. Intact, crude mitochondria were obtained as sediment fraction of sucrose gradient centrifugation.

Mitochondria from NIH3T3 cells were obtained by cell fractionation. For harvesting, a T75 flask of NIH3T3 cells was incubated with 1 ml trypsin/EDTA (0.005 µg trypsin/µl; Gibco) for 5 min at 37 °C and stopped with 9 ml DMEM containing 10% (v/v) fetal bovine serum. After centrifugation (100 x g, 4 min), the cell sediment was washed with PBS and resuspended in STE buffer (270 mM sorbitol, 10 mM Tris-HCl, 0.1 mM EDTA, pH 7.6) containing Complete EDTA-free protease inhibitor. Lysates of NIH3T3 cell cultures were prepared by ultrasound in a water bath (1 min). Lysates were centrifuged at 1000 x g (10 min, 4 °C) to yield supernatant and pellet. Analytical aliquots were taken and the postnuclear supernatant was centrifuged at 15000 x g (10 min, 4 °C) to yield supernatant and crude mitochondrial pellet.

For further purification and subsequent extraction experiments, the pellets containing crude mitochondria (of either rat brain or NIH3T3 cells) were resuspended in STE buffer, transferred onto a sucrose gradient of 1.7 M and 1 M sucrose (dissolved in 10 mM Tris-HCl and 0.1 mM EDTA, pH 7.6) and ultracentrifuged at 51000 x g and 4 °C[63,64]. The purified mitochondrial fraction was taken at the interface of 1.7 M and 1.0 M sucrose, washed with 1.5 ml STE buffer and collected by centrifugation at 15000 x g (10 min; 4 °C). The resulting pellet of purified mitochondria was then dissolved in PBS.

For analyses of the association of rhotekin2 to the outer mitochondrial membrane, isolated purified mitochondria from rat brain were incubated with alkaline Na$_2$CO$_3$, as extractability by Na$_2$CO$_3$ is typical for membrane-associated proteins[64]. Na$_2$CO$_3$ (100 mM, pH 11.0) was applied on ice for 45 min. Subsequent to a centrifugation at 100,000 x g for 1 h at 4 °C, the resulting supernatant and the pellet of Na$_2$CO$_3$-extracted mitochondria were analyzed by SDS-PAGE and immunoblotting using a LI-COR Odyssey system.

## Reconstitutions of protein interactions at mitochondria

Mitochondria, which had been isolated and purified from rat brain (see above), were resuspended in STE buffer (8 mg protein/ml). Purified mitochondria were either left untreated or were treated with 7 µM

trypsin (Thermo Fisher Scientific) for 20 min at 37 °C. Trypsin digestions were stopped with 1 µM E-64 and 1 mM PMSF. The mitochondria were then washed with E-64 and PMSF-containing STE buffer and resuspended in STE buffer.

For interaction studies, untreated and trypsin-treated mitochondria were incubated with 1.6 µM purified untagged syndapin I, with 3.5 µM purified GST-tagged rhotekin 2 or GST, respectively (1 h, 4 °C). Using centrifugation at 20000 x g for 30 min at 4 °C mitochondria were washed 3 times and the mitochondrial pellet was then analyzed for syndapin I binding in dependence of rhotekin2 presence and rhotekin2 readdition, respectively, by immunoblotting analyses against syndapin I and rhotekin2.

For reconstitutions of syndapin I-mediated tethering of mitochondria to syndapin I-coated surfaces, GFP and syndapin I-GFP were purified from bacteria as GST fusion proteins and liberated from the GST tag by proteolysis as described above. 4 µl of purified GFP (control) and GFP-syndapin I (4 µM final), respectively, were added to poly-D-lysine-coated, black 96-well plates with transparent bottom (Thermo Fisher) and let stand and dry for 1 h. The samples were then fixed with 0.1% (v/v) glutaraldehyde in PBS for 2 min. After glutaraldehyde removal and drying, the samples were incubated for 1 h at RT with purified, MitoTracker-stained brain-derived mitochondria, which were either left untreated or were treated with trypsin (see above) before. The MitoTracker staining was performed at room temperature by incubating 30 µl untreated or trypsin-treated mitochondria in 300 µl STE buffer containing a protease inhibitor cocktail and 1 µM Mito-Tracker Deep Red. After 5 min of incubation, the excess of MitoTracker Deep Red was removed by collecting the mitochondria via centrifugation (20000 x g, 30 min, 4 °C) and washing them three times with PBS. The final mitochondrial sediment was resuspended in 500 µl PBS and then used for the reconstitution experiments. After incubating the MitoTracker-stained mitochondria with recombinant purified proteins and washing with PBS, fluorescence measurements via a LICOR Odyssey system were used to show and quantify the surface-coating with proteins (GFP signal) and the amount of mitochondria recruited to the protein-enriched surfaces (MitoTracker Deep Red signal).

## Coprecipitation analyses

Coprecipitation experiments were done as described earlier[34]. In brief, GST or GST fusion proteins were expressed in *E. coli*, purified and coupled to a glutathione-matrix and then incubated with lysates from HEK293 cells overexpressing the putative binding partner.

For endogenous coprecipitation studies, mouse brain lysates generated as described above were incubated with immobilized GST fusion proteins.

After washing, bound proteins were eluted by incubation for 30 min at room temperature in elution buffer (20 mM glutathione, 120 mM NaCl, 50 mM Tris/HCl, pH 8.0). The resulting samples were analyzed by immunoblotting.

## Coimmunoprecipitation analyses

Heterologous coimmunoprecipitation experiments were in principle done according to literature procedures[20,65,66]. In brief, Flag-rhotekin2′ was coexpressed with either GFP fusion proteins of syndapin I, syndapin II-s or syndapin II-l in HEK293 cells. The cells were lyzed by addition of lysis buffer containing 150 mM NaCl and Complete EDTA-free protease inhibitor. The lysates were incubated with 5 µg mouse monoclonal anti-Flag antibodies (M2) and with 5 µg non-immune murine IgG (Santa Cruz Biotechnology), respectively (2 h at 4 °C). Subsequently, 15 µl of a suspension of protein A/G-agarose (Santa Cruz Biotechnology) in lysis buffer containing 150 mM NaCl and protease inhibitor was added to each sample (1:2 v:v). After incubation for another 3 h the protein/antibody complexes bound to the agarose were isolated by centrifugation, washed three times with lysis buffer

containing 150 mM NaCl and protease inhibitor, eluted with 2x SDS-sample buffer (95 °C, 5 min) and then analyzed by immunoblotting using rabbit anti-Flag antibodies for detection of immunoprecipitated proteins and rabbit anti-GFP antibodies for detection of coimmunoprecipitated GFP-fusion proteins.

For coimmunoprecipitations of endogenous RTKN2 together with endogenous syndapin I, protein A/G agarose beads were coated for about 1 h with affinity-purified guinea pig anti-rhotekin2 antibodies and non-immune guinea pig IgG, respectively. The antibody-coated beads were then washed twice with PBS and one time with IP buffer before being incubated for 2 h at 4 °C with mouse brain lysate.

After washing, immunoprecipitated RTKN2 and coimmunoprecipitated proteins were eluted with SDS-PAGE sample buffer and the eluates were immunoblotted with anti-rhotekin2 and anti-syndapin I antibodies.

### Analyses of direct protein/protein interactions

Direct protein/protein interactions were demonstrated by coprecipitations with recombinant proteins purified from *E. coli* according to procedures described before[34]. In brief, immobilized TrxHis-RTKN2' and TrxHis (control), respectively, were incubated with either GST-syndapin I (GST-SdpI) or GST (control) in lysis buffer containing 300 mM NaCl. Eluted proteins were analyzed by SDS-PAGE and subsequent immunoblotting.

### Liposome-binding assays

Liposomes were prepared with Folch-fraction type I lipids (Sigma) using 0.4 mg lipids in 0.56 ml chloroform/methanol (1:53 v/v) as described previously[33,34,66]. In brief, dried and subsequently water-saturated lipids were incubated overnight at 37 °C in 30 ml cytosol buffer (CB) (25 mM HEPES pH 7.2, 25 mM KCl, 2.5 mM magnesium acetate and 100 mM potassium glutamate), collected at 28 °C for 1 h at 33,734x $g$ and resuspended in 500 µl of the supernatant.

Floatation experiments were essentially done as described previously[14,32]. 5 µM fusion protein was mixed with 30 µl liposomes in 100 µl CB and incubated at 37 °C (15 min at 800 rpm). Control samples were not supplemented with liposomes. The samples were then mixed with 150 µl 75% (w/v) sucrose in CB, overlaid with 200 µl 35% (w/v) sucrose in CB and 200 µl CB and subsequently centrifuged at 200000 x $g$ for 30 min at 28 °C. Density gradient fractions of 100 µl were collected, ethanol-precipitated and subjected to immunoblotting.

### Cell culture, transfection and immunostaining

HEK293 and NIH3T3 cells were maintained in 10 ml DMEM containing 2 mM L-glutamine, 10% (v/v) fetal bovine serum and penicillin/streptomycin (Invitrogen) at 37 °C and 5% $CO_2$. HEK293 and NIH3T3 cells were transfected with plasmid DNA using TurboFect (Thermo Fisher Scientific) according to manufacturer´s instructions.

NIH3T3 cells were furthermore transfected with control (Qiagen Allstars Control siRNA) or syndapin II-specific siRNA at about 60 % confluence using the HiPerfect transfection reagent according to the manufacturer's instructions (Qiagen) and as described and verified previously[36]. For rescue experiments, cells were cotransfected with GFP-SdpII-l*. Transfected cells were maintained for additional 72 h before they were subjected to immunolabeling experiments.

For immunofluorescence analyses, NIH3T3 cells were fixed with 4% (w/v) paraformaldehyde (PFA) for 7 min, quenched with 25 mM glycine in PBS for 30 min, and finally permeabilized and blocked with 10% (v/v) horse serum, 5% (w/v) BSA in PBS (blocking solution) with 0.2% (v/v) Triton X-100. Primary and secondary antibody incubations were done according to procedures described previously[56]. Coverslips were dived in distilled water and mounted on glass slides using Mowiol 4-88 (Roth).

### Culturing of primary rat hippocampal neurons, transfection and immunostaining

Primary rat hippocampal neuronal cultures for immunofluorescence analyses were prepared from E18 rat embryos of mixed sex. The embryos were decapitated and the brains were collected in HBSS buffer (Gibco, #14175) on ice and split into the hemispheres. The hippocampus of each hemisphere was isolated, pruned of the meninges and collected in HBSS buffer on ice. Once washed again with HBSS buffer, the hippocampi were incubated in preheated 1x trypsin (Gibco, #15400; dissolved in HBSS buffer) for 20 minutes at 37 °C. To stop the digestion process, the trypsin-containing solution was exchanged by preheated Neurobasal+ medium (Neurobasal medium containing 1x L-glutamate (Gibco, #11500626, 1x GlutaMax (Gibco, #35050), 1x B-27™ (Gibco, #17504) and 100 µg/ml streptomycin and 100 U/ml penicillin) and the hippocampi were triturated with a fire-polished Pasteur pipette to dissociate the cells. Additionally, the cell suspension was passed through a sterile strainer (40 µm cut-off). Neurons were seeded onto poly-D-lysine coated glass coverslips at densities of about 180000/well (12-well plate) for 3D-live analyses using a spinning disc microscope and 60000/well (24-well plate) for immunofluorescence analyses. Neurons were maintained at 37 °C with 90% humidity and 5% $CO_2$.

Transfection of rat hippocampal neurons at 4 DIV was performed using 1.0 µg DNA in Optimem/Lipofectamine 2000 mixture (Invitrogen) per well (24-well plate) and the cells were fixed about 48 h later.

Other neurons were subjected to immunofluorescence stainings at DIV10. All fixations were done in 4% (w/v) PFA in PBS pH 7.4 at RT for 5 min.

### Immunofluorescence microscopy

Images were recorded using a Zeiss AxioObserver.Z1 microscope equipped with an ApoTome2 and with Plan-Apochromat 63x/1.4 and 40x/1.3 objectives and an AxioCam MRm CCD camera (Zeiss).

SIM was done using an ELYRA 7 equipped with Plan-Apochromat 63x/1.4 Oil DIC M27 objectives (Zeiss).

Digital images from Zeiss microscopes were recorded by ZEN2012 or AxioVision Software (Vs40 4.8.2.0) and processed and quantitatively analyzed by ImageJ (National Institutes of Health; RRID: SCR_003070) and IMARIS 8.4 software (Bitplane; RRID:SCR_007370), respectively.

### Examinations of apoptotic effects in NIH3T3 cells and in primary hippocampal neurons

Putative inductions of apoptosis were analyzed by three independent means. Incubations with staurosporine (STS) for 16 h were used as positive controls for apoptosis induction[67].

First, proteolytic caspase 3 activation[68,69] was addressed by subjecting NIH3T3 cells to immunoblotting analyses using anti-cleaved caspase 3 antibodies. Immunoblotting with anti-β-actin antibodies and immunoblotting against the mCherry reporter of the pRNAT mCherryF vector in transfections with RTKN2 RNAi and scrambled RNAi, respectively, served as reference controls. 2 µM STS for 16 h was used to induce apoptosis. The cells were lyzed 44 h post-transfection and subjected to SDS-PAGE and Western blotting.

Second, the polarization status of mitochondrial membranes was addressed by the application of the dye JC-1. In brief, untreated NIH3T3 cells, cells 48 h after transfection and 16 h after treatment with 1 µM STS, respectively, were incubated with 4 µg/µl JC-1 for 20 min at 37 °C, washed with HBSS and transferred to FluoroBrite™ DMEM (Thermo Fisher) for live cell imaging with a spinning disc microscope. Images were taken with identical exposure times for all conditions (control, STS, scr. RNAi and RTKN2 RNAi). Sum intensity projections of z-stacks composed of 14 images were used to quantify the fluorescence intensities of depolarized (green) and polarized (red) JC-1 dye. Within

each cell to be analyzed, ten ROIs were placed randomly and additionally one ROI was placed at the background. After background subtraction, the ratio of fluorescence intensities representing the depolarized (green) and polarized (red) status was calculated and represented.

Third, apoptosis induction was assessed by staining both NIH3T3 cells as well as primary hippocampal neurons with FITC-labeled annexin V. In brief, 48 h after transfection or 16 h after treatment with 1 μM STS NIH3T3 cells and untreated control cells, respectively, were incubated with 5 μl annexin V-FITC in 500 μl binding buffer (eBioscience™ annexin V Apoptosis Detection kit FITC (88-8005)) for 5 min at RT. Excess of annexin V-FITC was removed by washing with HBSS. Thereafter, cells were fixed with 2% (v/v) PFA, incubated in 25 mM glycine in PBS, washed once with PBS including DAPI (1:10000) and once with distilled water. Coverslip mounting onto slides was done using Mowiol 4-88.

Images of all conditions (control, STS, scr. RNAi and RTKN2 RNAi) were taken with z-extension and identical exposure times for the annexin V-FITC using a Zeiss AxioObserver.Z1 microscope equipped with an ApoTome2. Summed intensity projections of the images were used for quantification of the annexin V-FITC signals within three randomly chosen ROIs per cell. An additional ROI at the edge of the image served as background control. After background subtraction, annexin V-FITC fluorescence intensities in transfected (scr. RNAi and RTKN2 RNAi) and in STS-treated cells were presented as normalized values compared to untreated cells.

### Analysis of mitochondrial length and density in NIH3T3 cells

The analyses of mitochondria were either based on cotransfected mitoGFP or on the immunofluorescence signals obtained by antibodies directed against the mitochondrial marker cytochrome c. The anti-cytochrome c signal was filtered using the *local normalization (LN)* plug-in of ImageJ. This tool equalizes signal variances and enables the discrimination of individual mitochondria with their various shapes[70]. The length of all individual mitochondria within the cells was measured manually using the *segmented line tool*.

As an alternative method of mitochondrial length determination, the *skeleton* tool in ImageJ was used for automatic thresholding and analysis. Note that the use of the *skeleton* tool often fuses neighbored fluorescence maxima and thereby tends to lead to higher mitochondrial length values than evaluations by a scientist (see Supplementary Fig. 7d versus 7e for direct comparison).

The often very bright signals of mitoGFP overexpression usually also lead to higher absolute values than staining mitochondria with antibodies against endogenous cytochrome c – the latter of which may have the technical disadvantage that cytochrome c may not always be evenly distributed and/or immunodetected inside of mitochondria leading to apparently higher mitochondrial numbers and to lower mitochondrial length values by this method.

Mitochondrial densities in NIH3T3 cells were determined as number of mitochondria detected by anti-cytochrome c immunostaining per cellular ROI. The outer border of the cellular ROIs was defined by the fluorescence signal of GFP (reporter of RNAi plasmids; reporter of IRES-driven coexpression of untagged rhotekin2) used as cell filler. The GFP signal was outlined and thereby the cellular ROIs were determined by using the *polygon selection tool* in ImageJ.

### Transmission electron microscopy and analyses of mitochondrial density and length

Anaesthetized adult male WT and *syndapin I* KO animals (4 animals per genotype) were perfused transcardially with 0.9 % (w/v) NaCl, and then with 2% (w/v) PFA, 2.5% (v/v) glutaraldehyde, 1% (w/v) sucrose, 0.1 M cacodylate buffer, pH 7.3. The brain was isolated and brain pieces (approximately 1 mm x 1 mm x 1 mm) of the hippocampal CA3 region

were prepared, washed extensively with PBS and sectioned into thin slices in random orientations. The thin slices obtained were incubated for 1 h at RT in 2% (w/v) OsO₄, 0.8% (w/v) K₄[Fe(CN)₆] in PBS, dehydrated in a graded ethanol series and incubated for 30 min with 2% uranyl acetate (in ethanol). After washing with ethanol, slices were embedded in EPON and prepared for EM.

Samples were examined with a Zeiss EM912 at 5000x magnification. Images were recorded digitally using a CCD camera 794 (2k) (Gatan).

The mitochondrial density analyses were based on randomly selected images and counting of mitochondria per image (four animals each; WT, 204 images; KO, 163 images; WT, 2263 mitochondria; KO, 1901 mitochondria).

Mitochondrial areas were outlined and determined by ImageJ. Mitochondrial length measurements also were conducted using ImageJ. The longest extension (outer membrane measurements) of the respective mitochondrion was defined as mitochondrial length. Note that due to thin sectioning, mitochondrial "length" determinations via electron microscopy offer an excellent resolution but in fact represent randomly orientated measurements, including cross-sections of ellipsoid or elongated mitochondria and therefore yield relatively low averages of mitochondrial "length" data.

Due to the high heterogeneity in mitochondrial distributions in the individual images and in individual sizes of mitochondria (and/or of their cross-sections) but not overall from animal to animal, mitochondrial parameters were calculated and statistically analyzed per image area (density) and per individual mitochondrion (length, i.e. maximal extension in section), respectively, to cover the highest biological deviations.

All analyses were performed in a blinded manner.

### Mitochondrial length and distribution analyses in the dendritic arbor and in the axon of developing neurons

Cultured rat hippocampal neurons were transfected at DIV4 with either rhotekin2 RNAi or scrambled RNAi and cotransfected with control pIRES-mitoGFP or RTKN2*-pIRES-mitoGFP or RTKN2*ᐃᴷᴿᴬᴾ-pIRES-mitoGFP, respectively, and fixed and immunostained for the dendritic marker MAP2 48 h post-transfection.

In alternative experiments, neurons were transfected at DIV4 with either rhotekin2 RNAi or scrambled RNAi, respectively, cotransfected with pIRES-EGFP, RTKN2*-pIRES-EGFP or RTKN2*ᐃᴷᴿᴬᴾ-pIRES-EGFP and fixed 48 h post-transfection. Mitochondria were immunostained with anti-cytochrome c antibodies and the dendritic arbor was visualized using anti-MAP2 antibodies

Images of transfected rat hippocampal neurons were recorded as z-series with 0.25-0.3 mm intervals using a Zeiss AxioObserver Z1 microscope equipped with an ApoTome2. Summed intensity projections were analyzed.

In additional experiments, images from MitoTracker-stained mitochondria from 3D-time-lapse analyses were used for mitochondrial length determinations.

The lengths of mitochondria were determined as described for the analyses in NIH3T3 cells (see above).

For analyses of mitochondrial density, the number of mitochondria of one randomly chosen primary dendrite were counted and divided by dendrite length.

For overall distribution analyses in dendrites, the definitions for proximal and peripheral dendritic segments were adapted from the literature[2], i.e. first 20 μm from cell soma were defined as proximal and following 20 μm as peripheral (please also see Fig. 3i for schematic depiction).

### 3D-time-lapse analyses of mitochondrial mobility

For mitochondrial tracking experiments, rat hippocampal neurons were plated on 18 mm coverslips coated with poly-D-lysine.

Transfections with rhotekin2 RNAi or scrambled RNAi vectors coexpressing farnesylated mCherry to highlight neuronal morphologies were conducted at DIV4. Time-lapse measurements of living transfected hippocampal neurons were performed 72 h post-transfection. Prior to analysis, mitochondria were labeled with MitoTracker DeepRed (1:10000) for 5 min at 37 °C, washed once with HBSS and imaged in FluoroBrite™ DMEM. 3D-time-lapse microscopy was performed by using a motorized Zeiss AxioObserver equipped with a spinning disc unit, an incubator, and an EMCCD camera, as described previously[14,20]. Full z series were recorded every 10 s over a time span of 10 min. In alternative experiments, to track the mobility of mitochondrial better, an improved temporal resolution (2.4 s/frame) was used at the expense of z resolution for the detection of dendritic branch induction (merely 3 multicolor z planes were recorded).

Dendritic branch induction events occurring during the 10 min recordings of neuronal development were identified. Subsequently, mitochondrial velocities at branch induction sites (identified by tracing dendritic morphology by CherryF; 1 μm dendritic segment around dendritic branch initiation) were analyzed in a period of 60 s prior to 60 s after dendritic branch induction. In parallel, mitochondrial movements in non-branching areas (not branching during entire 10 min time of recording; distance to any branch induction site at least 1 μm) were analyzed.

Dynamic analyses of labeled mitochondria were performed using the tracking function (*spot*) in IMARIS 8.4. From the generated tracks, maximum track speed, individual spot acceleration and displacement length were extracted as quantitative parameters. All of these parameters describing mitochondrial dynamics were determined in relation to assay-intrinsic control data.

### 3D-time-lapse analyses of mitochondrial positioning in relation to dendritic branch induction events

3D-time-lapse microscopy using spinning disc microscopy (full z series; interval, every 10 s) was used to record the morphological development of hippocampal neurons at DIV7 in relation to the positioning of mitochondria in the growing dendritic tree. Mitochondria were labeled with MitoTracker DeepRed. The morphology of the cells and dendritic branching events were identified and tracked by the fluorescence of mCherryF as reporter coexpressed with either rhotekin2 RNAi or syndapin I RNAi and scrambled RNAi, respectively, as described previously[14,20,56]. As described before, at least 10 min of dendritic development were recorded[56]. The first discernible local expansion of dendritic morphology was defined as t = 0 s of dendritic branch induction, as established before[20]. Mitochondrial positioning in correlation with identified branch induction events was tracked in time intervals ranging from t = −60 s to t = +60 s.

Frequencies of mitochondria at sites of dendritic branching area (1 μm-long dendritic segment at the branching site) as well as in non-branching areas (cumulated length 138.47 μm) were analyzed within the same cells and time frames.

Further analyses of the positioning probability were done inside of a 3 μm-dendritic segment around identified nascent and forming branch sites. The 3 μm segment was separated into the central position and the immediately neighboring dendritic segments on either the proximal side (i.e. towards the cell soma) or the distal side. Each of these subsegments was 1 μm in length. The positioning of mitochondria in the vicinity of dendritic branch site inductions and propagations were scored for their positioning in the distal, central and proximal dendritic segment at t = −60 s, t = 0 s and t = +60 s. The results were expressed as probabilities of occurrence ranging from 0 to 1.

### 3D-time-lapse analyses of dendritic branch induction events

Dendritic branch induction events were analyzed by 3D-time-lapse microscopy. Time periods of at least 10 min were recorded using spinning disc microscopy. In order to quantitatively determine the frequency of dendritic branch induction, the length of four randomly chosen dendritic segments was determined and the number of occurring branch induction events (minimal protrusion length, 1 μm) on those dendritic segments was counted, as described previously[19]. Numbers of dendritic branch initiation events were expressed per 20 μm dendrite and minute.

### Quantitative analyses of dendrites of hippocampal neurons in culture

For loss-of-function analyses and the corresponding rescue experiments, hippocampal neurons were transfected at DIV4 and fixed and immunostained for the dendritic marker MAP2 at DIV6. Transfected neurons were imaged by systematic sampling across the coverslips. 5 independent coverslips per condition from at least 2 independent neuronal preparations were analyzed.

Morphometric measurements of the dendritic arbors of individual transfected neurons (highlighted by expression of GFP and mCherry reporters, respectively) were based on immunolabelings of endogenous MAP2 as independent dendritic marker and were conducted by using IMARIS 8.4 software (Bitplane).

In confirmatory experiments, morphometric analyses were conducted based on the fluorescence signal of overexpressed GFP instead of endogenous anti-MAP2 immunostainings. Also in these analyses, MAP2 was immunostained to highlight which cellular protrusions are dendritic of nature and should be traced following the GFP fluorescence signal in quantitative, morphometric analyses.

To reconstruct the dendritic tree of individual neurons, filaments were manually drawn by the IMARIS *filament mode*. The origin of the *filament* was placed in the soma center. Minimal length for primary dendrites was set to 10 μm, as described previously for definitions of dendritic branches[15,16,56]. Analytical parameters, such as Sholl analyses[71], the number of dendritic branch points, the number of dendritic terminal points, the total dendritic filament length sum and the number of segments per branch depth as parameter for more specific dendritic tree complexity aspects were software-determined (IMARIS 8.4). For representation purposes, we focused our Sholl analyses on up to 50 μm around cell center, as this areas covers most of the dendritic tree and thus is most informative for DIV6 neurons. For branch depth analyses we considered dendritic segments up to the sixth branch depth level.

### Statistics

No statistical methods were used to predetermine sample size. Sampling was done in a systematic manner. Therefore, the *n* numbers reflect the outcome of the systematic searches and may differ.

No "outlier" suggestions were computed. No data points were excluded, but all technically sound quantitative evaluation data points were taking into account and averaged to fully represent biological and methodological variabilities. *n* numbers, numbers of independent assays and statistical significance analyses are reported directly in the figures and the legends, respectively.

Statistical analyses were done using GraphPad Prism 8 software. Suitable statistical tests were chosen according to normality distribution testings and the conditions to be compared. The respective statistical test used and *P* values calculated for statistical differences are reported in the figure legends. Note that extremely small *P* values (<0.0001) cannot be expressed by GraphPad Prism 8 software anymore and are therefore solely reported as **** $P < 0.0001$. In general, statistical significances are defined by * $P < 0.05$; ** $P < 0.01$; *** $P < 0.001$; **** $P < 0.0001$ throughout.

### Reporting summary

Further information on research design is available in the Nature Portfolio Reporting Summary linked to this article.

## Data availability

The authors declare that all data supporting the findings of this study are available within the paper and its supplementary files. Source data are provided with this paper.

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

## Acknowledgements

This manuscript is in memoriam of A. Bayrhammer. We thank S. Berr, A. Kreusch, and M. Öhler for technical support. We thank the Microverse Imaging Center (and Aurélie Jost) for providing microscope facility support for data acquisition (and data analysis). The ELYRA 7 (used for producing SIM images) was funded by the Free State of Thuringia with grant number 2019 FGI 0003. The Microverse Imaging Center is funded by the Deutsche Forschungsgemeinschaft (DFG, German Research Foundation) under Germany´s Excellence Strategy – EXC 2051 – Project-ID 390713860. This work was supported by the DFG (grants KE685/7-1 to M.M.K. and QU116/9-1 to B.Q.).

## Author contributions

J.T., R.D., A.B. and D.K. performed experiments. J.T. and M.M.K. visualized data. J.T. co-interpreted data and co-wrote parts of the manuscript. B.Q. and M.M.K. conceived the project, designed experiments and wrote the manuscript.

## Funding

## Competing interests

The authors declare no competing interests.
