## [Transparent Peer Review file · Nature Communications]

Mitochondria are positioned at dendritic branch induction sites, a process requiring rhotekin2 and syndapin I

Corresponding Author: Dr Michael Kessels

Version 0:

Reviewer comments:

Reviewer #1

(Remarks to the Author)

Troger et al present work arguing that mitochondria are spatiotemporally localized to sites of nascent dendritic branching, and play a pivotal role in the development of the dendritic arbor. They provide evidence of a potential mechanism that likely contributes to this localization in neurons. This is a fundamentally important question to the field and is of high interest to both developmental and neurodegeneration researchers.

The most noteworthy result centers around the live imaging observations the authors made showing that mitochondria are localized to the central location where a new branch will form at least 60 seconds before the new branch forms, and that this localization is impaired in rhotekin2-deficient neurons. This is quite novel as a few mechanisms have been identified for axonal positioning during development but none for dendrites.

While these are important and interesting results, I do have a number of concerns

1. Mitochondrial size measurements don't seem to be in line with other work in the field. Throughout the manuscript dendritic mitochondrial lengths are presented as less than 0.5 microns. In all other publications on the topic, dendritic mitochondria are suggested to be much longer/larger in size (see Li et al, Cell 2004; Popov et al, JCN 2005; Lewis et al, Nat Commun 2018; Rangaraju et al, Cell 2019). I realize that most of these are from more mature neurons but even in young neurons (see Chang, Reynolds Neuroscience 2006 or Hatsuda et al, Development 2023) dendritic mitochondria average length has been reported as 2 microns or longer in length.

a. Perhaps this is an artefact of fixation? If so, how can this be ruled out as a potential confounder of the mitochondrial size? Maybe co-transfecting a plasmid encoding for a mitochondrial targeted GFP or other fluorescent protein and performing live imaging could resolve this?

b. What is the magnification used for acquiring these images. Perhaps the magnification used could be increased?

2. In the loss of Rhotekin2 experiments, how can an effect on branch initiation independent of mitochondrial localization be ruled out? I think for the authors to make the claims about Rhotekin2's role in mitochondrial positioning regulating branching some sort of sufficiency evidence is necessary.

a. Does overexpression of Rhotekin2 result in increased localization of mitochondria to branch sites and result in increased branching?

b. How does loss of rhotekin2 affect branching dynamics? Is it branching isn't initialized or does the branch form but isn't stabilized?

c. Ruling out changes in actin structure at branch points following Rhotekin2 loss seems important.

3. The first half of the manuscript and the second half of the manuscript are never linked together. In the first half, a story is presented that RTKN2/Syndpin regulate mitochondrial size changes and RTKN2 controls this, while in the second half the story switches to RTKN2/Syndpin control localization to sites of branching and regulate branching.

a. Are these two parts related or independent? If independent and not related to the branching, it seems like this shouldn't be a part of the narrative.

4. It is claimed for the localization part of the story that calcium plays some role in this, (see model at the end of figure 7), but

no data is presented to support this in the manuscript.

a. How? Are calcium levels different at branch points? Are calcium dynamics different?

b. Perhaps addressing this may allow for the linkage of the mitochondrial size and tethering phenotypes (Hatsuda et al, Development 2023)?

Other comments

1. All the bar graphs should be presented with spread of the data represented (box plots, individual points or violin plots)
2. It is hard to visualize the stated results with the images presented in figure 1
3. Validation of anti-Rhotekin2 antibody should be presented in the manuscript somewhere. For instance, does staining decrease or go away when RNAi is used?
4. The text mentions overexpressed Rhotekin2 in neurons but no data is provided in the figures
5. Figure 2c panels have no labeling for what they are
6. Validation should be provided for results in Figure 2d-e. How can it be claimed as a mitochondrial protein if no validation of purity is shown? At least markers for nucleus, cytosol and ER should be presented
7. For figure 2f, is this axonal or dendritic? Perhaps a slightly expanded view could also be presented for context
8. Figure 3 images with high mag insets don't seem to agree with the quantification data. C looks more fused than B
9. Rhotekin2 is suggested to affect fission/fusion, this could be assessed with experiments using photoactivatable-GFP (see Karbowski et al, J Cell Biol 2004)
10. For mitochondrial motility data, either supplemental videos or kymographs should be included for assessment of motility
11. Supplemental videos for figures 4 & 5 data would really add to the impact of the work
12. The text and figure 6 don't seem to be in sync. Text refers to Fig6p but figure 6 has no panel p. Also the text description of figure 6o doesn't seem to line up with the data shown
13. For figure 7, it would be useful to include a panel with just the reporter to see the dendritic compartment without all the other overlapping information.

Reviewer #2

(Remarks to the Author)

The manuscript by Tröger and colleagues builds on previous work from their group and others, to further investigate the molecular mechanisms of branch formation in developing neurons. Following the authors' previous findings demonstrating syndapin function in dendritic branching, they identified a binding partner for syndapin which recruits mitochondria to nascent branching points in the growing dendrite. The authors provide series of biochemistry data that strongly support mitochondrial localization of rhotekin2 and its specific binding to syndapin I. Then they indicate that rhotekin2 regulates mitochondrial shape and positioning in dendrites and induces new branch formation by morphological analyses of fixed and live neurons.

If proven true with sufficient evidence, this study would provide important additional information about the function of mitochondria in dendrite formation. Unfortunately, however, the methods for mitochondrial dynamics analyses, which should be the cornerstone, are not adequate, and many data do not convince readers nor provide sufficient data to support the conclusions. I believe it is necessary to redo most microscopic analyses using appropriate molecular tools and microscope setups before the work can be reconsidered.

Major comments

1. Most of the histological analyses of mitochondria were done using cytochrome C as a mitochondrial shape marker. However, cytochrome C is unevenly distributed in the intermembrane space of mitochondria in a punctate manner, making it unsuitable for mitochondrial shape analysis. This is evident from the fragmented localization in the figures 2 and 3, which makes it difficult to distinguish individual mitochondria that are densely overlapped within dendrites. The measured values of the mean length in Fig. 3e are therefore much smaller than those in previous studies, raising concerns of their accuracy (Li et al., Cell 2004; Cho et al., Nat Comm 2017; Hatsuda et al., Development 2023). Shape analysis relies on consistent and homogeneous distribution of the marker proteins for accurate measurements. GFP or its derivatives with a mitochondrial localization sequence (mitoGFP) is widely used for this purpose.

2. The velocity of mitochondrial transport in pyramidal neuron dendrites are ranged between 0.3-1 $\mu\text{m}/\text{sec}$ in previous studies (MacAskill et al., Neuron 2009; van Spronsen et al., Neuron 2013; Loss and Stephenson, Mol Cell Neurosci. 2017). In this study, the live imaging of dendritic mitochondria was conducted at 10 sec intervals, during which mitochondria are expected to move several times their own length in a single frame, making accurate tracking difficult. The present study showed that even the maximum track speed of mitochondrial motility was about 0.06 $\mu\text{m}/\text{sec}$, an order of magnitude lower than previous studies. Images and videos should be shown for the readers to validate the analyses.

3. Related to the above, dendritic mitochondria move rapidly to and fro along the longitudinal axis of dendritic shafts. From the small-step irregular movements shown in Fig.6, I suspect that the authors might have observed Brownian motions of stationary mitochondria rather than tethering or posing of moving mitochondria at dendritic branch points. Even if this were active movements, it is questionable if such subtle difference in the stationarity of mitochondria inside and outside the branch point make such a decisive difference in the chemical environment for determination of branch initiation sites. Here mitochondria were labeled with MitoTracker and traced in dendrites demarcated by farnesylated mCherry. It is not easy to distinguish mitochondria from those of neighboring cells as MitoTracker stains the mitochondria of all cells in high-density primary culture. I am also concerned about the punctate appearance of membran-targeted mCherry signals which should evenly stain dendritic membrane. Low magnified views and the videos they used for analysis should be shown. I would

once again recommend mitoGFP or mitoDsRed for monitoring mitochondrial dynamics in live cells.

4. In Figs. 4 and 5 the authors insist that mitochondria are stalled at the newly formed dendritic branch sites by rhotekin2. Dendrites of differentiating pyramidal neurons in the first week of culture are highly dynamic and frequently extend transient filopodia along the shaft, and only few of them are stabilized to form new dendritic branches. Observation of a small dendritic compartment for only 2 minutes is not sufficient to correlate mitochondrial motility and dendrite dynamics. A long time-lapse imaging of over 30 min to several hours is required to claim that organelle tethering precedes new dendritic branch formation (Cui-Wang et al., Cell 2012).

5. Mitochondrial tethering has also been implicated in axon branch formation (Courchet et al., Cell 2013; Spillane et al., Cell Reports 2013). This paper avoids mentioning the localization and function of rhotekin2 in the axon. However, differences in dynamics and regulation of mitochondria in the axon and dendrites are important and the authors need to discuss whether rhotekin2 function is dendrite-specific. At least the length of axonal mitochondria and the number of axonal branch points should be comparatively analyzed in rhotekin2-depleted cells.

Minor comments

1. The TEM analysis is described as having been done on hippocampal tissue, but must have been done by identifying mitochondria in the dendrites of a specific type of neurons (such as CA3 pyramidal neurons). The size and shape of mitochondria are very different in distinct cell types and in different subcellular compartments in neurons.

2. The blot in Fig.1k contains many thick faint bands. Are they background signals?

3. The neuron transfected with RTKN2 RNAi in Fig.3b looks like an interneuron (with a small soma and no clear polarity). Due to a limit to the number of antibodies for multi-color IF, it may be better to use a pyramidal neuron marker (such as Math2) rather than Map2 for morphometry in Fig. 7, since axons and dendrites can be distinguished by morphology labeled by a volume marker.

4. I believe that analysis of functional interaction of syndapin I and rhotekin would strengthen the conclusion of the manuscript. If mitochondrial tethering to the nascent branch points via syndapin I-rhotekin2 interaction is a mechanism of dendrite branch formation, the dendritic overgrowth in syndapin I overexpressing neurons (Schwintzer et al., 2011) should be at least partly rescued by rhotekin2 knockdown.

5. Judging from the stretched signals and the horizontal stripes of artifacts, Fig.6j appears to be a forced reconstruction of z-serial SIM images with weak signals, which does not provide evidence supporting the signal co-localization. The authors should use single z-plane images to demonstrate that syndapin signals localize at the membrane curvature of branch points and they are juxtaposed by rhotekin2 and mitochondria.

6. The spatial resolution of Elyra 7 is only 120 nm and one cannot measure distances of 40 and 80 nm. Specific techniques such as fluorescence cross correlation spectroscopy (FCCS) or proximity ligation assay are more suitable to detect the association of molecules.

Version 1:

Reviewer comments:

Reviewer #1

(Remarks to the Author)

In the revised manuscript, the authors have provided new data using alternative methods to validate their mitochondrial size findings, as well as presented new data for Rhotekin2 overexpression on branching and the requested validations thus addressing most of my concerns.

However, the authors propose that Rhotekin2 is a mitochondria localized protein and syndapin is localized to branch points to recruit the mitochondria to this site. Figure 6J doesn't appear to support this model though. Syndapin appears to be all along the dendrite, while the Rhotekin2 appears to localize to the branch point, but not by surrounding the mitochondria. The authors may want to be more cautious in their interpretations/model and should explain this discrepancy in the discussion.

In response to main point 4 about the role of calcium, I was not intending to suggest that calcium be removed from the model but instead that showing data that calcium dynamics are altered in the nascent dendrites/branch points following Rhotekin2 manipulation would strengthen the suggested model, as well as provide a potential reason for Rhotekin2 to regulate both mitochondrial size and anchoring at the same time. It is well established that mitochondrial calcium uptake plays important roles both in the axon and dendrites (see work from Josef Kittler and Franck Polleux labs).

Paragraph 3 of page 9 (line 171) states "The reduction in mitochondrial length upon rhotekin2 RNAi" shouldn't this be increase?

The authors may want to be careful about the interpretation of Rhotekin2's interaction with Dynamin 2 but not Drp1 for mitochondrial fission as the dynamin 2' role is somewhat controversial in mitochondrial fission (see Fonseca TB et al,

Reviewer #2

(Remarks to the Author)

The authors strengthened the evidence of the functional interaction of RTKN2 and syndapin1 with additional experiments. However, they did not answer many concerns raised in the first review. The most important concern was about the quality of imaging, which is the key to the conclusions in the second half of the manuscript that under deficiency of RTKN, mitochondria are not tethered to the plasma membrane, causing increased movement of mitochondria and down-regulation of dendrite formation. Regrettably, the temporal and spatial resolution is insufficient to evaluate the mitochondrial motility to support their conclusion. In the present form, this study does not support sufficient evidence that RTKN2 and syndapin1 regulate dendritic branch formation by tethering mitochondria to the plasma membrane transiently at branch initiation sites. The mechanism of mitochondrial elongation in RTKN2 deficiency is also based on speculation from previous studies, and its significance in the dendritic phenotypes is not explained.

Fig.6 is a key to demonstrate the mechanism of RTKN2 function. I wonder how the spots were determined in the analyses of mitochondrial speed by spot tracking in Fig. 6a-i. The videos 4-1,5-1,7-1,7-2 show intermittent movement at 10-second intervals, with brightness fluctuations due to focus shifts, making it seem challenging to continuously track the same position of mitochondria which dynamically change their shape in the videos. It seems that spot object of IMARIS was used, but the authors need to indicate the method by which the spots were determined. Considering the length of the mitochondria measured using MitoTracker (~0.8 μm ; ExtFig. 3-1), it is speculated that even a slight shift in the spot's position could result in random movements similar to those shown in Fig. 6. I am afraid that the increased movements in RTKN2 KD could be due to the increased mitochondrial length to ~1.4 μm . It is hard to evaluate the results until it is clearly negated that the difference was caused by the inappropriate processing of the movies when the trajectories were overlaid. I believe it is necessary to improve the temporal resolution, which can be at a lower magnification, and quantify how the "slow movement" they define changes in the RTKN2 ND condition.

Given the quality of the image in Fig. 4b, there are concerns about the analyses in Figs. 4, 5, and 7, too. Nonetheless, I would accept the conclusion that new branches could be more likely to form at the sites where mitochondria are tethered, considering the recent studies demonstrating that tethered mitochondria provide chemical environment supporting dendritic branch maintenance (Faits et al., 2016; this paper indicates that mitochondria are condensed only after a branch forms) and spine maturation (Rangaraju et al., 2019). Rangaraju et al. compared spine maturation with or without ablation of mitochondria in 10-25 μm dendritic compartment. On the other hand, I wonder if such differences could occur on a scale of +/-1 μm where mitochondria quickly move around the tethered positions? Dendritic mitochondria are 2 μm in average in many studies, and in this paper 0.8~ μm by MitoTracker. I asked in the first round review whether the increase in this level of movement in RTKN KD or branch outside, if it exists at all, would create enough difference in the local chemical environment to determine whether or not a protrusion forms. Mitochondria are highly abundant in dendrites and occupy a large percentage of the arbor (Lewis et al. 2018 also shown in Ext Fig. 3-1). The authors also estimated ~0.8 mitochondria per 1 μm segment with or without RTKN2, suggesting that new branches could be often centered by a mitochondrion. One also wonders if overall dendritic length and branch number could significantly decrease just by the increase in slow diffusive movement of mitochondria if density and distribution are not affected (Fig. 3). In order to link the reduction of new branch formation to the altered morphology in RTKN2 KD, branch formation and elimination should be quantified by long term imaging at low magnification (Horton et al., Brain Cell Biol. 2007; Wu et al., PLoS ONE 2015).

The SIM image also remains unchanged and insufficient to support the localization of syndapin to the plasma membrane or the close proximity of syndapin/RTKN/mitochondria. Individual z-plane images instead of the 3D reconstruction should be shown for better evaluation.

The authors previously showed transient accumulation of syndapin 1 at branch initiation sites using live imaging of fluorescently labeled recombinant proteins (Izadi et al. 2021). I would be more convinced if they demonstrate that fluorescently labeled RTKN2 and mitochondria accumulate at branch initiation sites at this time resolution and magnification.

Mitochondria length

In the revised manuscript the authors have added new data using a volume marker and measured dendritic mitochondria as ~0.8 μm , slightly longer than they did with CytC (0.4 μm) but not yet comparable to previous literature. They included both sets of data, assuming that the variation would be seen using different methods. I disagree with this point and believe it is simply because the method using punctate CytC is inaccurate.

Mitochondrial size is very diverse in different cell types and subcellular compartments. Most studies quantifying dendritic mitochondria length in cultured hippocampal neurons by a volume marker reported similar length of around 2 μm as you indicated in the letter (Chang et al., 2006; Lewis et al. 2018; Hatsuda et al., 2023). Cho et al. (Nat Comm 2017) did not distinguish dendrites and axons, reporting 1 and 1.5 μm in two independent analyses, probably due to significantly smaller mitochondria sizes in the axon (mitochondrial length is not critical in their paper). Triolo's paper is not at all informative, as it uses adult muscle stem cells for measurements. I don't understand why the authors insist on values that are so different from results repeatedly shown in previous studies. The authors claim that the relative length of mitochondria was similarly altered by RTKN2 and syndapin1 deficiencies, but we cannot accept something for which the absolute length is not correctly measured.

Going back to the EM images in the manuscript, mitochondria were only 0.4 μm because they were seemingly cut in the transverse (cross) plane, judging from their round shape. Mitochondria in the longitudinal plane of dendrites should appear

tubular (Popov et al., J. Comp Neurol. 2005). It is also unclear whether the mitochondria being measured were in the dendrites. More detailed analyses and methods (which layer and at what angle were observed, how dendrites were identified) should be provided.

Mitochondrial velocity

'Mitochondria movement' in neuronal processes has long been studied and usually refers to microtubule-dependent transport (the 'fast movement' in this paper). Previous studies have generally classified mitochondria into motile and stationary, but here again the authors challenge this and divide them into fast- and slow- moving. The authors rename what has been considered 'stationary' as 'slow-moving', which should be clarified in the text as it is confusing if there are the third class slow-moving mitochondria in addition to the fast moving and stationary ones.

The authors describe the average speed of all mitochondrial movements ($\sim 0.07 \mu\text{m}$), which I believe should be removed for two reasons: At this time resolution, the fast-movement cannot be measured correctly as the authors admitted. It also does not seem to make sense to represent the average speed of constant directional transport and non-directional short-step diffusive motion. It would be more helpful for readers if the authors clarify the new categories in this paper by showing the distribution of actual speed and directionality of individual mitochondria.

Axonal mitochondria

Tethering of mitochondria in axonal branching points are well studied and the authors should at least discuss the possible function of RTKN2 and syndapin1 in the axon. The authors have newly measured the mitochondrial length in the axons and shows that they are longer in RTKN2 KD. In normal cells, mitochondria in axons should be shorter than those in dendrites, but they are almost the same length here, raising doubts about whether the measurements are accurate. Anyway, as mitochondrial length and tethering are not related, the effect of RTKN2 loss on tethering should be examined or at least discussed as it is the main conclusion of this study.

Dendrite morphometry of cultured hippocampal neurons

The authors stated that 'primary cultures prepared from the hippocampus are mostly reflecting pyramidal neurons', but this is incorrect. Hippocampal primary cultures prepared from both mice and rats contain significant proportions of non-pyramidal neurons which have very different dendritic arbor morphologies (Horton et al., 2007; Wu et al., 2015). I was not able to find the culture protocol used in this paper, as the reference cited as describing detailed methods still omit detailed methods by citing another paper, and another paper did the same. If the protocol by Banker's group was used, the culture likely contains significant proportions of granule cells and interneurons. The morphometries in Fig.8 may warrant reexamination.

For your reference in future analyses, Map2 is a good marker of dendrites, but it is not suitable for measuring dendritic length, as it does not fill the entire dendritic length. The author may want to reexamine Fig.8e if it was based on Map2 staining.

Version 2:

Reviewer comments:

Reviewer #1

(Remarks to the Author)

Although the authors cannot fully address all questions due to technical limitations, they produced new data with high resolution SIM imaging, and with increased temporal resolution. Overall, I am satisfied that the authors have addressed most of the concerns of the reviewers.

One point that remains unclear and un-discussed from my point of view is the fact that Rhotekin2 is only highly expressed until P8. While I'm not advocating for more experiments to look at Rhotekin2's role in later developmental stages or maturation, I think a discussion of this point (along with how it fits with syndapin-1) is warranted since the authors want to link it to both neurodevelopmental and neurodegenerative disorders in the last paragraph.

REVIEWER COMMENTS

Reviewer #1 (Remarks to the Author):

Troger et al present work arguing that mitochondria are spatiotemporally localized to sites of nascent dendritic branching, and play a pivotal role in the development of the dendritic arbor. They provide evidence of a potential mechanism that likely contributes to this localization in neurons. This is a fundamentally important question to the field and is of high interest to both developmental and neurodegeneration researchers.

The most noteworthy result centers around the live imaging observations the authors made showing that mitochondria are localized to the central location where a new branch will form at least 60 seconds before the new branch forms, and that this localization is impaired in rhotekin2-deficient neurons. This is quite novel as a few mechanisms have been identified for axonal positioning during development but none for dendrites.

While these are important and interesting results, I do have a number of concerns

1. Mitochondrial size measurements don't seem to be in line with other work in the field. Throughout the manuscript dendritic mitochondrial lengths are presented as less than 0.5 microns. In all other publications on the topic, dendritic mitochondria are suggested to be much longer/larger in size (see Li et al, *Cell* 2004; Popov et al, *JCN* 2005; Lewis et al, *Nat Commun* 2018; Rangaraju et al, *Cell* 2019). I realize that most of these are from more mature neurons but even in young neurons (see Chang, *Reynolds Neuroscience* 2006 or Hatsuda et al, *Development* 2023) dendritic mitochondria average length has been reported as 2 microns or longer in length.
 - a. Perhaps this is an artefact of fixation? If so, how can this be ruled out as a potential confounder of the mitochondrial size? Maybe co-transfecting a plasmid encoding for a mitochondrial targeted GFP or other fluorescent protein and performing live imaging could resolve this?
 - b. What is the magnification used for acquiring these images. Perhaps the magnification used could be increased?

We acknowledge the concern of the reviewer that some of the data in the literature was obtained by use of ectopically overexpressed mitochondrial fillers (e.g. mitoGFP), while we used endogenous anti-Cytc signals to highlight mitochondria and that this difference in method may lead to different values of mitochondrial sizes in our comparisons of conditions when compared to absolute values reported in the literature.

As the reviewer suggested, we first again carefully reviewed the literature concerning mitochondrial sizes. Second, we also addressed these concerns by different sets of experimentation included in the revised manuscript (see below).

Literature: While we unfortunately did not find any quantitative mitochondrial length analyses in Rangaraju et al., 2019 *Cell*, the remaining literature cited by the reviewer used standard (confocal) light microscopy and described that mitochondria in hippocampal neurons were globular or tubular in shape and highly variable in size:

Li et al., 2004 *Cell* (confocal analyses of mitochondria visualized with overexpressed MitoDsRed) reported the average to be 2.3 μm . However, it has to be noted that this average was affected by a small proportion of dendritic mitochondria that appeared extremely long ($>10 \mu\text{m}$, 2.6% of 230 mitochondria). It remained unclear whether these indeed represented single mitochondria or trains of several individual mitochondria that were not resolved by standard light microscopy.

Popov et al 2005 *J. Comp. Neurol.* conducted a comprehensive EM study in different hippocampal areas based on serial sections and described continuous membrane structures with mitochondrial appearance that were up to 30 μm long. As a major focus of the study was the smooth endoplasmic reticulum (SER), its relation to mitochondria and in particular the relationship of the SER to postsynapses/dendritic spines and as the length of mitochondrial structures was not quantified, it remained somewhat open how abundant these structures were. In light of the light microscopic data of Li et al., 2004 *Cell*, which in comparison have a much lower resolution to discriminate individual mitochondria and reported these structures as an extremely minor fraction, it is safe to assume that the structures focused on by Popov et al., 2005 represent the same, extremely small population of especially elongated dendritic mitochondria.

Chang & Reynolds, 2006 *Neuroscience* used overexpressed mito-eYFP and reported mitochondrial length (calculated from skeletonized fluorescent objects) to be $2.01 \pm 0.14 \mu\text{m}$ in 5 DIV neurons.

Hatsuda et al, 2023 *Development* reported that mitochondria had an average length of 2 μm in dendrites (1.2 μm in axon) in DIV5 neurons transfected with GFP and Mito-DsRed.

Cho et al., 2017 *Nat Comm* reported that mitochondria were 1 μm in length (cortical neurons, sh mock transfected) – a value that is considerably closer to the values we report and closer to values obtained by electron microscopic analyses (0.4-0.6 μm) and also closer to values reported in other cells (e.g. see Triolo et al., 2023 STAR Protocols/Cell: light microscopy based on staining of endogenous Tom20; mitochondrial length ranges between $\sim 0.1\text{--}1.0 \mu\text{m}$, with an average of 0.4–0.75 μm .”).

The absolute numbers reported in the literature thus differ significantly. Probably this is due to variances among the cell systems studied and due to the different methods employed. The absolute numbers reported for “mitochondrial length” in our manuscript were obtained by Apotome microscopy (a structured illumination method providing 3D information) and were about 0.45 μm in average for NIH3T3 cells and neurons, when analyzed via anti-cytochrome c immunostaining. These data are about in the same range as our quantitative EM data and e.g. the mitochondrial length average reported by Cho et al., 2017 (see above literature review).

As the reviewer pointed out, it is very well possible that immunostaining of the endogenous cytochrome c may lead to a slight, general underestimation of mitochondrial sizes because of slightly heterogeneous stainings, which might lead to an incomplete filling of mitochondrial volumes and therefore to smaller length measurements and/or even to separation of one unevenly filled mitochondrion into two distinct objects during the image processing and quantitation. However, This putative technical bias will of course apply to all samples. Therefore, it is very unlikely that this will affect the relation of our data for syndapin I KO, syndapin II RNAi, and RTKN2 RNAi to their respective controls. The same applies to the syndapin I overexpression effects.

While, in light of this argumentation, the reviewer will acknowledge that the observed phenotypes are valid (they also are supported by respective rescue experiments), we nevertheless additionally addressed the technical concerns of the reviewer experimentally:

The revised Ext. Data Fig. 1-2 (former Ext. Data Fig. 1-1) now includes a colocalization study for anti-Cytc-stained and mitoGFP-filled mitochondria. As expected, the anti-Cytc immunolabeling overlapped very well with the mitoGFP signals (Manders coefficients, both ~ 0.75) (Ext. Data Fig. 1-2i). Whether mitochondria are tracked by immunostaining of endogenous cytochrome c or by overexpression of mitoGFP thus is a much smaller concern than anticipated.

In order to not just show an example image but to also reproduce key functional data the revised Extended Data Fig. 1-2 now additionally includes quantitative evaluations of the syndapin II loss-of-function data in NIH3T3 cells using mitoGFP and the skeleton tool of ImageJ, which is widely used in the literature to “skeletonize” and measure mitochondria (e.g. see Chang, Reynolds, *Neuroscience* 2006; Cho et al., *Nat Comm* 2017). The additionally conducted experimentation confirmed that syndapin II RNAi leads to enlarged mitochondria and that this effect was rescued by reexpression of RNAi-insensitive syndapin II-l (Ext. Data Fig. 1-2j-m).

Interestingly, in absolute data, these experiments showed a slight shift to higher values when compared to our cytochrome c-based evaluations. The mitochondrial length measured in controls was 0.7 μm (**Ext. Data Fig. 1-2g vs. m**).

The use of the skeleton tool often led to quite elaborate mitochondrial structures (**Ext. Data Fig. 1-2; Ext. Data Fig. 3-2**). Since variances in the averages of mitochondrial length measurements may not solely stem from the use of ectopically overexpressed mitochondrial fillers versus endogenous immunostainings but may also result from different methods of evaluation, we also followed this up. **Comparing examinations by a scientist with skeleton tool-based evaluations of the same anti-Cytc-traced mitochondria supported the observation that mitoGFP/skeleton-based data** often yielded higher averages (**Ext. Data Fig. 3-2d vs. e**). Apart from the slight differences in absolute numbers, however, the respective rhotekin2 loss-of-function phenotype was consistently and reliably revealed in both types of examination (**Ext. Data Fig. 3-2d vs. e**). The use of software tools and their limitations thus also contributes to the different absolute values for mitochondrial length determinations found in the literature.

Importantly, however, it has to be stressed that despite some general shifts in absolute values, all of these different methods and tools fully consistently revealed the same cell biological effects: An increase in mitochondrial length upon syndapin I KO in neurons (**Fig. 1a-e** using EM, morphological recognition of mitochondrial membranes and manual mitochondrial length determinations by ImageJ) or upon syndapin II RNAi in NIH3T3 cells (**Ext. Data Fig. 1-2a-g** using Apotome, anti-Cytc and manual mitochondrial length determinations by ImageJ; **Ext. Data Fig. 1-2j-m** using Apotome, mitoGFP and the skeleton tool of ImageJ) as well as a corresponding decrease in mitochondrial length upon syndapin I gain-of-function (**Ext. Data Fig. 1-2n-p** using Apotome, anti-Cytc and manual mitochondrial length determinations; **Ext. Data Fig. 1-2q-s** using Apotome, mitoGFP and the skeleton tool).

Similar to the above described examinations of syndapin functions, we also corroborated the functional rhotekin2 data by new experiments using mitoGFP and the skeleton tool for analyses of mitochondrial length (Ext. Data Fig. 3-1; Ext. Data Fig. 3-2; Ext. Data Fig. 3-3):

We confirmed the identified rhotekin2 gain-of-function phenotype in NIH3T3 cells with new determinations of mitochondrial length using mitoGFP and the skeleton tool. Again, the data obtained with mitoGFP were fully consistent with the anti-cytochrome c data shown previously. Also by the alternative method, rhotekin2 gain-of-function caused a decrease in mitochondrial length by about a quarter of the control values (**Ext. Data Fig. 3-3c vs. 3-3f**).

We furthermore supplemented our anti-Cytc-based rhotekin2 loss-of-function data in neurons as well as in NIH3T3 cells by new experiments using mitoGFP-tracked mitochondria and the skeleton tool for analyses (Ext. Data Fig. 3-1; Ext. Data Fig. 3-2g-j). The use of this alternative method nicely corroborated the RTKN2 loss-of-function phenotype we report in **Fig. 3e** and **Ext. Data Fig. 3-2a-e** using immunolabeling of the endogenous mitochondrial marker anti-Cytc for tracing mitochondria. With both methods, a statistically highly significant increase of mitochondrial length upon rhotekin2 RNAi was observed. Thus, both anti-Cytc-based quantitative evaluations and mitoGFP overexpression-based quantitative evaluations reveal an identical rhotekin2 loss-of-function phenotype – a significant increase in mitochondrial length.

The newly added experimentation furthermore confirmed that the syndapin binding-deficient rhotekin2 mutant ΔKRAP failed to rescue the rhotekin2 loss-of-function phenotype. In contrast, reexpression of an RNAi-insensitive but otherwise unaltered rhotekin2 fully rescued the mitochondrial length phenotype.

Also in this case, the new experimentation thus led to exactly the same conclusions as the experiments using anti-Cytc immunolabeling to visualize mitochondria (**Fig. 3e vs. Ext. Data Fig. 3-1a-e**).

The concern of the reviewer that fixation may result in changes of mean mitochondrial length was addressed by additional analyses of mitochondrial sizes in single frames of live recordings of MitoTracker-stained neurons (Ext. Data Fig. 3-1f). Comparisons with data derived from images of fixed neurons clearly showed that the results were similar (**Ext. Data Fig. 3-1e vs. Fig. 3-1f**). The mitochondria were thus not altered in length upon fixation. The rhotekin2 loss-of-function phenotype of an increased mitochondrial length shown by two independent methods in fixed cells (compare **Fig. 3e** and **Ext. Data Fig. 3-1e**) was also clearly demonstrated again in this alternative, newly added experimentation using live imaging (**; $P=0.0018$) (**Ext. Data Fig. 3-1f**).

2. In the loss of Rhotekin2 experiments, how can an effect on branch initiation independent of mitochondrial localization be ruled out? I think for the authors to make the claims about Rhotekin2's role in mitochondrial positioning regulating branching some sort of sufficiency evidence is necessary.

- a. Does overexpression of Rhotekin2 result in increased localization of mitochondria to branch sites and result in increased branching?
- b. How does loss of rhotekin2 affect branching dynamics? Is it branching isn't initialized or does the branch form but isn't stabilized?
- c. Ruling out changes in actin structure at branch points following Rhotekin2 loss seems important.

In our eyes, it was somewhat questionable whether a positioning of mitochondria at branch initiation sites that already is effectively secured by rhotekin2 at the mitochondrial surface and syndapin I at the plasma membrane could indeed be promoted further by overexpression of rhotekin2 (even assuming that enough mitochondrial docking places may be available) and thereby result in enhanced dendritic branching.

We nevertheless conducted experiments proposed by the reviewer and addressed putative effects of rhotekin2 overexpression on dendritic branching. As expected, an excess of rhotekin2 did not significantly promote dendritic branching further (Reviewer Figure 1).

Reviewer Figure 1

Reviewer Figure 1: An excess of rhotekin2 has no recognizable effects on the dendritic arbor of primary hippocampal neurons. Rat primary hippocampal neurons were transfected at DIV4 with either RTKN2 overexpression or control plasmids, fixed at DIV6 and analyzed for the indicated dendritic parameters based on anti-MAP2 immunostainings using IMARIS software. Note that rhotekin2 overexpression does not lead to any obvious effects in dendritic arbor formation despite reasonable cell numbers analyzed ($n=26/28$ neurons from 2 independent neuronal preparations). Data, mean \pm SEM visualized as Bar/dot plots (a-c) and bar plots (d), respectively. Statistical analyses, Mann-Whitney (a-c) and 2-way ANOVA/Bonferroni (d) (all n.s.).

In addition, we would like to refer the reviewer to the fact that new experimentation we added to the revised manuscript now directly demonstrates that also the rhotekin2 interaction partner we identified, syndapin I, which is a critical component in dendritic branch induction (Schwintzer et al., 2011 *EMBO J.*; Koch et al., 2020 *Cereb. Cortex*; Izadi et al., 2021 *eLIFE*; Izadi et al., 2023 *J. Cell Biol.*), plays a crucial role in proper mitochondrial positioning at branch initiation sites (newly added Fig. 7 of the revised manuscript). Thus, the syndapin I loss-of-function data very well phenocopy rhotekin2 loss-of-function phenotype in mitochondrial positioning.

These new data also are very well in line with the finding that two rhotekin2 loss-of-function-associated phenotypes we identified, i.e. the increase in mitochondrial length and also the defects in dendritic branching, were strictly dependent on the identified syndapin I binding site in rhotekin2, the KRAP motif, in the rescue experiments we conducted in the analyses of both rhotekin2 loss-of-function phenotypes (Fig. 3a-e,k and l; newly added Ext. Data Fig. 3-1; Fig. 8 of the revised manuscript).

Additionally, we conducted experiments suggested by reviewer 2 that are somewhat along the lines of the questions raised by reviewer 1. In the months of our revision work, we studied whether dendritic branch formation brought about by syndapin I overexpression would be dependent

on rhotekin2 (see Reviewer Figure 4 (addressing a question of Reviewer 2 – therefore see there)).

Also in these cotransfection experiments, we were able to observe the syndapin I overexpression phenotype. Similarly, the experiments also showed the rhotekin2 loss-of-function phenotype of defective dendritic arborization again. **Syndapin I-overexpressing but rhotekin2-deficient neurons, in contrast, showed dendritic arborization parameters similar to control cells.** Considering that syndapin I as F-BAR domain protein has i) the ability to shape membranes by itself and has ii) the means to promote F-actin formation by recruiting several components of the actin cytoskeleton promoting neuronal cell shape development, an only partial dependence of syndapin I functions in dendritic arbor formation on the mitochondria anchoring by rhotekin2 appears very well understandable. **In case the reviewers or the editors feel that - instead of reporting these data as a reviewer figure - these data shall rather be included into our manuscript, we shall be happy to show them as yet another Extended Data Figure.**

In our revision work, we furthermore answered the important question of the reviewer whether rhotekin2 deficiency impairs dendritic branch induction or rather has some detrimental effects on dendritic branch stabilization (newly added Fig. 8j). As clearly shown by quantitative 3D-live experiments with developing neurons, deficiency of the mitochondrial docking protein rhotekin2 led to a massive defect in the initiation of protrusions from the dendritic arbor. The rate of dendritic branch induction in rhotekin2-deficient neurons was only about half as high as in control neurons (**newly added Fig. 8j of the revised manuscript**).

As far as actin cytoskeletal contributions to the observed mitochondrial positioning and tethering are concerned, we fail to understand why putative changes in F-actin structures at branch points need to be ruled out. It seems very well possible that the dynamic F-actin, which arises at dendritic branch initiation sites (Hou et al., 2015), may be influenced by defects in calcium homeostasis and/or by defects in local energy supply caused by impaired mitochondrial anchoring at these sites upon rhotekin2 deficiency and that this contributes to the defects in dendritic branch induction observed upon rhotekin2 deficiency (**please see discussion of the revised manuscript**).

3. The first half of the manuscript and the second half of the manuscript are never linked together. In the first half, a story is presented that RTKN2/Syndpin regulate mitochondrial size changes and RTKN2 controls this, while in the second half the story switches to RTKN2/Syndpin control localization to sites of branching and regulate branching. a. Are these two parts related or independent? If independent and not related to the branching, it seems like this shouldn't be a part of the narrative.

We thank the reviewer for making us aware of that we should point our clearer how different functional aspects of the manuscript are related. **We have improved this by changes to the manuscript and to the figures as well as by addition of new experimentation (see below).**

The model (Fig. 9 in the revised manuscript) and the revised discussion now point out clearer how the different aspects are related. Rhotekin2, which we identified as a mitochondrial protein, obviously acts as an important mitochondrial anchor protein for syndapins, which as dimers/multimers then in turn have the ability to connect mitochondria/rhotekin to further SH3 domain interaction partners, such as dynamin 2, which was demonstrated to play a role in mitochondrial fission (Lee et al., 2016 *Nature*; **please also see our answer to point #9 of the reviewer and the reviewer figure 2, which shows that syndapin I is unable to interact with Drp1 but does interact with dynamin 2**) and/or to furthermore tether mitochondria to special membrane domains, whose membrane curvatures are either preferred by the syndapin I F-BAR domain or which are actively shaped by syndapin I (such as dendritic branch induction sites).

While the reviewer will acknowledge that the syndapin and rhotekin2 functions studied in the manuscript are in general well-linked by biochemical evidence, by comparisons of both gain- and loss-of-function phenotypes and by the use of syndapin-binding-defective rhotekin2 mutants, which were unable to rescue the rhotekin2 deficiency phenotypes in both mitochondrial morphology and dendritic arbor formation, the reviewer is right that this was not the case for the mitochondrial tethering at dendritic branch induction sites. Here, the focus was on rhotekin2 alone. Only subsequently, for dendritic arbor formation, related syndapin I functions and the need of complex formation with syndapin I via the syndapin-binding KRAP motif of rhotekin2 then came into focus again. **As this missing relationship between rhotekin2 and syndapin I at the level of mitochondrial tethering could indeed be viewed as a caveat of our work, we addressed this important point experimentally during our revision work (please see newly added Fig. 7 of the revised manuscript).**

The ability of syndapin I to associate with the plasma membrane and the finding that it also can associate with rhotekin2-decorated mitochondria, as depicted in the model, raised the hypothesis that, similar to rhotekin2 deficiency, syndapin I deficiency may also impair mitochondrial positioning. **Strikingly, syndapin I RNAi indeed led to defects in mitochondrial positioning in the central zone of branch initiation sites (newly added Fig. 7 of the revised manuscript) – and thereby indeed phenocopied a cell biological impairment identified for rhotekin2 RNAi (compare Fig. 5).** The syndapin I deficiency-mediated defects in mitochondrial positioning were statistically significant at all three time points studied, i.e. 60 s before, at (t=0 s) and 60 s after dendritic branch induction (**newly added Fig. 7 of the revised manuscript**).

In line with the other data presented in the manuscript, these results underscore that rhotekin2 and syndapin I work together in a closely related manner.

4. It is claimed for the localization part of the story that calcium plays some role in this, (see model at the end of figure 7), but no data is presented to support this in the manuscript.

- a. How? Are calcium levels different at branch points? Are calcium dynamics different?
- b. Perhaps addressing this may allow for the linkage of the mitochondrial size and tethering phenotypes (Hatsuda et al, Development 2023)?

We have demonstrated before that some short-lived Ca^{2+} transients also occur in dendritic arbors of developing neurons and that these Ca^{2+} signals seem to precede dendritic branch induction, whereas right during branch induction, Ca^{2+} levels usually again seem to be very low (Hou et al., 2015 *PLoS Biol.*).

In general, Ca^{2+} homeostasis and energy supply are general and well-established functions of mitochondria. Therefore, Ca^{2+} was mentioned in the model. Also, Ca^{2+} promotes the assembly of syndapin I/Cobl and Cobl-like complexes, which accumulate at nascent branch sites prior to dendritic branch induction (Hou et al., 2015 *PLoS Biol.*; Izadi et al., 2021 *eLIFE*).

Since the reviewer seems to think that indicating Ca^{2+} homeostasis as mitochondrial function in the model may be confusing in the context of this manuscript, we left out the Ca^{2+} symbol in the revised model (please see Fig. 9 of the revised manuscript).

Other comments

1. All the bar graphs should be presented with spread of the data represented (box plots, individual points or violin plots)

According to general editorial guidelines, we have replaced all bar plots of experiments with lower n-numbers by graphs that also display all individual data points. We set the threshold for this at $n < 44$.

Therefore, the quantitative data panels of Ext. Data Fig. 1-2h, Ext. Data Fig. 1-4e, Fig. 3i and j, Ext. Data Fig. 3-2f, Ext. Data Fig. 3-3d, Ext. Data Fig. 3-4f,k and p, Fig. 4a, Fig. 5a, as well as Fig. 6m and o have been replaced in the revised manuscript. For Fig. 8 (former Fig. 7) we raised the n numbers by additional experimentation during our revision work and obtained n numbers well above the bar/dot plot threshold, as the Sholl analyses and the dendritic branch depth analyses with their multiple conditions are too complex graphs to be shown as bar/dot plots (**revised Fig. 8**). This also further increased the statistical significances (now all **, *** or **** in the **revised Fig. 8**). Frequency distributions (**Ext. Data Fig. 3-3e**) have no scattering data points.

Of the newly added experimentation, **Ext. Data Fig. 3-5a and c, and Fig. 7a and h-j** are shown as bar/dot plots, while all data with high n-numbers is shown as bar plots.

We hope that the reviewer will agree with us that, in experimentation with high n-numbers, the scattering of some individual data points is far less relevant and that it is instead scientifically much more valuable to be able to clearly visualize the cell biological consequences of rhotekin2 and syndapin gain- and loss-of-function in comparison to controls instead of trying to cover e.g. thousands of data points individually, which in fact would just obscure and distract from the more informative mean data. In case the reviewer or the editors for some reasons would nevertheless like to see even more panels being exchanged for variants covering individual data points, we shall be happy to do so.

2. It is hard to visualize the stated results with the images presented in figure 1

We acknowledge that the EM pictures in Figure 1a and b indeed are small, do not explicitly visualize the evaluations of mitochondria conducted and also unfortunately were of relatively low resolution. **The criticized six images in Fig. 1a and b have been replaced by higher resolution versions in the revised manuscript (revised Fig. 1).**

The newly inserted Extended Data Fig. 1-1 now additionally displays enlarged EM pictures corresponding to the panels shown in the main figure and also displays the ImageJ markings for mitochondrial area and length determinations.

3. Validation of anti-Rhotekin2 antibody should be presented in the manuscript somewhere. For instance, does staining decrease or go away when RNAi is used?

For validation of the rhotekin2 antibody, we would like to refer the reviewer to the set of different experiments shown in **Ext. Data Fig. 1-4 (corresponding to former Ext. Data Fig. 1-3 and 3-1 a and b and including new additional data)**. This figure validates the anti-rhotekin2 antibody by different experimentation (Western blots of overexpressed GFP-rhotekin2¹⁴⁵⁷⁻⁶⁰¹ and corresponding negative control (**Ext. Data Fig. 1-4a**), immunofluorescence of overexpressed GFP-rhotekin2¹⁴⁵⁷⁻⁶⁰¹ and corresponding negative control (**Ext. Data Fig. 1-4b**) and immunofluorescence of endogenous rhotekin2 with corresponding non-immune IgG control (**Ext. Data Fig. 1-4c**).

Finally, the anti-rhotekin2 antibody is also validated by immunoblotting of wild-type rhotekin2 associating with syndapin I, II and III, while the corresponding Δ KRAP mutant of rhotekin2 not binding to syndapins was specifically detected in the supernatants (**Fig. 1k**; different sizes and distinct binding behavior clearly prove rhotekin2 immunodetection).

In our revision work, we additionally conducted further validations of the anti-rhotekin2 antibody by immunofluorescence validation experiments, as the reviewer suggested. Primary hippocampal neurons in culture were either transfected with RTKN2 RNAi or with scrambled RNAi. Images recorded under identical settings show a reduced anti-RTKN2 signal upon RTKN2 knock-down (**newly added Ext. Data Fig. 1-4f**).

As far as the mitochondrial localization of rhotekin2 in general is concerned, we would like to point out that this finding is backed-up by biochemical and visual examinations demonstrating that rhotekin2 is enriched in mitochondrial preparations by Western blotting and immunofluorescence analyses and that rhotekin2 is associated with the outer mitochondrial membrane (**Fig. 2d-i**).

4. The text mentions overexpressed Rhotekin2 in neurons but no data is provided in the figures

An overexpression of rhotekin2 in neurons is not mentioned in the text (verified by search function in Word). The reviewer must have confused this with some other data.

We conducted overexpression analyses in NIH3T3 cells (examinations of mitochondrial length) but not in neurons. Besides overexpressing rhotekin2 in NIH3T3 cells (**Extended Data 3-3**), we also conducted a syndapin II overexpression study in NIH3T3 cells (**Extended Data 1-2**).

We meanwhile did such analyses during our revision work although we felt that it was somewhat questionable whether a positioning of mitochondria at branch initiation sites that already is effectively secured by rhotekin2 at the mitochondrial surface and syndapin I at the plasma membrane could indeed be promoted further by overexpression of rhotekin2 (even assuming that enough mitochondrial docking places may be available) leading to enhanced dendritic branching. As expected, an excess of rhotekin2 did not significantly promote dendritic branching further (**see our response to major point #2 above and corresponding reviewer figure**).

5. Figure 2c panels have no labeling for what they are

We apologize for this embarrassing mistake of ours and thank the reviewer for carefully going through our work and noticing this. For some reasons, the anti-rhotekin2 and the anti-Cytc description of the immunolabelings shown disappeared to a lower layer of the multilayered figure file.

We have corrected this mistake in the revised version of Fig. 2. We also specified that the additional blue immunolabeling in the merge is anti-MAP2.

6. Validation should be provided for results in Figure 2d-e. How can it be claimed as a mitochondrial protein if no validation of purity is shown? At least markers for nucleus, cytosol and ER should be presented

The figure legend of the revised manuscript now points out clearer that the data in Fig. 2d clearly show the enormous enrichment of mitochondria by the purification (using a published/cited standard method), as shown by immunoblotting of the mitochondrial marker COX IV. Similar to the mitochondrial marker COX IV, anti-rhotekin2 immunosignals also rise from lysate to crude mitochondria and then again to purified mitos. Rhotekin2 thus clearly is a mitochondrial protein, not a contamination of the preparation.

In addition, the revised version of Fig. 2 now includes immunoblottings of further additional components, which represent other cellular compartments, in both the non-neuronal analysis (revised Figure 2d) and the brain analysis (revised Fig. 2e). Shown are insulin receptor tyrosine kinase (IRTK) representing the plasma membrane, GM130 representing the Golgi apparatus, laminB1 representing the nucleus (**revised Fig. 2e only**) and GAPDH representing soluble, cytosolic proteins. Importantly, in contrast to rhotekin2 and the mitochondrial marker COX IV, none of these newly added immunoblotting analyses showed any enrichments of these non-mitochondrial proteins in the mitochondrial fractions. Instead, these proteins were merely visible as faint bands or completely

absent from the mitochondrial preparations (**please see the data added to the revised Fig. 2d and e**).

We hope the reviewer will agree with us that the additionally added new experimentation nicely confirms the specificity of the mitochondrial preparation and the rhotekin2 enrichment in the mitochondrial fractions.

7. For figure 2f, is this axonal or dendritic? Perhaps a slightly expanded view could also be presented for context

We thank the reviewer for noting this missing information. Fig. 2f shows a very high magnification of a single, dendritic mitochondrion immunostained with anti-Cytc and anti-rhotekin2 to visualize that the anti-rhotekin2 immunosignal surrounds the immunostaining of Cytc, which in contrast to rhotekin2 is an internal mitochondrial protein. **This information is now included in the revised manuscript and also directly in the revised Fig. 2.**

Furthermore, the revised version of Fig. 2 now includes a second example of a high-magnification view of a mitochondrion in a dendrite segment thin enough to be delineated (by a dashed line) in the high-magnification images shown in the revised Fig. 2f.

8. Figure 3 images with high mag insets don't seem to agree with the quantification data. C looks more fused than B

The criticized exemplary images for the quantitative data have been replaced (revised Fig. 3b,c).

9. Rhotekin2 is suggested to affect fission/fusion, this could be assessed with experiments using photoactivatable-GFP (see Karbowski et al, J Cell Biol 2004)

We appreciate the suggestion of the reviewer, however, we think that expanding this part of the study would even more develop seemingly disconnected aspects in one study by shifting the focus away from the revelation of rhotekin2 functions at dendritic branch initiation sites. We suspect that promoting smaller mitochondria represents a syndapin (dimerization)-linked function (see **Fig. 1 and Ext. Data Fig. 1-2**) that will use rhotekin2 as mitochondrial anchor and will employ dynamin2 as effector of fission. The interaction of syndapin I with dynamin II is published (Koch et al. 2011 *EMBO J.*) and also the role of dynamin2 in mitochondrial dynamics is established (Lee et al., 2016 *Nature*). **Since, however, also readers of our study may have similar questions as the reviewer, we covered the side-aspect of syndapin/dynamin2 interactions and dynamin2's role in membrane fission side aspect in the discussion of the revised manuscript.**

In case the reviewer feels strongly about this line of additional research, which somewhat deviates from the main line of the study, and prefers that we should clearly experimentally dismiss a syndapin I interaction with the fission main component dynamin-related protein 1 (Detmer and Chan, 2007 *Nat. Rev. Cell Biol.*) and experimentally show the dynamin2 interaction with syndapin I again instead of just citing the relevant literature and discussing this as plausible mechanism, **we shall be happy to include such experimentation into the manuscript (see following Reviewer Figure).**

Reviewer Figure 2. Syndapin I fails to associate with the mitochondrial fission component Drp1 but instead binds to dynamin 2 – a component that can power mitochondrial fission, too.

a-c) Immunoblotting analyses of coprecipitation studies with immobilized GST-Sdpl and GST, respectively, and with lysates of HEK293 cells expressing epitope-tagged versions of dynamin-related protein 1 (Drp1; **a**), dynamin1 (Dyn1; **b**) and dynamin2 (Dyn2; **c**). Note that dynamin 2, which – similar to Drp1 - also is capable to power mitochondrial fission (Lee et al, 2016 *Nature*) associates with syndapin I as well as with dynamin 1 - the originally identified syndapin I interaction partner (Qualmann et al., 1999 *Mol. Biol. Cell*; Qualmann et al., 2000 *J. Cell Biol.*).

10. For mitochondrial motility data, either supplemental videos or kymographs should be included for assessment of motility

The overall motility of mitochondria in both control and rhotekin2-deficient neurons was unchanged, as shown before (Figure 3m). Also, the newly added live experimentation data based on both MitoTracker staining and mitoGFP overexpression did not show any overall differences (Ext. Data Fig. 3-5). We therefore focused on the mitochondrial positioning to comply with the reviewer's request to additionally show videos (see below; 11.)

11. Supplemental videos for figures 4 & 5 data would really add to the impact of the work

Supplementary videos have been added to the previously shown analyses of scr. RNAi and rhotekin2 RNAi in Figure 4 and Figure 5 (see Extended Data Videos 4-1 and Extended Data Video 5-1). Although it is difficult to document and represent the careful frame-to-frame analyses of mitochondrial positioning at branch induction sites, we hope that the reviewer and the readers will appreciate the mitochondria that are usually placed and tethered (trajectories are shown as well) in the central area at branch initiation sites, whereas such a positioning before and during dendritic branch initiation in rhotekin2-deficient neurons is impaired (as quantitatively shown in Figure 4 and 5).

Along the lines of the reviewer's request concerning the rhotekin2 RNAi and control data previously focused on, we also added further video examples to the newly added experimentation demonstrating the additional requirement of the rhotekin2 interaction partner syndapin I in mitochondrial positioning at dendritic branch initiation sites (Extended Data Videos 7-1 and 7-2).

12. The text and figure 6 don't seem to be in sync. Text refers to Fig6p but figure 6 has no panel p. Also the text description of figure 6o doesn't seem to line up with the data shown

We very much thank the reviewer for noticing this. **The text of the revised manuscript was corrected accordingly.**

13. For figure 7, it would be useful to include a panel with just the reporter to see the dendritic compartment without all the other overlapping information.

Immunostaining of the dendritic marker MAP2 defines the dendritic compartment, while GFP and mCherry used as reporters for the RNAi and rescue plasmids reliably mark the transfected neurons. In principle, GFP and mCherry as cell fillers of course also track cell morphologies, yet, these fluorescent molecules do of course not discriminate between axonal and dendritic compartments. We thank the reviewer for pointing out that the images in the main figure (former Fig. 7, now Fig. 8) may be overloaded by too much image information to appreciated the dendritic compartments of the transfected neurons. Although the pictures shown in Fig. 8 mostly serve as illustrative examples of the quantitative analyses, **an new Extended Data Figure now shows the anti-MAP2 immunosignals and the mCherry fluorescence (reporter/cell filler) as separate data panels and as a merge (please see newly added Ext. Data Fig. 8-1).**

Reviewer #2 (Remarks to the Author):

The manuscript by Tröger and colleagues builds on previous work from their group and others, to further investigate the molecular mechanisms of branch formation in developing neurons. Following the authors' previous findings demonstrating syndapin function in dendritic branching, they identified a binding partner for syndapin which recruits mitochondria to nascent branching points in the growing dendrite. The authors provide series of biochemistry data that strongly support mitochondrial localization of rhotekin2 and its specific binding to syndapin I. Then they indicate that rhotekin2 regulates mitochondrial shape and positioning in dendrites and induces new branch formation by morphological analyses of fixed and live neurons.

If proven true with sufficient evidence, this study would provide important additional information about the function of mitochondria in dendrite formation. Unfortunately, however, the methods for mitochondrial dynamics analyses, which should be the cornerstone, are not adequate, and many data do not convince readers nor provide sufficient data to support the conclusions. I believe it is necessary to redo most microscopic analyses using appropriate molecular tools and microscope setups before the work can be reconsidered.

Major comments

1. Most of the histological analyses of mitochondria were done using cytochrome C as a mitochondrial shape marker. However, cytochrome C is unevenly distributed in the intermembrane space of mitochondria in a punctate manner, making it unsuitable for mitochondrial shape analysis. This is evident from the fragmented localization in the figures 2 and 3, which makes it difficult to distinguish individual mitochondria that are densely overlapped within dendrites. The measured values of the mean length in Fig. 3e are therefore much smaller than those in previous studies, raising concerns of their accuracy (Li et al., *Cell* 2004; Cho et al., *Nat Comm* 2017; Hatsuda et al., *Development* 2023). Shape analysis relies on consistent and homogeneous distribution of the marker proteins for accurate measurements. GFP or its derivatives with a mitochondrial localization sequence (mitoGFP) is widely used for this purpose.

We acknowledge the concern of the reviewer that some of the data in the literature was obtained by use of ectopically overexpressed mitochondrial fillers (e.g. mitoGFP), while we used endogenous anti-Cytc signals to highlight mitochondria and that this difference in method may lead to different values of mitochondrial sizes in our comparisons of conditions when compared to absolute values reported in the literature.

As the reviewer suggested, we carefully reviewed the literature concerning mitochondrial sizes. We also addressed these concerns by different sets of experimentation included in the revised manuscript (see below).

Literature: The literature cited by the reviewer used standard (confocal) light microscopy and described that mitochondria in hippocampal neurons were globular or tubular in shape and highly variable in size:

Li et al., 2004 *Cell* (confocal analyses of mitochondria visualized with overexpressed MitoDsRed) reported the average to be 2.3 μm . However, it has to be noted that this average was affected by a small proportion of dendritic mitochondria that appeared extremely long ($>10 \mu\text{m}$, 2.6% of 230 mitochondria). It remained unclear whether these indeed represented single mitochondria or trains of several individual mitochondria that were not resolved by standard light microscopy.

Chang & Reynolds, 2006 *Neuroscience* used overexpressed mito-eYFP and reported mitochondrial length (calculated from skeletonized fluorescent objects) to be $2.01 \pm 0.14 \mu\text{m}$ in 5 DIV neurons.

Hatsuda et al, 2023 *Development* reported that mitochondria had an average length of 2 μm in dendrites (1.2 μm in axon) in DIV5 neurons transfected with GFP and Mito-DsRed.

Cho et al., 2017 *Nat Comm* reported that mitochondria were 1 μm in length (cortical neurons, sh mock transfected) – a value that is considerably closer to the values we report and closer to values obtained by electron microscopical analyses (0.4-0.6 μm) and also closer to values reported in other systems (e.g. see Triolo et al., 2023 STAR Protocols/Cell: light microscopy based on staining of endogenous Tom20; mitochondrial length ranges between \sim 0.1–1.0 μm , with an average of 0.4–0.75 μm .“!).

The absolute numbers reported in the literature thus differ significantly. Probably this is due to variances among the cell systems studied and due to the different methods employed. The absolute numbers reported for “mitochondrial length” in our manuscript were obtained by Apotome microscopy (a structured illumination method providing 3D information) and were about 0.45 μm in average for NIH3T3 cells and neurons, when analyzed via anti-cytochrome c immunostaining. These data are about in the same range as our quantitative EM data and e.g. the mitochondrial length average reported by Cho et al., 2017 (see above literature review).

As the reviewer pointed out, it is very well possible that immunostaining of the endogenous cytochrome c may lead to a slight, general underestimation of mitochondrial sizes because of slightly heterogeneous stainings, which might lead to an incomplete filling of mitochondrial volumes and therefore to smaller length measurements and/or even to separation of one unevenly filled mitochondrion into two distinct objects during the image processing and quantitation. However, this putative technical bias will of course apply to all samples. Therefore, it is very unlikely that this will affect the relation of our data for syndapin I KO, syndapin II RNAi, and RTKN2 RNAi to their respective controls. The same applies to the syndapin I overexpression effects.

While, in light of this argumentation, the reviewer will acknowledge that the observed phenotypes are valid (they also are supported by respective rescue experiments), we nevertheless additionally addressed the technical concerns of the reviewer experimentally:

The revised Ext. Data Fig. 1-2 (former Ext. Data Fig. 1-1) now includes a colocalization study for anti-Cytc-stained and mitoGFP-filled mitochondria. As expected, the anti-Cytc immunolabeling overlapped very well with the mitoGFP signals (Manders coefficients, both \sim 0.75) (Ext. Data Fig. 1-2i). Whether mitochondria are tracked by immunostaining of endogenous cytochrome c or by overexpression of mitoGFP thus is a much smaller concern than anticipated.

In order to not just show an example image but to also reproduce key functional data, the revised Extended Data Fig. 1-2 now additionally includes quantitative evaluations of the syndapin II loss-of-function data in NIH3T3 cells using mitoGFP and the skeleton tool of ImageJ, which is widely used in the literature to “skeletonize” and measure mitochondria (e.g. see Chang, Reynolds, *Neuroscience* 2006; Cho et al., *Nat Comm* 2017). The additionally conducted experimentation confirmed that syndapin II RNAi leads to enlarged mitochondria and that this effect was rescued by reexpression of RNAi-insensitive syndapin II-I (Ext. Data Fig. 1-2j-m).

Interestingly, in absolute data, these experiments showed a slight shift to higher values when compared to our cytochrome c-based evaluations. The mitochondrial length measured in controls was 0.7 μm (Ext. Data Fig. 1-2g vs. m).

The use of the skeleton tool often led to quite elaborate mitochondrial structures (Ext. Data Fig. 1-2; Ext. Data Fig. 3-2). Since variances in the averages of mitochondrial length measurements may not solely stem from the use of ectopically overexpressed mitochondrial fillers versus endogenous immunostainings but may also result from different methods of evaluation, we also followed this up.

Comparing examinations by a scientist with skeleton tool-based evaluations of the same anti-Cytc-traced mitochondria supported the observation that mitoGFP/skeleton-based data often yielded higher averages (Ext. Data Fig. 3-2d vs. e). Apart from the slight differences in absolute numbers, however, the respective rhotekin2 loss-of-function phenotype was consistently and reliably revealed in both types of examination (Ext. Data Fig. 3-2d vs. e). The use of software tools and their limitations thus also contributes to the different absolute values for mitochondrial length determinations found in the literature.

Importantly, however, it has to be stressed that despite some general shifts in absolute values, all of these different methods and tools fully consistently revealed the same cell biological effects: An increase in mitochondrial length upon syndapin I KO in neurons (**Fig. 1a-e** using EM, morphological recognition of mitochondrial membranes and manual mitochondrial length determinations by ImageJ) or upon syndapin II RNAi in NIH3T3 cells (**Ext. Data Fig. 1-2a-g** using Apotome, anti-Cytc and manual mitochondrial length determinations by ImageJ; **Ext. Data Fig. 1-2j-m** using Apotome, mitoGFP and the skeleton tool of ImageJ) as well as a corresponding decrease in mitochondrial length upon syndapin I gain-of-function (**Ext. Data Fig. 1-2n-p** using Apotome, anti-Cytc and manual mitochondrial length determinations; **Ext. Data Fig. 1-2q-s** using Apotome, mitoGFP and the skeleton tool).

Similar to the above described examinations of syndapin functions, we also corroborated the functional rhotekin2 data by new experiments using mitoGFP and the skeleton tool for analyses of mitochondrial length (Ext. Data Fig. 3-1; Ext. Data Fig. 3-2; Ext. Data Fig. 3-3):

We confirmed the identified rhotekin2 gain-of-function phenotype in NIH3T3 cells with new determinations of mitochondrial length using mitoGFP and the skeleton tool. Again, the data obtained with mitoGFP were fully consistent with the anti-cytochrome c data shown previously. Also by the alternative method, rhotekin2 gain-of-function caused a decrease in mitochondrial length by about a quarter of the control values (**Ext. Data Fig. 3-3c vs. 3-3f**).

We furthermore supplemented our anti-Cytc-based rhotekin2 loss-of-function data in neurons as well as in NIH3T3 cells by new experiments using mitoGFP-tracked mitochondria and the skeleton tool for analyses (Ext. Data Fig. 3-1; Ext. Data Fig. 3-2g-j). The use of this alternative method nicely corroborated the rhotekin2 loss-of-function phenotype we report in **Fig. 3e** and **Ext. Data Fig. 3-2a-e** using immunolabeling of the endogenous mitochondrial marker anti-Cytc for tracing mitochondria. With both methods, a statistically highly significant increase of mitochondrial length upon RTKN2 RNAi was observed. Thus, both anti-Cytc-based quantitative evaluations and mitoGFP overexpression-based quantitative evaluations reveal an identical rhotekin2 loss-of-function phenotype – a significant increase in mitochondrial length.

The newly added experimentation furthermore confirmed that the syndapin binding-deficient rhotekin2 mutant Δ KRAP failed to rescue the rhotekin2 loss-of-function phenotype. In contrast, reexpression of an RNAi-insensitive but otherwise unaltered rhotekin2 fully rescued the mitochondrial length phenotype.

Also in this case, the new experimentation thus led to exactly the same conclusions as the experiments using anti-Cytc immunolabeling to visualize mitochondria (**Fig. 3e vs. Ext. Data Fig. 3-1a-e**).

As further proof, we also confirmed that the identified phenotype was not a fixation artifact.

In order to do so, mitochondrial sizes in single frames of live recordings of MitoTracker-stained neurons were quantitatively analyzed (**Ext. Data Fig. 3-1f**). Comparisons with data derived from images of fixed neurons clearly showed that the results were similar (**Ext. Data Fig. 3-1e vs. Fig. 3-1f**). The mitochondria were thus not altered in length upon fixation and the rhotekin2 loss-of-function phenotype of an increased mitochondrial length (already shown by two independent methods in fixed cells in the revised manuscript (compare **Fig. 3e** and **Ext. Data Fig. 3-1e**)) was also clearly demonstrated again in this third method of evaluation (live imaging: **, $P=0.0018$; **Ext. Data Fig. 3-1f**).

2. The velocity of mitochondrial transport in pyramidal neuron dendrites are ranged between 0.3-1 $\mu\text{m}/\text{sec}$ in previous studies (MacAskill et al., Neuron 2009; van Spronsen et al., Neuron 2013; Loss and Stephenson, Mol Cell Neurosci. 2017). In this study, the live imaging of dendritic mitochondria was conducted at 10 sec intervals, during which mitochondria are expected to move several times

their own length in a single frame, making accurate tracking difficult. The present study showed that even the maximum track speed of mitochondrial motility was about 0.06 $\mu\text{m}/\text{sec}$, an order of magnitude lower than previous studies. Images and videos should be shown for the readers to validate the analyses.

The reviewer refers to **Fig. 3m**. We thank the reviewer for pointing out that we obviously need to communicate clearer that we are not focusing on mitochondria undergoing active transport along microtubules. **As now pointed out clearer in the revised manuscript, we did not study the coupling or the action of motor proteins in this study but identified a mitochondrial tethering and positioning that correlated with dendritic branch initiation sites and relies on rhotekin2 and - as we now unravel upon our revision work - also on syndapin I.** Therefore, we focused on ALL mitochondria in **Fig. 3m**, while the studies cited by the reviewer were focusing on obviously motile mitochondria only and excluded all not so fast-moving mitochondria.

Literature review:

MacAskill et al., *Neuron*, 2009: Neurons at DIV9-14; only “motile” mitochondria were included (only ~20% were considered motile! All others were excluded); mean velocity of the few selected motile ones in dendrites $0.91 \pm 0.26 \mu\text{m}/\text{s}$.

Van Spronsen et al., *Neuron*, 2013: Hippocampal neurons transfected DIV9 (mito-dsRed), analyzed DIV13; imaged every 1s. Motile mitochondria: ~40% were considered, all others were excluded!; velocity: ~0.8 $\mu\text{m}/\text{s}$ in dendrites (however, not reported clearly whether only “motile” mitochondria were included in quantitation).

Loss O, Stephenson FA. *Developmental changes in trak-mediated mitochondrial transport in neurons*. *Mol Cell Neurosci*. 2017: Evaluated were exclusively moving mitochondria (in cortical axons, cortical dendrites, hippocampal axons and hippocampal dendrites); tracked by expression of the mitochondrial marker DsRed-Mito in young (6 DIV), maturing (10 DIV) and mature (14 DIV) neurons: only $27.0 \pm 1.1\%$ were considered motile at DIV6; velocity $0.43 \pm 0.03 \mu\text{m}/\text{s}$. Moving mitochondria were defined as those that are displaced $>2 \mu\text{m}$ from one site. This allowed the tracking of the movement of each moving mitochondria within the chosen sample together with the distance travelled, the direction (anterograde or retrograde) and the minimum, maximum and mean velocity (imaged with 3 s intervals).

During the last 20 years, mitochondrial transport studies have furthermore revealed that active transport speeds differ at bit with age (see also consistent data in Loss & Stephenson 2017 for DIV4,10 and 14).

Ligon LA & Steward O (2000): DIV7 hippocampal neurons with Mitotracker; in dendrites motile mitochondria 23-35%, their mean 0.4-0.5 $\mu\text{m}/\text{s}$; “non-moving mitos” 0.03 $\mu\text{m}/\text{s}$.

Taken together, all three studies cited by the reviewer and e.g. also Ligon LA & Steward O (2000) only studied a minor fraction of mitochondria and reported velocity data for those. This non-representative data focusing on the minor group of fast, actively transported mitochondria indeed is one magnitude above our average for all mitochondria, which is shown in **Fig. 3m**.

Although, as also the reviewer pointed out, our way of imaging (3D-imaging with an image stack of all channels every 10s) may not be optimal for catching very fast-moving mitochondria because it is optimized for correlating mitochondrial tethering and positioning with longer-term correlative recordings of dendritic outgrowth, **we nevertheless added a supplementary figure to the revised manuscript that dissects the data included in Fig. 3m into the two types of mitochondrial behaviors. The newly added Ext. Data Fig. 3-5 highlights that there are fast-moving, i.e. actively transported mitochondria (velocity $\geq 0.1 \mu\text{m}/\text{s}$; minority fraction) and mitochondria that seem to be completely tethered and/or move only slowly (velocity $< 0.1 \mu\text{m}/\text{s}$; the vast majority) (newly added Extended Data Fig. 3-5a,b).** With our imaging settings we noticed and individually tracked 24% of all mitochondria as fast-moving ones (**Ext. Data fig. 3-5a**). These data are very well in line with those reported by e.g. Loss & Stevenson, 2017 *Mol. Cell Neurosci.* (27%) or MacAskill et al. 2009 *Neuron* (~20% considered as fast-moving).

It is still possible that, due to the lower frame rates we had to use in order to track dendritic development in 3D, we still missed some of the fastest of the fast-moving, i.e. presumably actively transported mitochondria. This may explain why the minor group of fast-moving mitochondria in our newly added examinations show maximum track speeds of about 0.2 $\mu\text{m/s}$ (**Ext. Data Fig. 3-5b**) but e.g. Loss & Stevenson 2013 *Neuron* or Ligon & Steward, 2000 report values around 0.4 $\mu\text{m/s}$ for this minor group of mitochondria. However, it is important to note that the group slow-moving mitochondria representing the vast majority of mitochondria in dendrites and the focus of this study only change their positions with speeds of about 0.03 $\mu\text{m/s}$ (DIV7) (**newly added Extended Data Fig. 3-5a,b**). **The overall maximum track speeds of the major group of mitochondria therefore were absolutely identical to the quantitative data reported in the literature. Please compare e.g. Ligon & Steward's recordings with ours: in both cases exactly 0.03 $\mu\text{m/s}$ (Extended Data Fig. 3-5b).**

The overall motility of mitochondria in both control and rhotekin2-deficient neurons was unchanged, as shown before (Figure 3m). Also, the newly added live experimentation data based on both MitoTracker staining and mitoGFP overexpression did not show any overall differences (Ext. Data Fig. 3-5). We therefore focused on the mitochondrial positioning to comply with the reviewer's request to additionally show videos (see below; 3.).

3. Related to the above, dendritic mitochondria move rapidly to and fro along the longitudinal axis of dendritic shafts. From the small-step irregular movements shown in Fig.6, I suspect that the authors might have observed Brownian motions of stationary mitochondria rather than tethering or posing of moving mitochondria at dendritic branch points. Even if this were active movements, it is questionable if such subtle difference in the stationarity of mitochondria inside and outside the branch point make such a decisive difference in the chemical environment for determination of branch initiation sites. Here mitochondria were labeled with MitoTracker and traced in dendrites demarcated by farnesylated mCherry. It is not easy to distinguish mitochondria from those of neighboring cells as MitoTracker stains the mitochondria of all cells in high-density primary culture. I am also concerned about the punctate appearance of membran-targeted mCherry signals which should evenly stain dendritic membrane. Low magnified views and the videos they used for analysis should be shown. I would once again recommend mitoGFP or mitoDsRed for monitoring mitochondrial dynamics in live cells.

As already explained above, we were indeed not interested in the minor group of actively transported, fast-moving mitochondria but in the major group of dendritic mitochondria that shows some tethering and therefore change their positions only slowly and not always in a longitudinal manner related to microtubule tracks. **This is now more clearly pointed out in the revised manuscript.**

In addition, the individual movement of mitochondria and their positioning at dendritic branch induction sites are now visualized in form of supplementary movies attached to the revised manuscript. Supplementary videos have been added to the previously shown analyses of scr. RNAi and rhotekin2 RNAi in Figure 4 and Figure 5 (see Extended Data Videos 4-1 and Extended Data Video 5-1). Although it is difficult to document and represent the careful frame-to-frame analyses of mitochondrial positioning at branch induction sites, we hope that the reviewer and the readers will appreciate the mitochondria that are usually placed and tethered (trajectories are shown as well) in the central area at branch initiation sites, whereas such a positioning before and during dendritic branch initiation in rhotekin2-deficient neurons is impaired (as quantitatively shown in Figure 4 and 5).

Along the lines of the reviewer's request concerning the rhotekin2 RNAi and control data previously focused on, we also added further video examples to the newly added experimentation demonstrating the additional requirement of the rhotekin2 interaction partner

syndapin I in mitochondrial positioning at dendritic branch initiation sites (Extended Data Videos 7-1 and 7-2).

As far as MitoTracker stainings are concerned, this commercial filler of mitochondria is of course fully valid to label mitochondria. In order to back this up by experimental evidence and to also address concerns that putative difficulties in mitochondrial assignments to dendrites of transfected cells in MitoTracker samples may distort the mitochondrial length measurements, the newly added Ext. Data Fig. 3-5 clearly shows that the results obtained by MitoTracker and by overexpression of cotransfected mitoGFP are fully consistent and even nearly identical in absolute numbers (Ext. Data Fig. 3-5a,b vs. c,d).

In general, we like to note that difficulties to assign mitochondria to certain cells may indeed be some concern arising in dense neuronal networks but that at DIV 6 and at the neuronal concentrations optimized for tracing the dendritic arbor of cells this is not a concern. As e.g. seen in **Fig. 2c** or **Fig. 8**, the cultures of immature, still developing neurons used for analyses have plenty of empty space between individual dendrites of the respective cell and also between dendrites of surrounding cells (please also compare literature cited in the manuscript, such as Ahuja et al., 2007 *Cell*; Izadi et al., 2018 *J. Cell Biol.*; Wolf et al., 2019 *Nat. Cell Biol.*).

In addition, we would like to refer the reviewer to the fact that i) the tracking of mitochondria in **Fig. 4b** and **c** (via MitoTracker) is clearly not hampered by mitochondria that are not assigned to the dendrite under analysis and that ii) the dendritic segment in **Fig. 6j** also shows that anti-CytC-marked mitochondria can clearly be assigned to the dendrite under investigation (see also **Ext. Data Fig. 6-1** for individual anti-Cytc channel of the image).

As far as CherryF is concerned, this tracker for the plasma membrane also is established very well in many, including some of our studies. Slight discontinuities in CherryF stainings outlining cells are usually seen in correctly reported, i.e. not overexposed images. Such images may visualize that CherryF can be underrepresented in membrane areas occupied by other membrane proteins. Additionally, slight CherryF discontinuities may represent changes of membrane topology (particularly in Z-direction). None of this is of any concern for plasma membrane detections (please also see CherryF plasma membrane outlinings in the literature, such as in Schneider et al. 2014 *J. Cell Biol.*; Wolf et al., 2019 *Nat. Cell Biol.* or Englisch et al., 2024 *Cell Rep.*).

4. In Figs. 4 and 5 the authors insist that mitochondria are stalled at the newly formed dendritic branch sites by rhotekin2. Dendrites of differentiating pyramidal neurons in the first week of culture are highly dynamic and frequently extend transient filopodia along the shaft, and only few of them are stabilized to form new dendritic branches. Observation of a small dendritic compartment for only 2 minutes is not sufficient to correlate mitochondrial motility and dendrite dynamics. A long time-lapse imaging of over 30 min to several hours is required to claim that organelle tethering precedes new dendritic branch formation (Cui-Wang et al., *Cell* 2012).

This must be a misconception and we apologize for apparently not having described this well enough in the Material and Method section but instead just citing the respective literature for this.

It is correct that **Fig. 4 and 5** only show the two minutes around dendritic protrusion induction but, as we described before in several of our publications tracking the development of the dendritic arbor by 3D-live analyses (e.g. see Wolf et al., 2019 *Nat. Cell Biol.* or Izadi et al., 2023 *eLIFE*), it is of course necessary to image much longer periods of time to identify and follow dendritic branches being formed. As we are mostly interested in the step of dendritic branch induction (and not in branch stabilization, microtubule entry, branch propagation and such), we usually image 10-20 min, which – as published - due to the dynamics of the cells is more than enough to i) catch forming protrusions from the existing dendritic arbor with decent probabilities and to ii) track these processes well before and after dendritic protrusion induction and at the same time iii) image in full 3D resolution with the

required high time resolution in the seconds range (e.g. see Wolf et al., 2019 *Nat. Cell Biol.* or Izadi et al., 2023 *eLIFE*).

Similar to Wolf et al., 2019 *Nat Cell Biol.*, the current study of dendritic protrusion induction was done with imaging at least 10 min intervals. **This information is now explicitly repeated in the revised manuscript instead of just citing the literature and also the fact that we focus on branch induction is pointed out clearer in the revised manuscript.**

Similar to the literature cited by the reviewer, we also use a length cut-off of 10 μm during dendritic arbor analyses to exclude small and putatively short-lived structures, for which it may be unclear whether they may develop into bona fide dendritic branches or not. This information was covered by the literature we cited (Ahuja et al., 2007 *Cell*; Izadi et al., 2018; Schwintzer et al., 2011 *EMBO J.*) and was additionally repeated in the Material and Method section of the manuscript.

Apart from this, we would like to note that whether or whether not morphological structures protruding out from existing dendritic arbor are later stabilized by microtubule invasion and how far they are extended or whether they themselves will be further branched at some point during the development of neuronal networks is not relevant for our current study, as **the defect we identified for RTKN2 deficiency clearly is at the branch induction step, as shown by our new experimentation added (Fig. 8j of the revised manuscript). Also the mitochondrial positioning and tethering examinations explicitly focus on branch initiation step (Fig. 4 to Fig. 7 of the revised manuscript).**

5. Mitochondrial tethering has also been implicated in axon branch formation (Courchet et al., *Cell* 2013; Spillane et al., *Cell Reports* 2013). This paper avoids mentioning the localization and function of rhotekin2 in the axon. However, differences in dynamics and regulation of mitochondria in the axon and dendrites are important and the authors need to discuss whether rhotekin2 function is dendrite-specific. At least the length of axonal mitochondria and the number of axonal branch points should be comparatively analyzed in rhotekin2-depleted cells.

The reviewer is right, mitochondrial dynamics in the axon also is a very interesting and important aspect. However, this is a completely different topic and, in part, the processes in the axon are mechanistically fundamentally different from those in dendrites.

These differences also include the induction of new branches, as for example syndapin I, Abp1 and Arp2/3 complex loss-of-function at least in culture lead to an apparently uncontrolled outgrowth and branching of the axonal compartment (Pinyol et al., 2007; Dharmalingam et al., 2009), whereas, in dendrites all of these components are required for dendritic arborization in different neuronal systems (e.g. Schwintzer et al., 2011; Stürner et al., 2019; Hasegawa et al., 2022).

Syndapin I also seems to occur in axons and we also have no indication that rhotekin2 may be restricted to the soma and dendritic arbor.

Although we did not explicitly study rhotekin2's distribution to the dendritic and axonal compartment, we have out of curiosity experimentally addressed in our revision work whether also mitochondria in the axonal compartment would show size changes upon rhotekin2 loss-of-function (see Reviewer Figure 3 below). We based these new analyses on coexpression of mitoGFP as mitochondrial tracer and used the skeleton tool of ImageJ for analyses of 3D images of axons obtained by using an APOTOME providing a good resolution of individual mitochondria. The mean length of axonal mitochondria in control cells (0.600 μm) fit well with our different measurements of dendritic mitochondria. Interestingly, rhotekin2 deficiency clearly resulted in an increase of mitochondrial length also in the axonal compartment (**Reviewer Figure 3** (below)).

The newly identified rhotekin2 loss-of-function phenotype in the axonal compartment is in line with the phenotype identified in the dendritic compartment (**Fig. 3; Ext. Data Fig 3-1**). Both observations furthermore are fully consistent with our studies in non-neuronal cells (**Ext. Data Fig 3-2**).

If the reviewer finds the new axonal data informative and suitable for the manuscript rather than being rather distracting from the focus on dendritic arbor development, we shall be happy to also include these new examinations as a further Extended Data Figure into the revised manuscript.

Together, our neuronal and non-neuronal experiments demonstrate that the identified rhotekin2 loss-of-function phenotype represents a general, i.e. neuronal cell compartment-independent and cell type-independent defect.

Reviewer Figure 3. Rhotekin2 deficiency also leads to increased length of axonal mitochondria. Quantitative determinations of mitochondrial length in axons of developing hippocampal neurons transfected with either scr. RNAi (control) and RTKN2 RNAi coexpressing mitoGFP for mitochondrial tracing reveal that also axonal mitochondria are increased in length upon rhotekin2 deficiency. Data are mean±SEM. Scr. RNAi, n=97; RTKN2 RNAi, n=105 axonal mitochondria evaluated based on mitoGFP tracing and using the skeleton tool of ImageJ. Statistical significance, Mann-Whitney. * P<0.05. For exact P value see figure.

Minor comments

1. The TEM analysis is described as having been done on hippocampal tissue, but must have been done by identifying mitochondria in the dendrites of a specific type of neurons (such as CA3 pyramidal neurons). The size and shape of mitochondria are very different in distinct cell types and in different subcellular compartments in neurons.

The analyses were done in the CA3 region. **The revised version of the manuscript now contains this information.**

2. The blot in Fig.1k contains many thick faint bands. Are they background signals?

The faint thick bands are the background of the GST fusion proteins used. Often, such low signals are falling below the detection level using ECL and/or they are erased by image processing (background subtraction). In contrast, we show unprocessed fluorescence images recorded by a LICOR Odyssey Imaging System, which records signals in a range of 6 orders of magnitude, and we did not artificially erase low grey values.

The revised manuscript points this out and also highlights the bands that are of interest better by increased sizes of the green arrowheads marking them (Fig. 1k of the revised manuscript).

3. The neuron transfected with RTKN2 RNAi in Fig.3b looks like an interneuron (with a small soma

and no clear polarity). Due to a limit to the number of antibodies for multi-color IF, it may be better to use a pyramidal neuron marker (such as Math2) rather than Map2 for morphometry in Fig. 7, since axons and dendrites can be distinguished by morphology labeled by a volume marker.

We have no indication that rhotekin2 expression is restricted to any special type of neuron (see its expression in e.g. NIH3T3 cells). We study the mechanisms of early morphogenesis and our focus is the development of the dendritic arbor and, in principle, we do not have a focus on any processes, which may require to look at excitatory neurons exclusively. However, primary cultures prepared from the hippocampus are mostly reflecting pyramidal neurons. This quite high homogeneity is one of the reasons why hippocampal neurons are so widely used. Whether or whether not other types of neurons may also be included in the analyses is not a concern, as these will remain a small minority and may statistically occur with very low rates in all conditions. By morphology, interneurons are anyway not clearly distinguishable from pyramidal neurons in the dissociated cultures at these early stages of development in vitro, as pyramidal neurons do not necessarily show their typical appearance that is well-known from brain tissue. **As the mitochondrial phenotype was not clearly visible in the example shown, we anyways replaced the image in Fig. 3b by an image of another more typical cell in the revised version of the manuscript and hope the reviewer is more content with the example now shown in Fig. 3b in the revised manuscript.**

Apart from this, the reviewer will acknowledge that the dendritic marker MAP2 of course is the suitable tracer for dendritic arborization – as it is very well established in the literature that MAP2 is not only an endogenous tracer independent from any of the to be tested transfection conditions but also a marker for the dendritic arbor specifically. As a microtubule binding protein, it i) identifies dendrites (vs. axons) and ii) highlights the structures that are fully MT-invaded, i.e. are fully established, mature dendrites, because MAP2 is a MT-binding protein.

It is therefore undisputable that a putative distinction of the dendritic and the axonal compartment and tracking neuronal morphology by volume markers clearly is inferior to the MAP2-based analysis of specifically the dendritic arbor.

4. I believe that analysis of functional interaction of syndapin I and rhotekin would strengthen the conclusion of the manuscript. If mitochondrial tethering to the nascent branch points via syndapin I-rhotekin2 interaction is a mechanism of dendrite branch formation, the dendritic overgrowth in syndapin I overexpressing neurons (Schwintzer et al., 2011) should be at least partly rescued by rhotekin2 knockdown.

We conducted the experiments suggested by reviewer 2. In the months of our revision work, we studied whether dendritic branch formation brought about by syndapin I overexpression would be dependent on rhotekin2 (see Reviewer Figure 4 (addressing a question of Reviewer 2 – therefore see there)).

Also in these cotransfection experiments, we were able to observe the syndapin I overexpression phenotype. Similarly, the experiments also showed the rhotekin2 loss-of-function phenotype of defective dendritic arborization again. **Syndapin I-overexpressing but rhotekin2-deficient neurons, in contrast, showed dendritic arborization parameters similar to control cells.** Considering that syndapin I as F-BAR domain protein has i) the ability to shape membranes by itself and has ii) the means to promote F-actin formation by recruiting several components of the actin cytoskeleton promoting neuronal cell shape development, an only partial dependence of syndapin I functions in dendritic arbor formation on the mitochondria anchoring by rhotekin2 appears very well understandable.

In case the reviewers or the editors feel that - instead of reporting these data as a reviewer figure (please see Reviewer Figure 4 below) - these data shall rather be included into our manuscript, we shall be happy to show them as another Extended Data Figure.

Reviewer Figure 4. Syndapin I-mediated dendritic arborization is partially suppressed by rhotekin2 RNAi.

(a-d) Quantitative evaluations of dendritic parameters of double-transfected primary hippocampal neurons. Analyzed was the dependence of syndapin I-mediated dendritic arborization on rhotekin2. Note that both the syndapin I gain-of-function phenotype and also the rhotekin2 loss-of-function phenotype is recognizable upon double-transfections. Importantly, in combination with rhotekin2 RNAi, all of the syndapin I gain-of-function phenotypes are suppressed to about control levels by rhotekin2 RNAi. This may argue for a partial dependence of syndapin I-mediated dendritic arborization on rhotekin2 but dendritic parameters of Sdpl/RTKN2 RNAi neurons are still elevated when compared to rhotekin2 loss-of-function values. Data, mean \pm SEM visualized as bar plots. N numbers as indicated in figure. Statistical analyses, 1-way ANOVA/Tukey's **(a-c)** and 2-way ANOVA/Bonferroni **(d)**. * $P < 0.05$; ** $P < 0.01$; *** $P < 0.001$; **** $P < 0.0001$.

Importantly, we furthermore conducted experiments that directly address the importance of syndapin I in the rhotekin2-dependent mitochondrial positioning we observed at nascent dendritic branch induction sites. We have demonstrated by reconstitution experiments that syndapin I is able to recruit rhotekin2-containing mitochondria. This suggested that, besides the identified rhotekin2 loss-of-function phenotype, also syndapin I deficiency may lead to an impairment of the observed mitochondrial tethering, if syndapin I is crucial in this process. **The new experimentation added to the revised manuscript now indeed demonstrates that the rhotekin2 interaction partner we identified, syndapin I, which is a critical component in dendritic branch induction** (Schwintzer et al., 2011 *EMBO J.*; Koch et al., 2020 *Cereb. Cortex*; Izadi et al., 2021 *eLIFE*;

lzadi et al., 2023 *J. Cell Biol.*), **plays a crucial role in proper mitochondrial positioning at branch initiation sites. The finding that syndapin I loss-of-function data very well phenocopies the identified rhotekin2 loss-of-function phenotype in mitochondrial positioning is included as the newly added Fig. 7 of the revised manuscript.**

This additional importance of syndapin I also is very well in line with the finding that two rhotekin2 loss-of-function-associated phenotypes we identified, i.e. the increase in mitochondrial length and also the defects in dendritic branching, were strictly dependent on the identified syndapin I binding site in rhotekin2, the KRAP motif, in the rescue experiments we conducted in the analyses of both rhotekin2 loss-of-function phenotypes (**Fig. 3a-e,k and l; newly added Ext. Data Fig. 3-1; Fig. 8 of the revised manuscript**).

5. Judging from the stretched signals and the horizontal stripes of artifacts, Fig.6j appears to be a forced reconstruction of z-serial SIM images with weak signals, which does not provide evidence supporting the signal co-localization. The authors should use single z-plane images to demonstrate that syndapin signals localize at the membrane curvature of branch points and they are juxtaposed by rhotekin2 and mitochondria.

The reviewer is right, the 3D rotation shown in Fig. 6j in part shows a somewhat striped appearance of fluorescence signals in a few locations (such as in the lower left corner). This is not due to a forced reconstruction of z-serial SIM images of weak intensities but due to a limited number of z-sections and their optical spacing. With dendritic branch induction occurring in many directions, the reviewer will acknowledge that one just has to select a 3D rotation that is biologically most informative and that one may therefore has to accept that some minor areas of the large picture may not represent the most optimal appearance of fluorescence signals. This may be esthetically unfavorable but is not of a concern for the 3D reconstruction of the vast majority of fluorescence accumulations shown in picture.

In case the reviewer strongly feels about this, we shall be happy to omit the left part of the image.

6. The spatial resolution of Elyra 7 is only 120 nm and one cannot measure distances of 40 and 80 nm. Specific techniques such as fluorescence cross correlation spectroscopy (FCCS) or proximity ligation assay are more suitable to detect the association of molecules.

We apologize for this mistake and the irritation this may have caused. **The revised manuscript we herewith submit now matches the (revised) figures.**

REVIEWER COMMENTS

Reviewer #1 (Remarks to the Author):

In the revised manuscript, the authors have provided new data using alternative methods to validate their mitochondrial size findings, as well as presented new data for Rhotekin2 overexpression on branching and the requested validations thus addressing most of my concerns.

We thank the reviewer his/her efforts to improve our manuscript. We are happy to see that the reviewer is assessing the work that we included in our revision work as suitable to address the concerns of the reviewer.

The few concerns that apparently still remained we shall be happy to address below, by the requested changes to the newly revised manuscript and by new experimentation included into the newly revised manuscript, respectively.

However, the authors propose that Rhotekin2 is a mitochondria localized protein and syndapin is localized to branch points to recruit the mitochondria to this site. Figure 6J doesn't appear to support this model though. Syndapin appears to be all along the dendrite, while the Rhotekin2 appears to localize to the branch point, but not by surrounding the mitochondria. The authors may want to be more cautious in their interpretations/model and should explain this discrepancy in the discussion.

We replaced the high-resolution 3D reconstruction images in Fig. 6j by new SIM imaging data. We hope the reviewer will find this better suited to visualize spatial overlap of mitochondria, rhotekin2 and syndapin I (please see revised Fig. 6j). As the reviewer suggested, these topics also are described and discussed more extensively in the revised manuscript.

In general, the reviewer is right, the syndapin I localization is not restricted to sites that overlap with mitochondria, as syndapin I is a plasma membrane-associated protein (as e.g. demonstrated by Immunogold labeling of freeze-fractured plasma membranes of both neurons and in brain tissue (Schneider et al., 2014 *J. Cell Biol.*; Izadi et al., 2023 *J. Cell Biol.*)) and furthermore may also have a rather cytoplasmic pool. As a further complication, it is possible that complete colocalizations of rhotekin2 and syndapin I are disturbed by steric hindrances, as the epitope of the anti-rhotekin2 antibodies overlaps with rhotekin2's syndapin binding site (**please see newly revised manuscript**). Yet, we hope that the reviewer will agree with us that high resolution imaging by SIM shows that there are mitochondrial surfaces with overlapping signals of both rhotekin2 and syndapin I at sites of dendritic protrusion (**please see revised Fig. 6j**). Complying with the reviewers' suggestions, the revised figure now focusses on showing the spatial overlap by presenting the single fluorescence channels rather than showing surface reconstructions (**please see revised Fig. 6j**).

In response to main point 4 about the role of calcium, I was not intending to suggest that calcium be removed from the model but instead that showing data that calcium dynamics are altered in the nascent dendrites/branch points following Rhotekin2 manipulation would strengthen the suggested model, as well as provide a potential reason for Rhotekin2 to regulate both mitochondrial size and anchoring at the same time. It is well established that mitochondrial calcium uptake plays important roles both in the axon and dendrites (see work from Josef Kittler and Franck Polleux labs).

It would indeed be great if it were possible to show alterations of mitochondria-mediated Ca²⁺ dynamics at branch induction sites by direct experimental evidence for e.g. alterations in the decay times of calcium transients at branch induction points in dependence of a proper mitochondrial tethering in the central zone and the lack of such a correct positioning and tethering upon rhotekin2 deficiency.

However, although we very much appreciate the interest of the reviewer in this topic, one has to honestly admit that the imaging conditions required to reliably detect and temporally resolve profiles of Ca²⁺ transients in immature neurons for quantitative analyses (high speed imaging in the lower millisecond range for a few seconds; high laser power due to weak signals) and the imaging conditions required for parallel and independent recordings of dendritic tree development (full 3D image stacks for cellular morphology reconstructions at very low laser power over a long time) are completely incompatible. Besides risking massive cytotoxicity artifacts, which preclude any meaningful analysis, also the time needed for 3D image stack recording sets a general technical limit for time frame lowering. This incompatibility cannot be overcome by contemporary techniques.

This is now briefly explained in the revised manuscript. We hope that the reviewer is content with this solution.

In addition, we expanded the introduction and discussion about the established role of mitochondria in calcium homeostasis. **The revised manuscript now also includes citations of the seminal work of the Kittler and Polleux labs** (López-Doménech et al., 2016 *Cell Rep*; Courchet et al., 2013 *Cell*).

Paragraph 3 of page 9 (line 171) states “The reduction in mitochondrial length upon rhotekin2 RNAi” shouldn’t this be increase?

We sincerely thank the reviewer for noticing this embarrassing mistake. It was indeed an increase that was observed upon rhotekin2 deficiency. **The mistake has been corrected in the newly revised manuscript.**

The authors may want to be careful about the interpretation of Rhotekin2’s interaction with Dynamin 2 but not Drp1 for mitochondrial fission as the dynamin 2’ role is somewhat controversial in mitochondrial fission (see Fonseca TB et al, Nature, 2019).

We are grateful to the reviewer for pointing this out. We were aware of the study by Fonseca et al., 2019 and therefore tried to word this hypothesis carefully. Following the advice of the reviewer and instead of just leaving open whether dynamin2/dynamins also are critical for this process by merely stating that dynamin2 was shown to “*be capable to power mitochondrial fission*”, **the newly revised manuscript now explicitly mentions the dispute caused by both Nature papers. Consequently, also Fonseca et al., 2019 was added to the reference list of the newly revised manuscript.**

Reviewer #2 (Remarks to the Author):

The authors strengthened the evidence of the functional interaction of RTKN2 and syndapin1 with additional experiments.

We are happy to see that the reviewer is content with the additional phenotypical analyses revealing a previously unknown syndapin I loss-of-function phenotype in mitochondrial positioning at central dendritic branch sites of developing neurons that strengthened the identified functional interaction of RTKN2 and syndapin I.

However, they did not answer many concerns raised in the first review.

While we already addressed many concerns raised by the reviewer in his/her first review and this also has been acknowledged by reviewer1 who in part brought up similar issues, **we have addressed the points of the reviewer raised in this second review (please see below, the newly revised manuscript and the experimentation added in the second revision work phase).**

The most important concern was about the quality of imaging, which is the key to the conclusions in the second half of the manuscript that under deficiency of RTKN, mitochondria are not tethered to the plasma membrane, causing increased movement of mitochondria and down-regulation of dendrite formation. Regretfully, the temporal and spatial resolution is insufficient to evaluate the mitochondrial motility to support their conclusion. In the present form, this study does not support sufficient evidence that RTKN2 and syndapin1 regulate dendritic branch formation by tethering mitochondria to the plasma membrane transiently at branch initiation sites.

As demanded by the reviewer, we have replaced the SIM imaging data in Fig. 6 by new experimentation (also see suggestion of the reviewer below; revised Fig. 6j).

We agree with the reviewer that imaging conditions often are limitations in cell biology. Our study already contains electron microscopy, SIM, 3D spinning disc live microscopy and structured illumination by Apotome imaging – almost all of these in form of quantitative data sets. Analyses of mitochondrial positioning (**Fig. 4, Fig. 5, Fig. 7**) only require three pictures: before, during and after branch induction. Thus, for all of these figures, temporal resolution is not of any concern.

For the remaining Fig. 6a-i, we conducted new 3D live experiments with improved temporal resolution (please see newly added Extended Data Fig. 6-1). The temporal imaging conditions criticized by the reviewer are largely technically dictated by the need to unambiguously detect and follow dendritic branches to be studied. **This is now better explained in the revised manuscript.** On one hand it is obviously required to record multicolor z stacks of images over a period long enough to follow dendritic arbor development (at least 20 min), while, on the other hand, mitochondrial movements are best tracked in the low second range.

While we already employed spinning disc microscopy to speed up imaging by technical means as much as possible (**Fig. 6**), we have now worked on pushing the conditions of temporal imaging to a further limit. **We are happy to report that - mainly by sacrificing as much 3D information as possible (reduction to merely 3 z planes) - we have been able to speed up imaging by more than factor 4 in our revision work. The frame rate achieved thereby was in the lower second range (2.4 s per multicolor z stack frame).**

Using this much faster imaging, we have then repeated all of the quantitative analyses of the temporal imaging shown in Fig. 6 (please see Ext. Data Fig. 6-1). These new data sets fully corroborated the observations shown in Fig. 6. They again consistently revealed the identified reduction of mitochondrial movements at dendritic branch induction sites versus at non-branching sites in control cells (**compare Fig. 6b,c with Ext. Data Fig. 6-1a,b**). They also again clearly showed a

disruption of this tethering at branch induction sites upon rhotekin2 loss-of-function. Also in the new experimentation with the more than 4fold improved temporal resolution, rhotekin2 deficiency consistently led to statistically significantly higher maximum track speeds, spot acceleration and displacement length at dendritic branching sites (**compare Fig. 6e-i with Ext. Data Fig. 6-1c-g**). **The temporal resolution of both imaging conditions thus turned out to be sufficient to detect motility changes of the mitochondria tethered at dendritic branch induction sites.**

For reviewer2, it may additionally be noteworthy that when, instead of MitoTracker (**Fig. 6; Ext. Data Fig. 6-1**), mitoGFP overexpression was evaluated at dendritic branch sites in confirmatory live microscopy experiments, the same increase of mitochondrial mobility by about 30% was observed upon rhotekin2 RNAi (**see reviewer Figure below and please compare to Fig. 6g and to newly included Ext. Data Fig. 6-1**).

Reviewer Figure 1. Mitochondrial velocity determinations at dendritic branch initiation sites obtained by tracing mitochondria by mitoGFP overexpression.

Note that also these experiments highlight the same rhotekin2 loss-of-function phenotype as in the MitoTracker tracings of mitochondria (**Fig. 6g; Extended Data Fig. 6-1e**).

Similar to the mitochondrial length phenotypes, which we corroborated by different methods, this additional data set using mitoGFP again demonstrates that also irrespective of the method of mitochondrial tracing employed, the phenotypes identified by our study are consistently observed. **We shall be happy to include this confirmatory data, too, if deemed necessary.**

The mechanism of mitochondrial elongation in RTKN2 deficiency is also based on speculation from previous studies, and its significance in the dendritic phenotypes is not explained.

We substantiated the mechanistic considerations regarding mitochondrial length changes by including experimentation previously merely shown as a reviewer 1 Figure to address his/her questions along these line (please see newly added Extended Data Fig. 3-5). The additional data provided show that the respective mechanistic considerations were not just pure speculation but have an experimental data basis.

We accordingly also changed this part in the discussion of the newly revised manuscript.

Fig.6 is a key to demonstrate the mechanism of RTKN2 function. I wonder how the spots were determined in the analyses of mitochondrial speed by spot tracking in Fig. 6a-i. The videos 4-1,5-1,7-1,7-2 show intermittent movement at 10-second intervals, with brightness fluctuations due to focus shifts, making it seem challenging to continuously track the same position of mitochondria which dynamically change their shape in the videos. It seems that spot object of IMARIS was used, but the authors need to indicate the method by which the spots were determined.

The tracking function in IMARIS 8.4 was used for spot determinations. The same applies for the newly added faster imaging at reduced z resolution that fully corroborated the findings reported in **Fig. 6 (Fig.**

6a-i and newly added Ext. Data Fig. 6-1) (please see the Material and Method section of the revised manuscript).

Considering the length of the mitochondria measured using MitoTracker (~0.8 μm ; ExtFig. 3-1), it is speculated that even a slight shift in the spot's position could result in random movements similar to those shown in Fig. 6. I am afraid that the increased movements in RTKN2 KD could be due to the increased mitochondrial length to ~1.4 μm . It is hard to evaluate the results until it is clearly negated that the difference was caused by the inappropriate processing of the movies when the trajectories were overlaid.

We thank the reviewer for carefully thinking through whether it could be that the observed movements may in fact represent spot position shifts within a given, non-motile mitochondrion. It may indeed appear as if the mean values for displacement length and mitochondrial length may theoretically be almost able to cover such a behavior, although this would demand that such fluorescence movements would be solely along the longitudinal axis of a given mitochondrion and would furthermore fully cover all the available mean length up to the very tips of the mitochondria.

While it already sounds highly unlikely that maxima of fluorescence filling an ellipsoid body always are at opposing tips during the start and the end of an examination period instead of somewhat central, it furthermore is immediately clear that the trajectories are of course not single lines along the mitochondrial length axis but extend in all directions (as also obvious from **Figure 6a, d**).

Most importantly, such putative artifacts also can be controlled for and of course, we did this. It can easily be excluded visually that movements are not fluorescence fluctuations within an in fact non-motile mitochondrion, as in such a case, the trajectory would fall into the area of the examined mitochondrion and would fail to extend beyond it. **We now added the mitochondrial shapes and positions to the images in Figure 6a, d and also marked the start and the end of the trajectories by colored dots to improve the mitochondrial tracking information.**

The hypothetical concern raised by reviewer2 is thereby now directly addressed by the experimentation shown and readers can now easily see that the trajectories of course by far extend over the area of the given mitochondrion, i.e. there are no hypothetical fluorescence maxima of moving from tip to tip inside of mitochondria but our recordings track mitochondria that are changing their positions.

In order to appreciate the general validity of these observations and that the images in **Fig. 6a** and **Fig. 6d** are representative, it also has to be noted that the examples of mitochondria shown represent the median of size and/or even have slightly longer lengths than the median of the mitoGFP-based mitochondrial length data for the respective condition (or mean \pm SEM see **Fig. 3e**).

Also this information is now included in the revised manuscript.

Data as follows:

Fig. 6a (control): mitochondrion #1 (outside of central branch induction area), 0.51 μm ; mitochondrion #2 (inside of branch induction area), 0.59 μm ; median of mitochondrial length in control cells, 0.51 μm .

Fig. 6d (rhotekin2 RNAi): mitochondrion tracked, 0.79 μm ; median of mitochondrial length in rhotekin2-deficient cells, 0.72 μm .

I believe it is necessary to improve the temporal resolution, which can be at a lower magnification, and quantify how the "slow movement" they define changes in the RTKN2 ND condition.

While the magnification cannot be lowered, as mitochondria and branch inductions as well as the central area of the branch induction site need to be visualized clearly, **we are happy to report that we reproduced all the quantitative data on mitochondrial movements at branching and at non-branching sites in control and in rhotekin2-deficient neurons with more than factor 4 faster imaging by sacrificing as much 3D information for dendritic branch induction as possible. As described above, the results obtained were fully consistent (Fig. 6a-i and Ext. Data Fig. 6-1).**

Addressing the concern of reviewer2, the temporal resolution of both imaging conditions thus turned out to be sufficient to detect motility changes of the tethered mitochondria at dendritic branch induction sites.

Given the quality of the image in Fig. 4b, there are concerns about the analyses in Figs. 4, 5, and 7, too. Nonetheless, I would accept the conclusion that new branches could be more likely to form at the sites where mitochondria are tethered, considering the recent studies demonstrating that tethered mitochondria provide chemical environment supporting dendritic branch maintenance (Faits et al., 2016; this paper indicates that mitochondria are condensed only after a branch forms) and spine maturation (Rangaraju et al., 2019). Rangaraju et al. compared spine maturation with or without ablation of mitochondria in 10-25 μm dendritic compartment.

We are happy to see that with the new experimentation added during the first revision, the reviewer accepted the conclusion that new branches are more likely to form at sites where mitochondria are tethered.

The mitochondrial detection with MitoTracker and the mCherryF channel shown as highly magnified merged images in Fig. 4b and c is now additionally shown as individual panels in the Extended Data Figure 4-2. The individual presentation of the MitoTracker channel demonstrates that the detection of mitochondria by MitoTracker (highlighted by a green surrounding line to help the reader to see their position easily) is of no concern. The same applies for Fig. 5 and Fig. 7.

On the other hand, I wonder if such differences could occur on a scale of $\pm 1 \mu\text{m}$ where mitochondria quickly move around the tethered positions? Dendritic mitochondria are $2 \mu\text{m}$ in average in many studies, and in this paper $0.8 \sim \mu\text{m}$ by MitoTracker. I asked in the first round review whether the increase in this level of movement in RTKN KD or branch outside, if it exists at all, would create enough difference in the local chemical environment to determine whether or not a protrusion forms. Mitochondria are highly abundant in dendrites and occupy a large percentage of the arbor (Lewis et al. 2018 also shown in Ext Fig. 3-1). The authors also estimated ~ 0.8 mitochondria per $1 \mu\text{m}$ segment with or without RTKN2, suggesting that new branches could be often centered by a mitochondrion. One also wonders if overall dendritic length and branch number could significantly decrease just by the increase in slow diffusive movement of mitochondria if density and distribution are not affected (Fig. 3).

We appreciate the thoughtful considerations of the reviewer. The density of mitochondria in dendrites was indeed about $0.3/\mu\text{m}$ in proximal and distal areas of the dendritic tree (Fig. 3j,k) and the length was about $1 \mu\text{m}$ (Fig. 3e). That obviously leaves many dendritic segments without mitochondria. It may, however, be even more insightful to look at the probability of mitochondrial distribution. We have explicitly determined this experimentally in Fig. 4 (and the loss of the identified mitochondrial positioning upon either rhotekin2 or syndapin I deficiency was quantitatively addressed in Fig. 5 and Fig. 7). Fig. 4 (and also the control data in Fig. 5 and Fig. 7) clearly demonstrates that the mitochondrial distribution is not random but mitochondria have a high probability to occur specifically at dendritic branch induction sites at the time of dendritic branch induction and thereafter. With almost 80% probability at the dendritic branch area (Fig. 4e,f), this leaves very few mitochondria for the dendritic segments distal and proximal from it – as also clearly shown in Fig. 4. The differences observed are about factor 4-18, depending on which time point is examined and whether distal or proximal areas are compared to the central branch induction site (Fig. 4e,f). The reviewer will agree with us that, in general, it is more than reasonable to assume that factor 4 to factor 18 differences in some cell biological parameter indeed do matter for life functions.

In line, our experimental data clearly shows that these mitochondria-containing dendritic arbor segments are privileged over the remaining areas, as without this local discontinuity in mitochondrial distribution and without proper tethering of mitochondria at nascent dendritic branch sites, dendritic arbor development and specifically the process of branch induction turned out to be massively impaired (**Fig. 8**).

In order to link the reduction of new branch formation to the altered morphology in RTKN2 KD, branch formation and elimination should be quantified by long term imaging at low magnification (Horton et al., Brain Cell Biol. 2007; Wu et al., PLoS ONE 2015).

In compliance with the explicit demand of the reviewer to address whether the altered morphology of the developing neurons at DIV4+2 really is a result of impaired dendritic branch induction specifically, as suggested by our model, we would like to refer the reviewer to the live experimentation quantifying the frequency of dendritic protrusion initiation events newly added during first revision (Fig. 8j).

The reviewer may have overlooked this quantitative panel. It clearly shows that explicitly the initiation of dendritic protrusions is impaired upon rhotekin2 deficiency.

In order to also have the accompanying technical information better accessible for readers, we converted this into a separate subchapter in the Material and Method section of the newly revised manuscript.

The SIM image also remains unchanged and insufficient to support the localization of syndapin to the plasma membrane or the close proximity of syndapin/RTKN/mitochondria. Individual z-plane images instead of the 3D reconstruction should be shown for better evaluation.

We replaced the high-resolution 3D reconstruction images in Fig. 6j by microscopy images (new SIM imaging data) in Fig. 6 of the revised manuscript.

We hope that the reviewer will agree with us that high-resolution imaging by SIM shows that there are mitochondria with overlapping signals of both rhotekin2 and syndapin I (**please see revised Fig. 6j**). Complying with the reviewers' suggestions to rather focus on showing the spatial overlap, surface reconstructions were omitted and the presentation of the corresponding single fluorescence channels was directly integrated into the **revised Fig. 6j**.

The authors previously showed transient accumulation of syndapin 1 at branch initiation sites using live imaging of fluorescently labeled recombinant proteins (Izadi et al. 2021). I would be more convinced if they demonstrate that fluorescently labeled RTKN2 and mitochondria accumulate at branch initiation sites at this time resolution and magnification.

3D live microscopy analysis of rhotekin2 localization during dendritic branch formation similar to the syndapin I studies in Izadi et al. 2021 (which used a frame rate identical to the morphology and mitochondrial tracking done in this study (10 s/frame)) are unfortunately not possible. Please see manuscript text at page 7: *"In NIH3T3 fibroblasts, our affinity-purified anti-RTKN2 antibodies immunolabeled subcellular structures showing a clear spatial overlap (Pearson's coefficient, 0.8) with the mitochondrial marker cytochrome c (Fig. 2a,b). Also overexpressed rhotekin2 localized to mitochondria when expressed untagged"*. Due to the complication that e.g. GFP-tagged rhotekin2 seems to be non-functional, we built systems for the expression of untagged rhotekin2 (GFP-reported) for all functional work in both non-neuronal and neuronal cells and/or studied endogenous rhotekin2 where ever possible. We clearly demonstrated by a variety of means that rhotekin2 is strongly associated with the outer mitochondrial membrane (**Fig. 2; Fig. 6**).

Furthermore, we would like to refer the reviewer to the revised **Fig. 6j**, which shows spatial overlap of endogenous rhotekin2 and syndapin I at a dendritic branching site by SIM (**revised Fig. 6j**).

Mitochondria length

In the revised manuscript the authors have added new data using a volume marker and measured dendritic mitochondria as $\sim 0.8 \mu\text{m}$, slightly longer than they did with CytC ($0.4 \mu\text{m}$) but not yet comparable to previous literature. They included both sets of data, assuming that the variation would be seen using different methods. I disagree with this point and believe it is simply because the method using punctate CytC is inaccurate.

Mitochondrial size is very diverse in different cell types and subcellular compartments. Most studies quantifying dendritic mitochondria length in cultured hippocampal neurons by a volume marker reported similar length of around $2 \mu\text{m}$ as you indicated in the letter (Chang et al., 2006; Lewis et al. 2018; Hatsuda et al., 2023). Cho et al. (Nat Comm 2017) did not distinguish dendrites and axons, reporting 1 and $1.5 \mu\text{m}$ in two independent analyses, probably due to significantly smaller mitochondria sizes in the axon (mitochondrial length is not critical in their paper). Triolo's paper is not at all informative, as it uses adult muscle stem cells for measurements. I don't understand why the authors insist on values that are so different from results repeatedly shown in previous studies. The authors claim that the relative length of mitochondria was similarly altered by RTNK2 and syndapin1 deficiencies, but we cannot accept something for which the absolute length is not correctly measured.

We have replaced the entire quantitative data sets in Fig. 3 obtained by tracing the endogenous component cytochrome c by corresponding work using mitoGFP overexpression (please see revised Fig. 3 for the extended analyses based on mitoGFP).

Importantly, the new mitoGFP/skeleton-based experimentation consistently, and even more clearly than the former cytochrome c-based data, show that rhotekin2 RNAi specifically and KRAP-motif-dependently leads to increased mitochondrial length in developing neurons (**Fig. 3; Extended Data Fig. 3-1, 3-2**).

The results also were in line with data obtained by using MitoTracker in live experiments to examine mitochondrial length (**Extended Data Fig. 3-1**).

The reviewer will furthermore be happy to see that the data obtained during our revision work is virtually identical to e.g. data reported by the Lippincott-Schwartz and Mattson labs for immature neurons in culture (Yao et al., 2017 *Mol. Biol. Cell*) and to data reported by Cho et al. (2017, *Nat. Commun.*) – both reporting an average of $1 \mu\text{m}$ in length.

As the reviewer also pointed out, the literature data in both Cho et al. (2017) and Yao et al. (2017) may slightly underestimate dendritic mitochondrial sizes by the presence of some axonal mitochondria in the data sets and by the fact that axonal mitochondria are in average a bit smaller than dendritic ones - as also reflected in our data ($0.6 \mu\text{m}$ vs. $1.0 \mu\text{m}$; **Figure 3e vs. Extended Data Fig. 3-2 in the newly revised manuscript**). In pictures of the dendritic arbor of developing neurons, such as those used by Cho et al. (2017), we quantified the axonal contribution to be only 4% of all mitochondria in the neurites. Even if larger parts of the axon would be visible and the axonal contribution to the mean e.g. were 10%, this – based on a size difference of axonal vs. dendritic mitochondria of $0.4 \mu\text{m}$ – would only lead to a mean for dendritic mitochondria of about $1.04 \mu\text{m}$ (instead of the reported $1.0 \mu\text{m}$ of all neuritic mitochondria). Thus, the effect of the inclusion of a few axonal measurements in Cho et al's data clearly does not preclude a comparison of the literature data to the data reported in our manuscript and therefore our average length of $1.0 \mu\text{m}$ (**revised Fig. 3e**) is very well in line with the literature data reported by Yao et al. (2017) or Cho et al. (2017).

Additional length distribution analyses (**Fig. 3f added to the revised manuscript**) also demonstrate that our data is quite similar to the literature data. 68% of our mitochondria are smaller than $1 \mu\text{m}$ in length, whereas Cho et al. (2017) (including a 5-10% contribution of axonal mitochondria) reported that more than 80% of all mitochondria are smaller than $1 \mu\text{m}$. Cho et al. furthermore reported that 14% are between 1 and $2 \mu\text{m}$ (our data, 18%). Sizes between 2 and $3 \mu\text{m}$ make up only 3% in Cho et al. (our data, 10%). Sizes between 3 and $4 \mu\text{m}$ were too rare to be depicted in Cho et al. but we observed that 2% of our mitochondria were so large (**please see also Reviewer Figure below**).

Fig. 3f Tröger et al.

68% of all control mitochondria have a length of less than 1 μm

Suppl Fig. Cho et al., Nat Comm 2017

80% of all contro mitochondria have a length of less than 1 μm

Mito-length (μm)

Reviewer Figure 2. Distributions of mitochondrial length obtained by measurements based on overexpression of fluorescent protein inside of mitochondria in our study (Tröger et al.) vs. literature data copied from Cho et al. (2017, *Nat. Commun.*).

Note the similarity, as Cho et al. (2017) (including a 5-10% contribution of axonal mitochondria) also reported that more than 80% of all mitochondria are smaller than 1 μm in length (our data, 68%), 14% were between 1 and 2 μm (our data, 18%), mitochondria with sizes between 2 and 3 μm makes up 3% (our data, 10%) and mitochondria with sizes between 3 and 4 μm were too rare to be depicted in Cho et al. (our data, 2%).

Since reviewer1, who brought up similar questions in the first revision, explicitly approved the mitochondrial length data sets after the additional work using MitoTracker and mitoGFP in the first revision, we moved the original cytochrome c-based data into the Supplementary Information (**revised Extended Data Fig. 3-1**).

We hope that the reviewer will agree with us that this solution makes it possible to prominently display mitochondrial length data reflecting the higher literature values obtained by overexpressing fluorescent proteins and the use of the skeleton software tool without omitting other, also positively peer-reviewed data.

We also included some considerations concerning the methods of mitochondrial length evaluation including the technical limitations voiced by reviewer2 in the Material and Method section of the revised manuscript.

As the reviewer will acknowledge, if different techniques reveal the same types of phenotypes, this enormously solidifies the cell biological insights achieved by a scientific study.

Going back to the EM images in the manuscript, mitochondria were only 0.4 μm because they were seemingly cut in the transverse (cross) plane, judging from their round shape. Mitochondria in the longitudinal plane of dendrites should appear tubular (Popov et al., *J. Comp Neurol.* 2005). It is also unclear whether the mitochondria being measured were in the dendrites. More detailed analyses and

methods (which layer and at what angle were observed, how dendrites were identified) should be provided.

We improved the descriptions in the Material and Method section of the newly revised manuscript. The brain samples were from 1 mm x 1 mm x 1 mm samples of the CA3 region of the hippocampus. Sectioning of these brain pieces was done in random orientations. In brain tissue, mitochondria cannot exclusively be cut in transverse planes of the mitochondria but EM sectioning cuts cells in all random orientations. Furthermore, our quantitative electron microscopic analyses were blinded. We additionally specified that the outer mitochondrial membrane diameter was used for measurements.

We hope that the reviewer will be content with the further experimental details now reported in the Material and Method section of the newly revised manuscript

Mitochondrial velocity

‘Mitochondria movement’ in neuronal processes has long been studied and usually refers to microtubule-dependent transport (the ‘fast movement’ in this paper). Previous studies have generally classified mitochondria into motile and stationary, but here again the authors challenge this and divide them into fast- and slow- moving. The authors rename what has been considered ‘stationary’ as ‘slow-moving’, which should be clarify in the text as it is confusing if there are the third class slow-moving mitochondria in addition to the fast moving and stationary ones.

The authors describe the average speed of all mitochondrial movements (~0.07 μm), which I believe should be removed for two reasons: At this time resolution, the fast-movement cannot be measured correctly as the authors admitted. It also does not seem to make sense to represent the average speed of constant directional transport and non-directional short-step diffusive motion.

It would be more helpful for readers if the authors clarify the new categories in this paper by showing the distribution of actual speed and directionality of individual mitochondria.

We agree with the reviewer that averaging velocity data for both very different subgroups of mitochondria (directionally transported mitochondria vs. non-directionally moving mitochondria reflecting short-step diffusive motions) is not informative. **As suggested by the reviewer, we have therefore omitted these overall examinations (former Fig. 3m and Ext. Data Fig. 3-5 added in the first revision) from the newly revised manuscript. The misleading terminology that may also confuse readers is therefore not used in the revised manuscript anymore.**

As emphasized in the above general statements of reviewer2, it is indeed more important to focus on the tethered mitochondria at specifically the branch induction sites and at the disruption of their local positioning and confinement by rhotekin2 RNAi and to try to validate the observations at such sites with experimentation at higher temporal resolution. **As also already mentioned above, we have therefore conducted sets of further 3D-live microscopy experiments with faster imaging at the expense of reduced z information that are now reported in the newly added Extended Data Fig. 6-1 of the revised manuscript.**

The reviewer will be happy to see that these examinations fully confirmed the previously presented data and phenotypes by independent experimentation.

Axonal mitochondria

Tethering of mitochondria in axonal branching points are well studied and the authors should at least discuss the possible function of RTKN2 and syndapin1 in the axon. The authors have newly measured the mitochondrial length in the axons and shows that they are longer in RTKN2 KD. In normal cells, mitochondria in axons should be shorter than those in dendrites, but they are almost the same length here, raising doubts about whether the measurements are accurate. Anyway, as mitochondrial length

and tethering are not related, the effect of RTKN2 loss on tethering should be examined or at least discussed as it is the main conclusion of this study.

In order to comply with the comment of the reviewer, we have added more axonal information to the newly revised manuscript. This includes the identified axonal length loss-of-function phenotype of rhotekin2 (new Extended Data Fig. 3-2) and the discussion of this data, which was previously merely reported as Reviewer Figure in the first revision.

As far as the values of the mitochondrial length measurements are concerned, we would kindly like to refer the reviewer to the data reported in **Extended Data Fig. 3-2** and in **Fig. 3**. **The mean mitochondrial length was determined to be 0.6 μm in axons and 1.0 μm in dendrites of control cells based on mitoGFP measurements.**

Thus, similar to the literature, also our examinations show that axonal mitochondria are smaller than dendritic ones leaving no doubt on the accuracy of our examinations.

Dendrite morphometry of cultured hippocampal neurons

The authors stated that ‘primary cultures prepared from the hippocampus are mostly reflecting pyramidal neurons’, but this is incorrect. Hippocampal primary cultures prepared from both mice and rats contain significant proportions of non-pyramidal neurons which have very different dendritic arbor morphologies (Horton et al., 2007; Wu et al., 2015). I was not able to find the culture protocol used in this paper, as the reference cited as describing detailed methods still omit detailed methods by citing another paper, and another paper did the same. If the protocol by Banker’s group was used, the culture likely contains significant proportions of granule cells and interneurons.

We expanded the Material and Method section of the revised manuscript. It now describes the preparation and maintenance of primary rat hippocampal neurons instead of citing literature. It thereby now provides more directly accessible information.

As far as the percentages of interneurons and granule cells are concerned, the reviewer may appreciate that we immunostained and quantified the fractions of the different neuronal types using anti-Math2 (pyramidal cells, as the reviewer suggested) and antibodies against the established markers GAD67 (interneurons) and PROX1 (granule cells). Our E18 rat hippocampal cultures at DIV7 usually show the following neuronal composition: 69% pyramidal cells, 25% granule cells, 2% interneurons (and 4% unclear). **Thus, pyramidal cells represent the vast majority of cells in the primary cultures, as mentioned in the first answer to reviewer2 (please see the Reviewer Figure below for visualization of the quantitative data we obtained).**

Reviewer Figure 3. Neuronal composition of E18 rat hippocampal cultures used.

Quantitative determinations of the neuronal composition of rat E18 cultures at DIV7 by immunostainings against established markers (anti-Math2, pyramidal cells; anti-GAD67, interneurons; anti-PROX1, granule cells). Evaluations were based on random imaging and systematic counting. 4% of the anti-MAP2 stained neurons show unclear results concerning the three listed markers and are not depicted in the graph. For exact percent

numbers of interneurons (2%), pyramidal cells (69%) and granule cells (25%) also see the figure. Data, mean±SEM. N=28 low magnification images (about 200 evaluated neurons each).

The morphometries in Fig.8 may warrant reexamination.

For your reference in future analyses, Map2 is a good marker of dendrites, but it is not suitable for measuring dendritic length, as it does not fill the entire dendritic length. The author may want to reexamine Fig.8e if it was based on Map2 staining.

In order to comply with the reviewer suggesting that the morphologies in Fig.8 should better be reexamined by the use of cell filler in order to exclude that anti-MAP2 immunostaining may not fill dendrites to the very end, we reanalyzed the dendritic arbor parameters by using GFP as cell filler to trace dendrites (defined as explicitly being dendrites by showing a presence of the dendritic marker MAP2).

GFP fluorescence is bright and fills cells entirely, i.e. is a suitable cell filler (whereas e.g. Math2 (NEUROD6) suggested by the reviewer in the first review is a member of the NEUROD family of basic helix-loop-helix transcription factors and resides in the nucleus).

The newly added GFP-based data is shown as Extended Data Fig. 8-2 in the newly revised manuscript. The GFP-based quantitative analyses of neuronal morphologies yielded the same results as the MAP2-based analyses (**Extended Data Fig. 8-2a-c vs. Fig. 8e-g**).

Importantly, the newly added data also includes a cell filler-based reevaluation of the parameter *total dendritic length*, which the reviewer highlighted as most critical (newly added Extended Data Fig. 8-2a). Also the absolute values obtained for dendritic length were almost identical when both methods of evaluation were compared (**Extended Data Fig. 8-2a vs. Fig. 8e**).

Point-to-point responses to REVIEWERS' COMMENTS

Reviewer #1 (Remarks to the Author):

Although the authors cannot fully address all questions due to technical limitations, they produced new data with high resolution SIM imaging, and with increased temporal resolution. Overall, I am satisfied that the authors have addressed most of the concerns of the reviewers.

We thank the reviewer for his/her positive assessment of our work added during revision 2 and are happy to see that the reviewer is satisfied by the fact that we were able to address most of the concerns of the reviewers by the additional experimentation included in the newly revised manuscript.

One point that remains unclear and un-discussed from my point of view is the fact that Rhotekin2 is only highly expressed until P8. While I'm not advocating for more experiments to look at Rhotekin2's role in later developmental stages or maturation, I think a discussion of this point (along with how it fits with syndapin-1) is warranted since the authors want to link it to both neurodevelopmental and neurodegenerative disorders in the last paragraph.

The reviewer is referring to **Extended Data Figure 4-1** showing a strong expression of rhotekin2 from E16 to P8 and declining but still persistent expression levels at P12, 4 weeks and 8 weeks (Results section) as well as to the discussion.

The revised manuscript now does not only discuss how well the highest rhotekin2 expression levels correlate with the ages of brain development as before but also points out that the lower levels of rhotekin2 during higher ages are well in line with the reduced dynamics of the dendritic arbor in fully developed brains (**please see both the Results section and the Discussion of the revised manuscript**).

The revised manuscript now furthermore points out that defects of dendritic arborization found in *syndapin 1* KO mice and upon expression of a schizophrenia-associated syndapin I mutant may be reflected by basal levels of syndapin I already detectable during early postnatal stages, while the subsequently rising syndapin I levels are reflecting syndapin I's additional roles in the forming synaptic connections (**please see the Results section and the Discussion of the revised manuscript**).

Considering the question of the reviewer, it furthermore seemed to be useful to point out in more obvious manner that the isolation of mitochondria shown in Figure 2e, which shows endogenous rhotekin2 as well as its enrichment in mitochondrial fractions, was conducted from brain material of adult rats (about 20 weeks). As already indicated by e.g. the expression in 4 weeks and 8 weeks old mice, rhotekin2 expression thus clearly persists well beyond the dendritogenesis stages in brain development and can also be detected in much later phases of life. **Thus far, the information that the fractionation and mitochondria isolation experiments were done with adult brains was only given in the Material and Method section. In order to point out the persisting expression of rhotekin2 during adulthood more clearly, we now repeat this information in both the Results section and in the legend of Figure 2 of the revised manuscript.**